# Learning and Transferring Sparse Contextual Bigrams with Linear Transformers

**Yunwei Ren**[*]   **Zixuan Wang**[*]   **Jason D. Lee**
Princeton University
{yunwei.ren, wangzx, jasonlee}@princeton.edu

## Abstract

Transformers have excelled in natural language modeling and one reason behind this success is their exceptional ability to combine contextual informal and global knowledge. However, the theoretical basis remains unclear. In this paper, first we introduce the Sparse Contextual Bigram (SCB), a natural extension of the classical bigram model, where the next token's generation depends on a sparse set of earlier positions determined by the last token. We then analyze the training dynamics and sample complexity of learning SCB using a one-layer linear transformer with a gradient-based algorithm. We show that when trained from scratch, the training process can be split into an initial sample-intensive stage where the correlation is boosted from zero to a nontrivial value, followed by a more sample-efficient stage of further improvement. Additionally, we prove that, provided a nontrivial correlation between the downstream and pretraining tasks, finetuning from a pretrained model allows us to bypass the initial sample-intensive stage. We also empirically demonstrate that our algorithm can outperform SGD in this setting and discuss its relationship with the usual softmax-based transformers.

## 1   Introduction

Transformers have played a central role in modern deep learning, achieving significant success across various fields, including language modeling (OpenAI, 2023), computer vision (Dosovitskiy et al., 2020), and natural sciences (Jumper et al., 2021). The core of transformers is the self-attention layer (Vaswani et al., 2017), which can attend to any subset of the input sequence to output a weighted linear combination of the (transformed) tokens.

Several capabilities of the transformers contribute to their success in language modeling. First, they can extract contextual information from the input token sequences, which is essential in some arithmetic tasks (Edelman et al., 2022; Liu et al., 2022; Nanda et al., 2023; Yao et al., 2021). In addition, transformers can memorize global in-domain knowledge (Petroni et al., 2019; Zhang et al., 2023a; Haviv et al., 2022; Carlini et al., 2021). These two abilities combined enable transformers to predict the next token based on the *in-context* information as well as *global* knowledge (OpenAI, 2023) acquired during training.

To theoretically understand how transformers learn both capabilities, we propose a minimalist data-generating model, the **Sparse Contextual Bigram** (SCB). This model builds on the classical bigram model and requires learning both contextual information and the (global) transition probabilities. Here, the next token depends on the transition matrix $P$ and a sparse set of prior tokens that is determined by the last token. In particular, SCB can be represented by a one-layer linear transformer — a simplified architecture that can serve as an abstraction for studying transformer optimization (Ahn et al., 2023), which makes it suitable for theoretical analysis.

---

[*]Equal contribution.

38th Conference on Neural Information Processing Systems (NeurIPS 2024).

In this paper, we investigate the training dynamics and sample complexity of training a linear transformer to learn the SCB task using a stochastic gradient-based algorithm. Our contributions are summarized as follows:

- **Data model:** We introduce the **Sparse Contextual Bigram** (SCB) model, a simple task that requires the model to learn both in-context and global information.
- **Convergence:** We prove convergence guarantees for a one-layer linear transformer trained on with the nonconvex $\ell_1$-regularized MSE loss using preconditioned projected proximal descent, given a dataset sampled from the SCB model.
- **Sample Complexity:** Under mild conditions on the data distribution, initialization, and hyperparameters, we prove that our algorithm can recover the ground-truth with polynomial dependence on the sequence length $T$, number of states $N$, and the sparsity parameter $Q \ll T$. We show that the training first goes through an initial sample-intensive stage which boosts the signal with $\text{poly}(T)$ samples, followed by a more sample-efficient stage to achieve final convergence with $\text{poly}(N, Q)$ samples. We empirically verify that our gradient-based methods converge to the ground truth with a small batch size, while unregularized stochastic gradient descent fails due to the large variance.
- **Transfer Learning:** We prove that, when there is a nontrivial correlation between the pretraining and downstream tasks, we can transfer a pre-trained model to bypass the first sample intensive stage, so that our algorithm converges to the ground truth of the downstream task with only $\text{poly}(N, Q)$ samples.

## 1.1 Related works

**Training dynamics of transformers.** Several works have studied the learnability aspects of specific transformer architectures. Jelassi et al. (2022) demonstrated that a Vision Transformer (ViT) (Dosovitskiy et al., 2020) trained through GD, augmented with positional-embedding attention matrices, can effectively capture spatial structures. Li et al. (2023) investigated the sample complexity necessary to achieve good generalization performance on a similar ViT model. Tarzanagh et al. (2023) established a connection between the optimization landscape of self-attention and the formulation of a hard-margin Support Vector Machine (SVM) problem that separates and selects specific optimal tokens and established global convergence under strong assumptions. Tian et al. (2023a,b) provided insights into the training dynamics of the self-attention and MLP layers, respectively, although they did not establish convergence guarantees.

Another line of work focuses on the training dynamics of in-context learning. Mahankali et al. (2023) was among the first to introduce linear regression as an in-context learning task, while Zhang et al. (2023b) proved global convergence of gradient flow for a single-layer linear self-attention layer on this task. Huang et al. (2023) provided a convergence guarantee for a one-layer transformer with softmax attention on a similar task where the in-context tokens are drawn from a specific data distribution. Chen et al. (2024) generalized the single-task linear regression task to a multi-task setting and proved the global convergence of multi-head attention architecture using gradient flow on the population loss with specific initialization. In contrast, our work focuses on the language modeling ability of transformers instead of their in-context learning ability.

Several recent works analyzed transformers from a Markov chain perspective. Bietti et al. (2024) studied the in-context bigram (phrased as *induction head*) from an associative memory viewpoint. Nichani et al. (2024) proved that a simplified two-layer transformer can learn the induction head and generalize it to certain latent causal graphs. Edelman et al. (2024) further investigated training process on bigram and general $n$-gram tasks, and observed multi-phase dynamics. Makkuva et al. (2024) studied the loss landscape of transformers trained on sequences sampled from a single Markov Chain. Our SCB model extends the classical bigram models to allow context-dependent sparse attention on previous tokens.

Several works, including Tian et al. (2023a); Zhang et al. (2023b); Huang et al. (2023); Tarzanagh et al. (2023); Nichani et al. (2024); Kim and Suzuki (2024), and ours, use a similar reparameterization, consolidating the key and query matrices into a single matrix $W$ to simplify the dynamics of the training process. Most previous studies (Tian et al., 2023a; Zhang et al., 2023b; Huang et al., 2023; Tarzanagh et al., 2023; Nichani et al., 2024; Kim and Suzuki, 2024; Wang et al., 2024; Chen et al., 2024) uses population loss to simplify the analysis. In contrast, our work goes beyond the population

loss to analyze the sample complexity of the stochastic gradient descent dynamics. Although Li et al. (2023) also investigated the sample complexity on a different task, their model requires a pre-trained initialization, while our model is trained from scratch.

**Transfer Learning.** Transfer learning (Devlin et al., 2018) has gained significant attention in this deep learning era. From a theoretical perspective, several works have investigated the statistical guarantees of transfer learning from the representation learning perspective (Tripuraneni et al., 2020; Du et al., 2020; Arora et al., 2019; Hanneke et al., 2023). Recent studies on transfer learning mostly focus on linear models (Li et al., 2022; Tian and Feng, 2023; Fei and Li, 2021; Zhang et al., 2022; Ju et al., 2023; Dar and Baraniuk, 2022). For dynamics of transfer learning, Lampinen and Ganguli (2018) studied the behaviors of multi-layer linear networks in a teacher-student setting, while Dhifallah and Lu (2021) analyzed single-layer perceptrons. Damian et al. (2022) showed that a two-layer neural network can efficiently learn polynomials dependent on a few directions, enabling transfer learning.

To the best of our knowledge, this is the first work studying transfer learning for transformers Moreover, unlike previous works that assume a shared structure between the pretraining and downstream tasks, we only require them to have a non-trivial correlation, which is a much weaker assumption.

## 1.2 Outline of this paper

In Section 2 we formalize the problem setup, including the SCB task, the transformer architecture, and the training algorithm. Section 3 consists of our main results, and we analyze the population dynamics to provide intuitions. Section 4 contains our transfer learning results. Experimental results can be found in Section 5.

## 2 Setup

In this section, we describe our data-generating model, the one-layer linear transformer architecture, and the training algorithm.

**Notations.** We use $[T]$ to denote the set $\{1, 2, ..., T\}$. Matrices and vectors are denoted in upper-case bold letters ($\boldsymbol{A}, \boldsymbol{V}, \boldsymbol{\Delta}$, etc.) and lower-case bold letters ($\boldsymbol{a}, \boldsymbol{q}$, etc.), respectively. For norm, $\|\cdot\|$ denotes $\ell_2$ norm and $\|\cdot\|_F$ denotes the Frobenius norm. Additionally, for $\boldsymbol{\mu} \in \mathbb{R}^N$, $\|\boldsymbol{A}\|_{\boldsymbol{\mu}}$ denotes $\mu$-norm for matrix $\boldsymbol{A} \in \mathbb{R}^{d \times N}$ for arbitrary $d$, which is defined as $\|\boldsymbol{A}\|_{\boldsymbol{\mu}}^2 := \text{Tr}\left(\boldsymbol{A}\text{diag}(\boldsymbol{\mu})\boldsymbol{A}^\top\right)$. We use $\mathbb{1}\{\cdot\}$ to denote the indicator function. We use $\tilde{O}(\cdot)$ to hide logarithmic factors in the asymptotic notations.

### 2.1 Data-generating model: Sparse Contextual Bigram

The bigram model, where the next token depends only on the current one, is arguably one of the simplest language models. To learn this model, it suffices to learn the transition probabilities $\boldsymbol{P} \in \mathbb{R}^{N \times N}$ where $P_{n,m} = \mathbb{P}[X_{t+1} = n \mid X_t = m]$, which is achievable through a linear model (0-layer transformer).

A natural way to extend the classical bigram model is to allow the next token to depend on a context-dependent set of previous tokens. This extension can model situations such as generating the words after the phrase "by Theorem 3.2", which requires us to retrieve the statement of "Theorem 3.2". Here, we propose a simple extension of this type, which we call the **Sparse Contextual Bigram** (SCB). The contextual information is encoded by a sparse probability vector $\boldsymbol{q}$ determined by the last token. To generate the next token, the model retrieves the tokens referenced by $\boldsymbol{q}$ and applies the transition matrix $\boldsymbol{P}$ (global knowledge) to one of them according to the distribution $\boldsymbol{q}$.

Formally, our data-generating model SCB can be described as follows. Let $T$ be the sequence length and $[N]$ the vocabulary. Let $\boldsymbol{P} \in \mathbb{R}^{N \times N}$ be a transition matrix, with column[1] $\boldsymbol{P}_k$ being the transition probability vector of token $k$. Suppose that $\boldsymbol{\mu} \in \mathbb{R}_{\geq 0}^N$ is the stationary distribution of $\boldsymbol{P}$.

Each input sequence consists of $T + 1$ tokens $(x_1, \ldots, x_{T+1})$, i.i.d. sampled from distribution $\boldsymbol{\mu}$. The output token (label) is generated as follows. For each $k \in [N]$, there is a probability vector $\boldsymbol{q}^{(k)} \in R_{\geq 0}^T$ that represents the tokens the model needs to attend to when the last token $x_{T+1}$ is $k$. For notational

---

[1]This differs from the convention of the usual Markov Chain literature where the rows are the transition probability vectors. We use this convention as it is more compatible with our notations.

simplicity, we will write $\boldsymbol{x} = (x_1, \ldots, x_T)$, $\boldsymbol{X} = (\boldsymbol{e}_{x_1}, \ldots, \boldsymbol{e}_{x_T})$ and $\boldsymbol{Q} = (\boldsymbol{q}^{(1)}, \ldots, \boldsymbol{q}^{(N)}) \in \mathbb{R}^{T \times N}$. When $x_{T+1} = k$, the output token $x_o$ is sampled from the distribution

$$\mathbb{P}(x_o = n \mid x_{T+1} = k, \boldsymbol{x}) = \sum_{t=1}^{T} q_t^{(k)} P_{n,x_t}, \quad \forall n \in [N]. \tag{1}$$

In words, we first sample a position $s \in [T]$ according to $\boldsymbol{q}^{(k)}$ and then run one step of the Markov Chain $\boldsymbol{P}$ from $x_s$ to generate $x_o$. Note that this model can be represented by a one-layer linear transformer (see the next subsection for details). We make the following assumptions on SCB task.

**Assumption 2.1** (Sparse Contextual Bigram, SCB). *In the* SCB *task, we assume the following:*

(a) *(Q-sparse) For some $Q \ll T$ and each of $\boldsymbol{q}^{(k)}$, at most $Q$ entries are nonzero.*

(b) *(Well-conditioned) There exists some constant $C \geq 1$ such that for every $k \in [N]$ and $t \in [T]$, $q_t^{(k)} \in [1/(CQ), C/Q]$ if it is nonzero, and $\mu_k \in [1/(CN), C/N]$.*

(c) *(Nontrivial transition) $\|\boldsymbol{P}\|_\mu^2 - \|\boldsymbol{\mu}\|^2 \geq \|\boldsymbol{\mu}\|^2$.*

(d) *(Long sequence) $T \geq (NQ)^{10}$.*

**Remark on condition (c).** We say the transition $\boldsymbol{P}$ is trivial if the transition probability vectors are all the same, i.e., $\boldsymbol{P} = \boldsymbol{\mu}\mathbf{1}^\top$. In this case, we have $\|\boldsymbol{P}\|_\mu^2 = \langle \boldsymbol{\mu}\mathbf{1}^\top, \boldsymbol{\mu}\boldsymbol{\mu}^\top \rangle = \|\boldsymbol{\mu}\|^2$. Requiring $\|\boldsymbol{P}\|_\mu^2 - \|\boldsymbol{\mu}\|^2 \geq \|\boldsymbol{\mu}\|^2$ rules out situations where $\boldsymbol{P}$ is too close to the trivial one. Also, note that for any well-conditioned $\boldsymbol{\mu}$, we have $\|\boldsymbol{\mu}\|^2 \geq \Omega(1/N)$. ♣

In this work, we focus on the case where $(\boldsymbol{x}, x_{T+1}, x_o)$ are given as (one data point of) the training data with $(x_1, \ldots, x_{T+1})$ i.i.d. sampled from $\mu$. The SCB task can be extended to a sequence-to-sequence model: we drop $x_1$ and append $x_o$ to get a new input sequence $(x_2, \ldots, x_{T+1}, x_o)$, and then repeat the same sampling procedure to generate another token. This generates a sequence $(x_t)_{t=1}^{\infty}$ where $(x_{T+2}, x_{T+1}, \ldots)$ are not independent, and this makes our model a true language model. We leave the study of the more complicated learning-from-$(x_t)_{t=1}^{\infty}$ task to future works.

## 2.2 Transformer architecture

Our learner model is a one-layer single-head linear transformer (Akyürek et al., 2022; Zhang et al., 2023b; Ahn et al., 2023). A general linear transformer can be expressed as: $\boldsymbol{F}(\boldsymbol{x}, x_{T+1}; \boldsymbol{V}, \boldsymbol{A}) = \boldsymbol{V}\boldsymbol{E}(\boldsymbol{E}^\top \boldsymbol{A} \boldsymbol{E})$, where $\boldsymbol{E}$ is the embedding of the input tokens and positions, and $\boldsymbol{A}, \boldsymbol{V}$ are the parameters of the attention and output layers, respectively. In our setting, we only need a simpler model:

$$\boldsymbol{F}(\boldsymbol{x}, x_{T+1}; \boldsymbol{V}, \boldsymbol{A}) := \boldsymbol{V}\boldsymbol{X}\left(\boldsymbol{I}_T \boldsymbol{A} \boldsymbol{e}_{x_{T+1}}\right) =: \boldsymbol{V}\boldsymbol{X}\boldsymbol{a}^{(x_{T+1})}, \tag{2}$$

where $\boldsymbol{V} \in \mathbb{R}^{N \times N}$ and $\boldsymbol{A} \in \mathbb{R}^{T \times N}$ are the trainable parameters, and $\boldsymbol{a}^{(k)}$ denotes the $k$-th column of $\boldsymbol{A}$. This model uses cross-attention (replacing the last $\boldsymbol{E}$ with $\boldsymbol{e}_{x_{T+1}}$), uses only the positional embeddings together with the last token to compute the attention weights (replacing the second $\boldsymbol{E}$ with $\boldsymbol{I}_T$), and discards the positional embeddings in the output layer (replacing the first $\boldsymbol{E}$ with $\boldsymbol{X}$). This is equivalent to manually set certain blocks in the weight matrices to 0, which is a common practice in the theoretical literature to simplify the analysis (Nichani et al., 2024; Huang et al., 2023; Zhang et al., 2023b).

Note that our data-generating model (1) can be represented using (2) by setting $\boldsymbol{A} = \boldsymbol{Q}$ and $\boldsymbol{V} = \boldsymbol{P}$. We will show that a modified version of SGD can approximately recover this ground-truth model.

## 2.3 Training algorithm

We assume that the stationary distribution $\boldsymbol{\mu}$ and certain norms of the ground-truth $\boldsymbol{P}$ and $\boldsymbol{Q}$ are known when choosing the initialization and learning rate. The goal here is to recover $\boldsymbol{P}$ and $\boldsymbol{Q}$. Our loss function is the $\ell_1$-regularized MSE loss. The standard way to optimize an $\ell_1$-regularized loss is to use the proximal gradient descent. We adopt this algorithm with several additional pre-conditioning and a projection step to ensure some basic properties.

Formally, let the per-sample loss be defined as

$$l(\boldsymbol{x}, x_{T+1}, x_o; \boldsymbol{V}, \boldsymbol{A}) := \frac{1}{2} \left\| \boldsymbol{e}_{x_o} - \boldsymbol{V}\boldsymbol{X}\boldsymbol{A}\boldsymbol{e}_{x_{T+1}} \right\|^2. \tag{3}$$

We initialize $\boldsymbol{A} = \mathbf{1}_T \mathbf{1}_N^\top / T$ to have uniform attention and $\boldsymbol{V} = \boldsymbol{\mu}\mathbf{1}^\top$ to be the trivial transition. At each step $\tau \geq 0$, we sample $B_\tau$ fresh samples $\{\boldsymbol{x}^{(i)}, x_{T+1}^{(i)}, x_o^{(i)}\}_{i=1}^{B_\tau}$ to form a mini-batch. The $\ell_1$-regularized mini-batch loss is defined as

$$l_{\text{reg}}^{(B_\tau, \lambda)} \left( \{\boldsymbol{x}^{(i)}, x_{T+1}^{(i)}, x_o^{(i)}\}_{i=1}^{B_\tau}; \boldsymbol{V}, \boldsymbol{A} \right) := \frac{1}{B_\tau} \sum_{i=1}^{B_\tau} l(\boldsymbol{x}^{(i)}, x_{T+1}^{(i)}, x_o^{(i)}; \boldsymbol{V}, \boldsymbol{A}) + \lambda \sum_{k=1}^{N} \left\| \boldsymbol{a}^{(k)} \right\|_1,$$

where $\lambda > 0$ is a parameter that controls the strength of regularization. Let $\nabla_{\boldsymbol{V}}^{(B_\tau)} l$ and $\nabla_{\boldsymbol{A}}^{(B_\tau)} l$ denote the mini-batch gradients of the original $l$ w.r.t. $\boldsymbol{V}$ and $\boldsymbol{A}$, respectively. We then define the preconditioned gradients as

$$\begin{aligned}
\hat{\nabla}_{\boldsymbol{V}}^{(B_\tau)} l &:= \left( \boldsymbol{I}_N - \frac{\mathbf{1}_N \mathbf{1}_N^\top}{N} \right) \left( \nabla_{\boldsymbol{V}}^{(B_\tau)} l \right) \text{diag}(1/\boldsymbol{\mu}) \left( \boldsymbol{I}_N - \frac{\boldsymbol{\mu}\boldsymbol{\mu}^\top}{\|\boldsymbol{\mu}\|^2} \right), \\
\hat{\nabla}_{\boldsymbol{a}^{(k)}}^{(B_\tau)} l &:= \frac{1}{\mu_k} \left( \boldsymbol{I}_T - \frac{\mathbf{1}_T \mathbf{1}_T^\top}{T} \right) \left( \nabla_{\boldsymbol{a}^{(k)}}^{(B_\tau)} l \right), \quad \forall k \in [N].
\end{aligned} \tag{4}$$

Here, the $1/\boldsymbol{\mu}$ rescaling plays a role similar to importance sampling. We multiply $\nabla_{\boldsymbol{V}}^{(B_\tau)} l$ with $\boldsymbol{I} - \mathbf{1}\mathbf{1}^\top/N$ and $\boldsymbol{I} - \boldsymbol{\mu}\boldsymbol{\mu}^\top/\|\boldsymbol{\mu}\|^2$ to ensure at least $\mathbf{1}_N^\top \boldsymbol{V} = \mathbf{1}_N^\top, \mathbf{1}_T^\top \boldsymbol{A} = \mathbf{1}_N^\top$, and $\boldsymbol{V}\boldsymbol{\mu} = \boldsymbol{\mu}$ always hold throughout training. Note that we project each column of $\boldsymbol{V}$ to the affine space $\{\boldsymbol{v} \in \mathbb{R}^T : \mathbf{1}^\top \boldsymbol{v} = 1\}$ instead of the probability simplex. This is sufficient for our analysis and is much easier to compute than the latter. We update the output layer using

$$\boldsymbol{V}_{\tau+1} = \boldsymbol{V}_\tau - \eta_{\boldsymbol{V}} \hat{\nabla}_{\boldsymbol{V}}^{(B_\tau)} l, \tag{5}$$

where $\eta_{\boldsymbol{V}} > 0$ is the step size. Now, consider the attention layer. Due to the existence of the $\ell_1$-regularization, the update rule becomes a simple variant of the standard proximal gradient descent. Formally, for step size $\eta_A > 0$, each $k \in [N]$ and $t \in [T]$, we have

$$\begin{aligned}
\boldsymbol{a}_{\tau+1}^{(k,')} &= \boldsymbol{a}_\tau^{(k)} - \frac{\eta_A}{\mu_k} \nabla_{\boldsymbol{a}^{(k)}}^{(B_\tau)} l, && \text{(preconditioned GD step)}, \\
a_{t,\tau+1}^{(k,'')} &= \begin{cases} a_{t,\tau+1}^{(k,')} - \lambda, & \text{if } a_{t,\tau+1}^{(k,')} \geq \lambda, \\ 0, & \text{if } \left| a_{t,\tau+1}^{(k,')} \right| \leq \lambda, \end{cases} && \text{(proximal step)}, \\
\boldsymbol{a}_{\tau+1}^{(k)} &= \text{Proj}_{\{\mathbf{1}^\top \boldsymbol{a}=1\}} \left( \boldsymbol{a}_{\tau+1}^{(k,'')} \right) = \boldsymbol{a}_{\tau+1}^{(k,'')} + \left( 1 - \mathbf{1}^\top \boldsymbol{a}_{\tau+1}^{(k,'')} \right) \frac{\mathbf{1}}{T}, && \text{(projection step)}.
\end{aligned} \tag{6}$$

For the proximal step, we will later show that no $a_t^{(k)}$ can ever become smaller than $-\lambda$, so it suffices to consider those two cases. During the proximal step, all small $a_t^{(k)}$ are set to 0, and $\lambda$ is subtracted from all large coordinates. For notational simplicity, we define $\boldsymbol{g}_{\lambda,\tau}^{(k)} := -\eta_A^{-1}(\boldsymbol{a}_{\tau+1}^{(k)} - \boldsymbol{a}_\tau^{(k)})$ so that we can write the update as $\boldsymbol{a}_{\tau+1}^{(k)} = \boldsymbol{a}_\tau^{(k)} - \eta_A \boldsymbol{g}_\tau^{(k)}$. We will choose $\lambda = 0$ in certain stages of training. In this case, (6) becomes the usual projected preconditioned gradient descent and we have

$$\boldsymbol{a}_{\tau+1}^{(k)} = \boldsymbol{a}_\tau^{(k)} - \eta_A \hat{\nabla}_{\boldsymbol{a}^{(k)}}^{(B_\tau)} l \qquad \text{(when } \lambda = 0\text{)}.$$

Our algorithm consists of three stages with different hyperparameters being used in different stages and certain rounding required between stages. The pseudocode is given in Algorithm 1 and more details on the projection/normalization steps are provided in Appendix E and F. When we train the model from scratch, all three stages are used and the initialization is $\boldsymbol{V}_0 = \boldsymbol{\mu}\mathbf{1}^\top$ and $\boldsymbol{A} = \mathbf{1}_T \mathbf{1}_N^\top / T$.

**Transfer learning.** When doing transfer learning, the initialization will be obtained from the weights of the pre-trained model and one step of gradient update. Then, we will run Algorithm 1 from Stage 2.

---

**Algorithm 1** Projected preconditioned $\ell_1$-proximal gradient descent

---

**Input:** Stationary distribution $\mu$; initialization $V_0, A_0$; learning rates $\eta_A^{(i)}, \eta_V^{(i)}, i \in [3]$; threshold $\lambda_0$; regularization strength $\hat{\lambda}$; times $\mathcal{T}_1, \mathcal{T}_2, \mathcal{T}_3$

   **Stage 1:** Run (5) and (6) with $\eta_A = \eta_A^{(1)}, \eta_V = \eta_V^{(1)}, \lambda = 0$ for $\mathcal{T}_1$ steps;

   **Thresholding-projection:** $\forall k \in [n], \hat{\boldsymbol{a}}^{(k)} = [a_t^{(k)} \mathbb{1}\{a_t^{(k)} \geq \lambda_0\}]_t, \boldsymbol{a}^{(k)} \leftarrow (\boldsymbol{I}_T - \boldsymbol{1}_T \boldsymbol{1}_T^\top / T) \hat{\boldsymbol{a}}^{(k)}$

   **Stage 2:** Run (5) and (6) with $\eta_A = \eta_A^{(2)}, \eta_V = \eta_V^{(2)}, \lambda = \hat{\lambda}$ for $\mathcal{T}_2 - \mathcal{T}_1$ steps;

   **Thresholding-normalization:** $\forall k \in [n], \hat{\boldsymbol{a}}^{(k)} = [a_t^{(k)} \mathbb{1}\{a_t^{(k)} \geq \Omega(1/Q)\}]_t, \boldsymbol{a}^{(k)} \leftarrow \hat{\boldsymbol{a}}^{(k)} / \boldsymbol{1}^\top \hat{\boldsymbol{a}}^{(k)}$

   **Stage 3:** Run (5) and (6) with $\eta_A = 0, \eta_V = \eta_V^{(3)}, \lambda = 0$ for $\mathcal{T}_3 - \mathcal{T}_2$ steps;

**Output:** $A_{\mathcal{T}_3}, V_{\mathcal{T}_3}$.

---

## 3 Results for training from scratch

In this section, we consider the situations where we train the model from scratch, i.e., the initialization is $V_0 = \mu \boldsymbol{1}^\top$ and $A_0 = \boldsymbol{1}\boldsymbol{1}^\top / T$ and discuss the ideas of the proof of the following theorem.

**Theorem 3.1** (Theorem G.1). *Let $\varepsilon > 0$ be our target accuracy and $\mathcal{T}_1 = \min\{\tau \geq 0 : \max\{\alpha_{V,\tau}, \alpha_{A,\tau}\} \geq \Theta(1/(QN))\}$. We can choose the hyperparameters in Algorithm 1 such that within $\text{poly}(N, Q, 1/\varepsilon, \log T)$ steps, we have $\|A - Q\|_\mu^2 \leq \varepsilon$ and $\|V - P\|_\mu^2 \leq \varepsilon$ with probability at least $1 - \delta$ and the numbers of samples used before and after $\mathcal{T}_1$ are $\text{poly}(T, \delta)$ and $\text{poly}(N, Q, 1/\varepsilon, \log T, \delta)$, respectively.*

The overall strategy is analyzing the population process and then controlling the distance between the mini-batch trajectory and the population process[2]. In Section 3.1, we discuss the key properties of the population process that simplify the analysis. After that, we describe the dynamics of the algorithm and the signal-noise-ratio (SNR) in each of the three stages of Algorithm 1 in Section 3.2~3.4.

### 3.1 The population process

In this subsection, we analyze the behavior of the population process and the evolution of the signal-noise ratio. More details can be found in Appendix C, where the so-called population projected process are defined and rigorously analyzed.

For ease of presentation, we assume $\lambda = 0$ and access to the population loss $\mathcal{L} := \mathbb{E}\, l$. In other words, we consider the projected preconditioned gradient descent. By Lemma B.8, the dynamics of the population process is controlled by

$$V_{\tau+1} = V_\tau - \eta_V \left( \|A\|_\mu^2 \left(V - \mu\boldsymbol{1}^\top\right) - \langle Q, A \rangle_\mu \left(P - \mu\boldsymbol{1}^\top\right) \right),$$

$$A_{\tau+1} = A_\tau - \eta_A \left( \left(\|V\|_\mu^2 - \|\mu\|^2\right) \left(\boldsymbol{a}^{(k)} - \frac{\boldsymbol{1}}{T}\right) - \left(\langle V, P \rangle_\mu - \|\mu\|^2\right) \left(\boldsymbol{q}^{(k)} - \frac{\boldsymbol{1}}{T}\right) \right). \tag{7}$$

One can prove via induction on $\tau$ that $V$ (resp. $A$) always stays on the straight line crossing $\mu\boldsymbol{1}^\top$ and $P$ (resp. $\boldsymbol{1}\boldsymbol{1}^\top / T$ and $Q$). In other words, there exists some time-dependent real numbers $\alpha_{V,\tau}, \alpha_{A,\tau}$, $\beta_{V,\tau} := 1 - \alpha_{V,\tau}, \beta_{A,\tau} := 1 - \alpha_{A,\tau}$ such that $V_\tau = \alpha_{V,\tau} P + \beta_{V,\tau} \mu\boldsymbol{1}^\top$ and $A_\tau = \alpha_{A,\tau} Q + \beta_{A,\tau} \boldsymbol{1}\boldsymbol{1}^\top / T$. The same calculation yields the following equations that govern the dynamics of $\alpha_V$ and $\alpha_A$:

$$\alpha_{V,\tau+1} = \alpha_{V,\tau} + \eta_V K_Q \left(1 - \alpha_A \alpha_V\right) \alpha_A + \eta_V \frac{1 - \alpha_V}{T},$$

$$\alpha_{A,\tau+1} = \alpha_{A,\tau} + \eta_A K_P \left(1 - \alpha_V \alpha_A\right) \alpha_V,$$

where $\alpha_{V,0} = \alpha_{A,0} = 0$, $K_P := \|P\|_\mu^2 - \|\mu\|^2 \gtrsim 1/N$ and $K_Q := \|Q\|_\mu^2 - 1/T \gtrsim 1/Q$. Choose $\eta_V = \eta / K_Q$ and $\eta_A = \eta / K_P$, and we can write the above in matrix form as

$$\begin{bmatrix} \alpha_{V,\tau+1} \\ \alpha_{A,\tau+1} \end{bmatrix} = \eta(1 - \alpha_A \alpha_V) \begin{bmatrix} 0 & 1 \\ 1 & 0 \end{bmatrix} \begin{bmatrix} \alpha_{V,\tau} \\ \alpha_{A,\tau} \end{bmatrix} + \frac{\eta}{K_Q} \begin{bmatrix} (1 - \alpha_V)/T \\ 0 \end{bmatrix}. \tag{8}$$

---

[2] Strictly speaking, what we actually control is the distance of the mini-batch trajectory to the subspace the population process lies. This allows us to prevent the potential exponential growth of the error caused by error compounding in the analysis. For details on this technique, see Appendix C.

Hence, in order to analyze the population process, it suffices to analyze the above 2-dimensional ODE. In what follows, when we say the signal, we usually refer to these $\alpha$'s or some quantities whose size is proportional to them. In particular, as one can see from (8), the size of the expected gradients is proportional to $\alpha_V$ and/or $\alpha_A$[3].

Note that when both $\alpha_V, \alpha_A$ are still small, the population dynamics of $\alpha_V, \alpha_A$ are a linear system with coefficient matrix $\eta \begin{bmatrix} 0 & 1 \\ 1 & 0 \end{bmatrix}$ and drift $\begin{bmatrix} \eta T/K_Q \\ 0 \end{bmatrix}$. The drift term will provide a small initial signal that guides the process toward the correct direction and then the linear term will amplify this signal. Since the linear term is close to 0 at initial and the initial signal provided by the drift term has order $1/T$, we should expect that poly $T$ samples are necessary to distinguish it from noises (Stage 1). After the signal becomes reasonably large, the first term will have order $1/\mathrm{poly}(N, Q)$, and we can then rely on it (combined with the $l_1$-regularization) instead of the drift term to learn the model (Stage 2).

### 3.2 Stage 1: boosting the signal

At initialization, we have $\alpha_V = \alpha_A = 0$. We define Stage 1 to be the phase until at least one of them has grown from 0 to some small $\sigma_1 = 1/\mathrm{poly}(N, Q)$. Note that in this stage, the mini-batch version of (8) is approximately equivalent to

$$\begin{bmatrix} \alpha_{V,\tau+1} \\ \alpha_{A,\tau+1} \end{bmatrix} \approx \eta \begin{bmatrix} 0 & 1 \\ 1 & 0 \end{bmatrix} \begin{bmatrix} \alpha_{V,\tau} \\ \alpha_{A,\tau} \end{bmatrix} + \frac{\eta}{K_Q} \begin{bmatrix} 1/T \\ 0 \end{bmatrix} + \boldsymbol{\varepsilon}_{\mathrm{noise}} + \boldsymbol{\varepsilon}_{\mathrm{approx}}, \tag{9}$$

where $\boldsymbol{\varepsilon}_{\mathrm{noise}}$ and $\boldsymbol{\varepsilon}_{\mathrm{approx}}$ represent the errors introduced by the difference between the mini-batch and population gradients, and the fact that we are not exactly on the population trajectory. If we had infinite amount of samples so that both $\boldsymbol{\varepsilon}_{\mathrm{noise}}$ and $\boldsymbol{\varepsilon}_{\mathrm{approx}}$ were 0, then the second term on the RHS of (9) could provide a small positive signal to $\alpha$ and the first term would quickly boost it to $\sigma_1$ within $\mathcal{T}_1 = \log(T)/\eta$ iterations. In order for the above analysis to work, we need both $\boldsymbol{\varepsilon}_{\mathrm{noise}}$ and $\boldsymbol{\varepsilon}_{\mathrm{approx}}$ to be at least $O(1)/T$ small. Since, unfortunately, $\boldsymbol{\varepsilon}_{\mathrm{noise}}$ does not scale with $1/T$, we need poly$(T)$ samples to ensure these conditions.

We conjecture that this poly$(T)$ dependence is unavoidable (when only a polynomial amount of computing time is available). That is because around the initialization, the only signal comes from the second term and the first term amplifies whatever the second term provides, even if it has been corrupted by the errors. It either takes poly$(T)$ fresh samples each step to reveal the signal or poly$(T)$ steps (whence also poly$(T)$ samples) for the random noises to (hopefully) cancel with each other.

### 3.3 Stage 2: learning the model

We know that at the end of Stage 1, at least one of $\alpha_V$ and $\alpha_A$ is $\sigma_1 = 1/\mathrm{poly}(Q, N)$ large. Hence, one may expect that the signal is $1/\mathrm{poly}(Q, N)$ large now so that we no longer need to make the noises $1/T$ small and therefore, only poly$(N, Q)$ samples are needed. Unfortunately, this argument will not work directly, since the variance of the mini-batch gradients scales with $T$.[4] Therefore, we still need poly$(T)$ samples to reduce the squared $\ell_2$-norm of $\boldsymbol{\varepsilon}_{\mathrm{noise}}$ from $\Omega(T)$ to $1/\mathrm{poly}(Q, N)$. To address this issue, we introduce the $\ell_1$-regularizer and use a variant of proximal gradient descent.

The idea is, while the concentration in the $\ell_2$ sense is difficult, controlling the $\ell_\infty$-error is easy as every entry of $\nabla_{\boldsymbol{a}^{(k)}} l$ is bounded whence subgaussian. As a result, we can make the coordinate-wise difference between the population and mini-batch gradients $1/\mathrm{poly}(N, Q)$ small using only poly$(N, Q) \log T$ samples by a standard concentration argument. Moreover, we have (cf. the proof of Lemma E.3)

$$-\mathbb{E}\, \partial_{a_t^{(k)}} l = \mu_k \alpha_V K_P \left( q_t^{(k)} - \alpha_V a_t^{(k)} \right) = \mu_k \alpha_V K_P \times \begin{cases} \Omega(1/Q), & \text{if } q_t^{(k)} \neq 0, \\ O(\alpha_V a_t^{(k)}), & \text{if } q_t^{(k)} = 0. \end{cases} \tag{10}$$

Thus, as long as $\alpha_V \geq 1/\mathrm{poly}(N, Q)$, the $\ell_\infty$-norm of the gradient noise being small is enough to create a separation between those useful entries ($q_t^{(k)} \neq 0$) and useless entries ($q_t^{(k)} = 0$) and ensure the $\ell_2$-error of those $Q$ useful entries is small.

---

[3]We will often drop the time subscript $\tau$ and write $\alpha_V := \alpha_{V,\tau}$ for simplicity.

[4]It is possible almost explicitly to compute the covariance matrix through some tedious calculation. Intuitively, the reason it scale with $T$ is $\nabla_{\boldsymbol{a}^{(k)}} l$ has $T$ entries with most of them almost uncorrelated in a certain sense.

The above analysis suggests removing all small entries from the gradient will work. Now, we claim that $\ell_1$-regularization and proximal gradient descent naturally implement this strategy, at least approximately. We believe softmax-based attention layers also automatically implement this strategy. See Section 5 for more discussion on the relationship between our model and softmax transformers.

Note that, at the end of Stage 1 and after the thresholding-projection step — which is approximately equivalent to running one proximal step first — we know that all useful $a_t^{(k)}$ are at least $\Omega(\alpha_V/Q) = 1/\text{poly}(N,Q)$, while all useless entries are of size $O(1)/T$. By our previous discussion, we know that if $\lambda$ is chosen appropriately, with $\text{poly}(N,Q,\log T)$ samples, the gradients w.r.t. those useful entries can be made approximately correct, while the gradients w.r.t. those useless entries are much smaller than $\lambda$. Thus, after one gradient step, the absolute value of each of those useless $a_t^{(k)}$ is still much smaller than $\lambda$. As a result, they will be set to 0 in the proximal step (and to $O(1/T)$ after the projection step), which is equivalent to filtering out all those entries, up to a small bias. Therefore, the proximal gradient updates stay close to the population trajectory, and the growth of the signals $\alpha_A, \alpha_V$ can be analyzed using the population dynamics.

We end Stage 2 when $(\alpha_V + \alpha_A)/2 \approx 1$. Similar to Stage 1, this also only takes $\mathcal{T}_2 = \tilde{O}(1/\eta)$ steps. We also show that the difference between the mini-batch trajectory and the "population trajectory" can decrease to a small value (cf. Lemma E.10). This allows us to decouple the error introduced by Stage 1 and the target accuracy. We defer the proof details to Appendix E.

### 3.4 Stage 3: final rounding and convergence

The purpose of Stage 3 is to fix a minor issue regarding $|\alpha_V - \alpha_A|$. Taylor expand (8) around $(\alpha_A, \alpha_V) = (1,1)$ and one will notice that although $(\alpha_V + \alpha_A)/2$ can converge to 1 at a linear rate (and the approximation error also decreases exponentially fast), the convergence rate of $\alpha_A - \alpha_V$ is much slower, and the process will get stuck around $(1 + \delta, 1 - \delta)$ for some small nonzero $\delta$, instead of converging to $(1,1)$. To accelerate this process, we directly round $A$ via normalization, which is possible only after the approximation error becomes small in Stage 2. Then we freeze $A$ and train $V$ to the desired accuracy. More details about this stage can be found in Appendix F.

## 4 Results for transfer learning

The transferability of neural networks and transformers and their benefits have been widely observed and studied in both practice and theory. It is often assumed that the downstream and pretraining tasks share a common structure or representations/features, and these models can learn these common structures during training, and then leverage them in fine-tuning.

In this section, we offer a different perspective: as long as there is a (potentially small) nontrivial correlation between the pretraining and downstream tasks, the pretrained model can be used to provide a nonzero initial signal, allowing us to bypass the initial sample-intensive signal-boosting stage.

Formally, we consider the following setting. Let $\hat{P}$ be the transition matrix of the pretraining task and $(P, Q)$ the transition matrix and $Q$-matrix of the downstream task. We still assume Assumption 2.1. In addition, we assume $\hat{P}$ and $P$ share the same stationary distribution $\mu$, $\|\hat{P}\|_\mu^2 = \Theta(1)\|P\|_\mu^2$ and $\langle \hat{P}, P \rangle_\mu - \|\mu\|^2 \geq \|\mu\|^2$. The last condition can be viewed as the transfer learning version of condition (c) of Assumption 2.1. Note that we allow the correlation between $\hat{P}$ and $P$ to be as small as $o(1)$.

**Theorem 4.1** (informal version of Theorem H.3)**.** *Consider the above setting. Initialize $A = \mathbf{1}\mathbf{1}^\top/T$, $V = \theta\hat{P} + (1 - \theta)\mu\mathbf{1}^\top$ for some small $\theta > 0$, and run one step of gradient update on $A$. Then, running Algorithm 1 from Stage 2 allows us to recover $(P, Q)$ to $\varepsilon$-accuracy with high probability with $\text{poly}(N, Q, 1/\varepsilon, \log T)$ samples.*

To intuitively see why using $\text{poly}(N, Q, 1/\varepsilon, \log T)$ samples is possible, recall from (9) that the reason we need $\text{poly}(T)$ samples in Stage 1 is the signal is additive and has order $O(1/T)$, so we need the size of the noise to be at most $O(1/T)$. On the other hand, when we initialize $V = \theta\hat{P} + (1 - \theta)\mu\mathbf{1}^\top$, we have $\alpha_V \geq \Theta(\theta/(NK_P)) \gg 1/T$. Then we can rely on $\alpha_V$, instead of the $1/T$-sized additive signal, to boost the signal of $\alpha_A$ to $\omega(1/T)$ in one step, which leads to a sample complexity that depends on $\alpha_V$ instead of the $1/T$-sized additive signal. Then, we can reuse the analysis of Stage 2 and 3 to show that the downstream $(P, Q)$ can be approximately recovered using $\text{poly}(N, Q, 1/\varepsilon, \log T)$.

Note that unlike the case of training from scratch, when performing transfer learning, the initial approximation error $\|\Delta_V\|_\mu$, i.e., the distance between $V$ and its population projection, can be much larger than the signal $\alpha_V$, and it might seem unreasonable to expect that we can leverage the small signal in the presence of a large approximation error. To handle this issue, we show that the *influence* of $\|\Delta_V\|_\mu^2$ on the dynamics scales with $\alpha_A(\approx \alpha_V)$, which is small. In addition, we also show that as long as $\alpha_A$ is bounded away from 0 and the batch size is large, the approximation error will not grow. This allows us to ignore the approximation errors in the signal-boosting stage until we enter the regime of the Stage 2 analysis.

## 5 Experiments and relationship with softmax transformers

This section contains our experimental results. We also discuss the relationship between our linear model and the usual softmax transformers.

**Experiment setup** We use the same shallow transformer model (2) to train on the synthetic data. The data distribution follows the SCB model (1) with a randomly sampled transition matrix $P$ together with its stationary $\mu$, and the ground truth attention pattern $Q$. We choose the number of states $N = 3$, sparsity $Q = 2$, and the sequence length $T = 5000 \gg N, Q$. We use a batch size $B = 64$ to run the online projected proximal gradient descent with $\lambda = $ 1e-5 and the vanilla SGD for $\mathcal{T} = 1000$ iterations. Through the signal boosting stage $\tau \in [0, 400]$, we use $\eta_1 = 0.01$ to accelerate the process. After $\tau > 400$, we use $\eta_2 = 0.005$ for further improvement. For SGD, we add another set of experiments with $\eta_2' = 0.001$ to prevent potential instability. For more details, see Appendix I.

### 5.1 Convergence

Our experiments (cf. Fig. 1) show that after switching to proximal gradient descent after Stage 1 (the signal-boosting stage), both $\|P - V\|_\mu$ and $\|A - Q\|_\mu$ decrease faster than SGD. The final distance to the ground-truth after normalization gets close to 0, and the similarity between the ground truth and parameters quickly converges close to 1. In comparison, SGD struggles to converge with the same small batch size and large learning rate, while the convergence rate is too slow when a smaller learning rate is used. This phenomenon verifies our theory that the variance of the original stochastic gradient will be too large for SGD to converge when $T \gg Q, N$, while proximal gradient descent with an $\ell_1$ regularizer can resolve this issue.

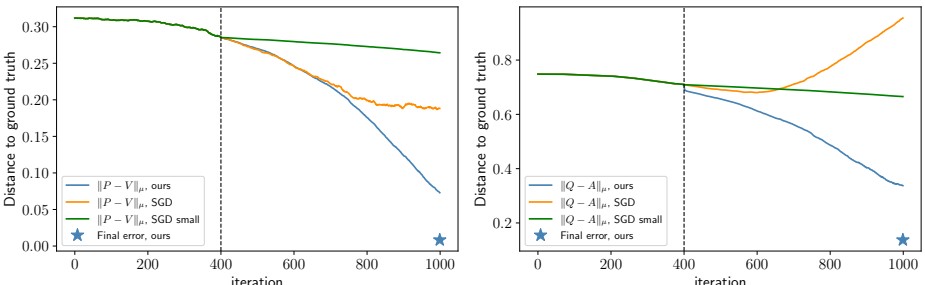

Figure 1: **Convergence analysis:** We plot the distance to the ground truth $\|V - P\|_\mu$, $\|A - Q\|_\mu$ in different settings. After stage 1 ends at $\tau = 400$ (when $\alpha_A, \alpha_V \approx 0.1$), we use vanilla SGD and our proximal gradient method to train the transformer. Compared with SGD, the $\ell_1$ regularized proximal gradient descent quickly converges, and the final solution (the star) recovers the ground truth. SGD either suffers from the large gradient variance (when $\eta_2$ is large) or a slow convergence rate (small $\eta_2'$).

### 5.2 Relationship between our model and softmax transformers

We claim that they have our linear model and softmax transformers have qualitatively similar behaviors: there will be a sample-intensive initial stage, and after the model and the target have a nontrivial correlation, proximal gradient descent/SGD will become much more sample efficient.

For ease of presentation, in the following, we will assume $N = 1$, write $a := a^{(1)}$, and assume the ground-truth $q := q^{(1)}$ is $e_1 = (1, 0, \ldots, 0)$. Most of our argument below can be generalized to the general setting at least at a heuristic level. Recall that our linear model is $f(X; V, a) = VXa$. By a softmax transformer, we mean the model $f_\sigma(X; V, w) = VX\sigma(w) =: VXa_\sigma$ where $\sigma$ is the softmax function and $w \in \mathbb{R}^T$ is the trainable first-layer weights.

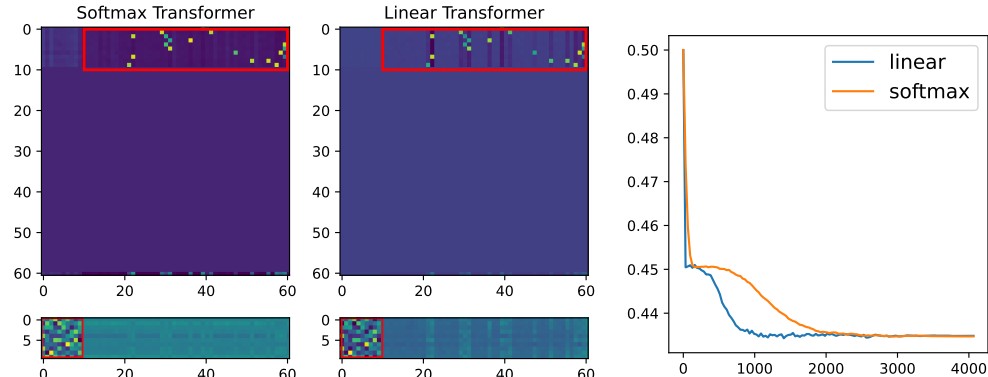

Figure 2: **Similarity between the softmax and linear attention.** We train two transformers with (1) (Left) *softmax* attention and (2) (Middle) *linear* attention layer on the SCB tasks with the same ground-truth ($T = 50, N = 10, Q = 2$). The attention pattern and the value matrix (learned transition matrix) are very similar (left two plots) and they converge to approximately the same loss (right plot).

Let $l$ denote the (per-sample) loss. We have $\nabla_w l(f_\sigma(X)) = \left(\text{diag}(\boldsymbol{a}_\sigma) - \boldsymbol{a}_\sigma \boldsymbol{a}_\sigma^\top\right)(VX)^\top \nabla l(f_\sigma(X))$. As a result, the dynamics of the attention weights are controlled by

$$\boldsymbol{a}(\tau + 1) \approx \boldsymbol{a}(\tau) - \eta \left(\boldsymbol{I} - \frac{\boldsymbol{1}\boldsymbol{1}^\top}{T}\right)(VX)^\top \nabla l(f(X)), \qquad \text{in our linear model,}$$

$$\boldsymbol{a}_\sigma(\tau + 1) \approx \boldsymbol{a}_\sigma(\tau) - \eta \left(\text{diag}(\boldsymbol{a}_\sigma) - \boldsymbol{a}_\sigma \boldsymbol{a}_\sigma^\top\right)^2 (VX)^\top \nabla l(f_\sigma(X)), \quad \text{in softmax transformers.}$$

In other words, the main difference is that there will be a preconditioning matrix $(\text{diag}(\boldsymbol{a}) - \boldsymbol{a}\boldsymbol{a}^\top)^2$ in the dynamics of softmax transformers.

Near initialization, i.e., when the attention pattern is still close to the uniform attention, we have $\left(\text{diag}(\boldsymbol{a}_\sigma) - \boldsymbol{a}_\sigma \boldsymbol{a}_\sigma^\top\right)^2 \approx \frac{1}{T^2}\left(\boldsymbol{I} - \frac{\boldsymbol{1}\boldsymbol{1}^\top}{T}\right)$. In other words, our linear model and softmax transformers are approximately equivalent up to a change in learning rates.

Now, suppose that there is a nontrivial correlation between $\boldsymbol{a}_\sigma$ and $\boldsymbol{q} = \boldsymbol{e}_1$, say, $a_{\sigma,1}$ is a small constant while all other entries are $O(1/T)$. In this case, we have $\left(\text{diag}(\boldsymbol{a}_\sigma) - \boldsymbol{a}_\sigma \boldsymbol{a}_\sigma^\top\right)^2 \approx a_{\sigma,1}(1 - a_{\sigma,1})\boldsymbol{e}_1\boldsymbol{e}_1^\top + O(1/T)$. Effectively, softmax transformers automatically adjust the learning rate according to $a_{\sigma,t}$ and roughly ignore those positions with a small attention weight to stabilize the gradients. Note that this is also what $\ell_1$-regularization does in our algorithm. In fact, mimicking this behavior is one of the motivations of using $\ell_1$-regularization in our linear setting. We run further experiments to highlight the resemblance between softmax attention and our linear attention model (Figure 2).

## 6 Conclusion and discussion

In this paper, we propose the Sparse Contextual Bigram (SCB) model, which is a natural extension of the bigram model, that requires both contextual and global information. Then, we analyze the problem of learning a SCB model using a one-layer linear transformer and a gradient-based algorithm. We prove quantitative bounds on the convergence rate and the sample complexity. In particular, we show when trained from scratch, the training process can be split into two stages, where the first stage uses a lot of samples to boost the signal from zero to a nontrivial value, while the second stage is much more sample-efficient. Then, we consider the problem in a transfer learning setting and prove that when there is a nontrivial correlation between the pretraining and downstream tasks, the first sample intensive stage can be bypassed.

Our data-generating model and results also lead to some interesting future directions. For example, can we improve the sample complexity of the first stage? What can we gain if the datapoints are sequences generated by repeatedly applying the SCB model?

## Acknowledgement

JDL acknowledges support of the NSF CCF 2002272, NSF IIS 2107304, and NSF CAREER Award 2144994.

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

# A  Limitation

In this section, we briefly discuss the limitation of this work.

First, we consider one-layer single-head linear transformers with certain (blocks of the) weights merged or fixed. Though this simplification are widely used in theoretical works and linear and nonlinear transformers share some training behaviors (Ahn et al., 2023), this architecture is still very far away from the transformers used in practice.

We also use a non-standard training algorithm that has several manually separated stages. Some parts of the modification are made to address certain issues of linear transformers, while the other are made to simplify the analysis. It would be interesting (and more challenging) to consider more natural/practical training algorithms.

Finally, for our data-generating model, we only use it to generate one next token, instead of repeatedly apply SCB on the previous generated results to obtain a long sequence. In our setting, the contextual tokens are independent. While this simplifies the analysis, it deviates from how natural language works.

# B  Probabilities, expectations, and variances

We collect in this section closed-form formulas for the probabilities of certain events, and the expectations and variances of some random vectors of interest. All proofs are deferred to the end of this section.

## B.1  Probabilities

**Lemma B.1.** *For any $t \in [T]$ and $k, n, m \in [N]$, we have*

$$\mathbb{P}(x_o = n, x_t = m \mid x_{T+1} = k) = q_t^{(k)} P_{n,m} \mu_m + \left(1 - q_t^{(k)}\right) \mu_n \mu_m.$$

**Lemma B.2.** *For any $s \neq t \in [T]$, $k, n, m, l \in [N]$, we have*

$$\mathbb{P}(x_o = n \mid x_{T+1} = k, x_s = m, x_t = l) = q_s^{(k)} P_{n,m} + q_t^{(k)} P_{n,l} + \left(1 - q_s^{(k)} - q_t^{(k)}\right) \mu_n.$$

## B.2  Gradient and Expectations

**Lemma B.3.** *Suppose the last input token $\boldsymbol{x}_{T+1}$ and $\boldsymbol{a} := \boldsymbol{A} \boldsymbol{x}_{T+1}$. The gradients of the objective are*

$$\nabla_{\boldsymbol{V}} l = \left(\boldsymbol{V} \boldsymbol{X} \boldsymbol{a} - \boldsymbol{e}_{x_o}\right) \left(\boldsymbol{X} \boldsymbol{a}\right)^\top,$$
$$\nabla_{\boldsymbol{a}^{(k)}} l = \mathbb{1}\{x_{T+1} = k\} (\boldsymbol{V} \boldsymbol{X})^\top \left(\boldsymbol{V} \boldsymbol{X} \boldsymbol{a} - \boldsymbol{e}_{x_o}\right)$$

**Lemma B.4.** *For any $k \in [N]$ and $s, t \in [T]$, we have*

$$\mathbb{E}^{(k)}[\boldsymbol{e}_{x_s} \boldsymbol{e}_{x_t}^\top] = \mathbb{E}[\boldsymbol{e}_{x_s} \boldsymbol{e}_{x_t}^\top] = \begin{cases} \mathrm{diag}(\boldsymbol{\mu}), & s = t, \\ \boldsymbol{\mu} \boldsymbol{\mu}^\top, & s \neq t. \end{cases}$$

**Lemma B.5.** *For any $\boldsymbol{V} \in \mathbb{R}^{N \times N}$ with $\boldsymbol{V} \boldsymbol{\mu} = \boldsymbol{\mu}$ and $s, t \in [T]$, we have*

$$\sum_{n=1}^N \mathbb{E}[V_{n,x_s} V_{n,x_t}] = \begin{cases} \|\boldsymbol{V}\|_\mu^2, & s = t, \\ \|\boldsymbol{\mu}\|^2, & s \neq t. \end{cases}$$

**Lemma B.6.** *For any $\boldsymbol{V} \in \mathbb{R}^{N \times N}$ with $\boldsymbol{V} \boldsymbol{\mu} = \boldsymbol{\mu}$ and $t \in [T]$, we have*

$$\mathbb{E}^{(k)} V_{x_o, x_t} = q_t^{(k)} \langle \boldsymbol{V}, \boldsymbol{P} \rangle_\mu + \left(1 - q_t^{(k)}\right) \|\boldsymbol{\mu}\|^2.$$

**Lemma B.7** (Expected gradients)**.**

$$\mathbb{E} \nabla_{\boldsymbol{V}} l = \|\boldsymbol{A}\|_\mu^2 \boldsymbol{V} \mathrm{diag}(\boldsymbol{\mu}) + \left(1 - \|\boldsymbol{A}\|_\mu^2\right) \boldsymbol{\mu} \boldsymbol{\mu}^\top$$

$$- \langle Q, A \rangle_\mu \, P \mathrm{diag}(\mu) - \left(1 - \langle Q, A \rangle_\mu\right) \mu \mu^\top,$$

$$\mathbb{E} \, \nabla_{a^{(k)}} l = \mu_k \left(\|V\|_\mu^2 - \|\mu\|^2\right) a^{(k)} + \mu_k \mathbf{1} \|\mu\|^2$$
$$- \mu_k q^{(k)} \langle V, P \rangle_\mu - \mu_k \left(1 - q^{(k)}\right) \|\mu\|^2 \, .$$

**Lemma B.8** (Expected preconditioned gradients).

$$\mathbb{E} \, \hat{\nabla}_V l = \|A\|_\mu^2 \left(V - \mu \mathbf{1}^\top\right) - \langle Q, A \rangle_\mu \left(P - \mu \mathbf{1}^\top\right),$$
$$\mathbb{E} \, \hat{\nabla}_{a^{(k)}} l = \left(\|V\|_\mu^2 - \|\mu\|^2\right) \left(a^{(k)} - \frac{1}{T}\right) - \left(\langle V, P \rangle_\mu - \|\mu\|^2\right) \left(q^{(k)} - \frac{1}{T}\right).$$

## B.3  Deferred proofs of this section

### B.3.1  Probabilities

*Proof of Lemma B.1.* For notational simplicity, define $\boldsymbol{x}_{-t} = (x_1, \ldots, x_{t-1}, x_{t+1}, \ldots, x_T)$. We compute

$$\mathbb{P}(x_o = n, x_t = m \mid x_{T+1} = k)$$
$$= \sum_{\hat{\boldsymbol{m}} \in [N]^{T-1}} \mathbb{P}(x_o = n, x_t = m, \boldsymbol{x}_{-t} = \hat{\boldsymbol{m}} \mid x_{T+1} = k)$$
$$= \sum_{\hat{\boldsymbol{m}} \in [N]^{T-1}} \mathbb{P}(x_o = n \mid x_t = m, \boldsymbol{x}_{-t} = \hat{\boldsymbol{m}}, x_{T+1} = k)$$
$$\times \mathbb{P}(x_t = m, \boldsymbol{x}_{-t} = \hat{\boldsymbol{m}} \mid x_{T+1} = k).$$

By the independence assumption, we have $\mathbb{P}(x_t = m, \boldsymbol{x}_{-t} = \hat{\boldsymbol{m}} \mid x_{T+1} = k) = \mathbb{P}(x_t = m, \boldsymbol{x}_{-t} = \hat{\boldsymbol{m}}) = \mathbb{P}(x_t = m) \mathbb{P}(\boldsymbol{x}_{-t} = \hat{\boldsymbol{m}})$. For the first factor, we have $\mathbb{P}(x_o = n \mid x_t = m, \boldsymbol{x}_{-t} = \hat{\boldsymbol{m}}, x_{T+1} = k) = Q_t^{(k)} P_{n,m} + \sum_{s \neq t} Q_s^{(k)} P_{n,\hat{m}_s}$. Therefore,

$$\mathbb{P}(x_o = n, x_t = m \mid x_{T+1} = k)$$
$$= \sum_{\hat{\boldsymbol{m}} \in [T]^{N-1}} \left(Q_t^{(k)} P_{n,m} + \sum_{s \neq t} Q_s^{(k)} P_{n,\hat{m}_s}\right) \mathbb{P}(x_t = m) \, \mathbb{P}(\boldsymbol{x}_{-t} = \hat{\boldsymbol{m}})$$
$$= \left(Q_t^{(k)} P_{n,m} + \sum_{s \neq t} Q_s^{(k)} \sum_{\hat{\boldsymbol{m}} \in [T]^{N-1}} P_{n,\hat{m}_s} \, \mathbb{P}(\boldsymbol{x}_{-t} = \hat{\boldsymbol{m}})\right) \mu_m.$$

Note that for any $s \neq t$

$$\sum_{\hat{\boldsymbol{m}} \in [T]^{N-1}} P_{n,\hat{m}_s} \, \mathbb{P}(\boldsymbol{x}_{-t} = \hat{\boldsymbol{m}}) = \sum_{\hat{\boldsymbol{m}}_{-s} \in [T]^{N-2}} \left(\sum_{\hat{m}_s=1}^{N} P_{n,\hat{m}_s} \mu_{\hat{m}_s}\right) \mathbb{P}(\boldsymbol{x}_{-s,-t} = \hat{\boldsymbol{m}}_{-s}) = \mu_n.$$

Thus,

$$\mathbb{P}(x_o = n, x_t = m \mid x_{T+1} = k) = \left(q_t^{(k)} P_{n,m} + \sum_{s \neq t} q_s^{(k)} \mu_n\right) \mu_m$$
$$= q_t^{(k)} P_{n,m} \mu_m + \left(1 - q_t^{(k)}\right) \mu_n \mu_m.$$

$\square$

*Proof of Lemma B.2.* For notational simplicity, let $\boldsymbol{x}_{-s,-t}$ denote the vector obtained by removing the $s, t$ coordinates from $\boldsymbol{x}$. Then, we compute

$$\mathbb{P}(x_o = n \mid x_{T+1} = k, x_s = m, x_t = l)$$
$$= \sum_{\hat{\boldsymbol{m}} \in [N]^{T-2}} \mathbb{P}(x_o = n, \boldsymbol{x}_{-s,-t} = \hat{\boldsymbol{m}} \mid x_{T+1} = k, x_s = m, x_t = l)$$

$$= \sum_{\hat{\boldsymbol{m}} \in [N]^{T-2}} \mathbb{P}(x_o = n \mid x_{T+1} = k, x_s = m, x_t = l, \boldsymbol{x}_{-s,-t} = \hat{\boldsymbol{m}}) \, \mathbb{P}(\boldsymbol{x}_{-s,-t} = \hat{\boldsymbol{m}})$$

$$= \sum_{\hat{\boldsymbol{m}} \in [N]^{T-2}} \left( q_s^{(k)} P_{n,m} + q_t^{(k)} P_{n,l} + \sum_{i \notin \{s,t\}} q_i^{(k)} P_{n,\hat{m}_i} \right) \mathbb{P}(\boldsymbol{x}_{-s,-t} = \hat{\boldsymbol{m}})$$

$$= q_s^{(k)} P_{n,m} + q_t^{(k)} P_{n,l} + \left( 1 - q_s^{(k)} - q_t^{(k)} \right) \mu_n.$$

$\square$

### B.3.2 Expectations

*Proof of Lemma B.3.* For each sample $\boldsymbol{X}$, we have

$$l(\boldsymbol{x}, x_{T+1}, x_o; \boldsymbol{V}, \boldsymbol{A}) := \frac{1}{2} \left\| \boldsymbol{e}_{x_o} - \boldsymbol{V}\boldsymbol{X}\boldsymbol{A}\boldsymbol{e}_{x_{T+1}} \right\|^2.$$

and $\boldsymbol{a}$ Then we have the matrix differential:

$$\mathrm{d}l = \left( \boldsymbol{V}\boldsymbol{X}\boldsymbol{a} - \boldsymbol{e}_{x_o} \right)^\top \mathrm{d}\boldsymbol{V}\boldsymbol{X}\boldsymbol{a} + \left( \boldsymbol{V}\boldsymbol{X}\boldsymbol{a} - \boldsymbol{e}_{x_o} \right)^\top \boldsymbol{V}\boldsymbol{X}\mathrm{d}\boldsymbol{a}$$

Therefore,

$$\nabla_{\boldsymbol{V}} l = \left( \boldsymbol{V}\boldsymbol{X}\boldsymbol{a} - \boldsymbol{e}_{x_o} \right) \left( \boldsymbol{X}\boldsymbol{a} \right)^\top,$$
$$\nabla_{\boldsymbol{a}^{(k)}} l = \mathbb{1}\{x_{T+1} = k\}(\boldsymbol{V}\boldsymbol{X})^\top \left( \boldsymbol{V}\boldsymbol{X}\boldsymbol{a} - \boldsymbol{e}_{x_o} \right)$$

$\square$

*Proof of Lemma B.4.* When $s \neq t$, we have $\mathbb{E}^{(k)}[\boldsymbol{e}_{x_s} \boldsymbol{e}_{x_t}^\top] = \mathbb{E}^{(k)}[\boldsymbol{e}_{x_s}]\mathbb{E}^{(k)}[\boldsymbol{e}_{x_t}^\top] = \boldsymbol{\mu}\boldsymbol{\mu}^\top$. When $s = t$, we have $\mathbb{E}^{(k)}[\boldsymbol{e}_{x_t} \boldsymbol{e}_{x_t}^\top] = \sum_{k=1}^N \mu_k \boldsymbol{e}_k \boldsymbol{e}_k^\top = \mathrm{diag}(\boldsymbol{\mu})$. $\square$

*Proof of Lemma B.5.* When $s \neq t$, we have

$$\sum_{n=1}^N \mathbb{E}[V_{n,x_s} V_{n,x_t}] = \sum_{n=1}^N \left( \mathbb{E} V_{n,x_s} \right)^2 = \sum_{n=1}^N \left( \sum_{k=1}^N \mu_k V_{n,k} \right)^2$$

$$= \sum_{k,l=1}^N \mu_k \mu_l \sum_{n=1}^N V_{n,k} V_{n,l} = \boldsymbol{\mu}^\top \boldsymbol{V}^\top \boldsymbol{V}\boldsymbol{\mu} = \|\boldsymbol{\mu}\|^2.$$

When $s = t$, we have $\sum_{n=1}^N \mathbb{E} V_{n,x_t}^2 = \sum_{n=1}^N \sum_{k=1}^N \mu_k V_{n,k}^2 = \|\boldsymbol{V}\|_{\boldsymbol{\mu}}^2$. $\square$

*Proof of Lemma B.6.* Recall Lemma B.1. Then, we compute

$$\mathbb{E}^{(k)} V_{x_o, x_t} = \sum_{n,m=1}^N V_{n,m} \, \mathbb{P}(x_o = n, x_t = m \mid x_{T+1} = k)$$

$$= q_t^{(k)} \sum_{n,m=1}^N V_{n,m} P_{n,m} \mu_m + \left( 1 - q_t^{(k)} \right) \sum_{n,m=1}^N V_{n,m} \mu_n \mu_m$$

$$= q_t^{(k)} \langle \boldsymbol{V}, \boldsymbol{P} \rangle_{\boldsymbol{\mu}} + \left( 1 - q_t^{(k)} \right) \boldsymbol{\mu}^\top \boldsymbol{V}\boldsymbol{\mu}.$$

$\square$

*Proof of Lemma B.7.* First, we consider $\nabla_{\boldsymbol{V}} l = \left( \boldsymbol{V}\boldsymbol{X}\boldsymbol{a} - \boldsymbol{e}_{x_o} \right) \left( \boldsymbol{X}\boldsymbol{a} \right)^\top$, and compute $\mathbb{E}(\boldsymbol{X}\boldsymbol{a})(\boldsymbol{X}\boldsymbol{a})^\top$ and $\mathbb{E}\boldsymbol{e}_{x_o}(\boldsymbol{X}\boldsymbol{a})^\top$. Write $\boldsymbol{X}\boldsymbol{a} = \sum_{t=1}^T a_t \boldsymbol{e}_{x_t}$. Then, we have

$$\mathbb{E}^{(k)} \left[ (\boldsymbol{X}\boldsymbol{a})(\boldsymbol{X}\boldsymbol{a})^\top \right] = \sum_{s,t=1}^T a_s^{(k)} a_t^{(k)} \mathbb{E}[\boldsymbol{e}_{x_s} \boldsymbol{e}_{x_t}^\top]$$

$$= \sum_{t=1}^{T} (a_t^{(k)})^2 \, \mathbb{E}[\boldsymbol{e}_{x_t} \boldsymbol{e}_{x_t}^\top] + \sum_{s \neq t} a_s^{(k)} a_t^{(k)} \, \mathbb{E}[\boldsymbol{e}_{x_s} \boldsymbol{e}_{x_t}^\top]$$

$$= \left\| \boldsymbol{a}^{(k)} \right\|^2 \mathrm{diag}(\boldsymbol{\mu}) + \left(1 - \left\| \boldsymbol{a}^{(k)} \right\|^2 \right) \boldsymbol{\mu}\boldsymbol{\mu}^\top,$$

where the last line comes from Lemma B.4. Then, we compute

$$\mathbb{E}^{(k)}[\boldsymbol{e}_{x_o} (\boldsymbol{X}\boldsymbol{a})^\top] = \sum_{t=1}^{T} a_t^{(k)} \mathbb{E}^{(k)}[\boldsymbol{e}_{x_o} \boldsymbol{e}_{x_t}^\top] = \sum_{t=1}^{T} a_t^{(k)} \, [\mathbb{P}(x_o = n, x_t = m \mid x_{T+1} = k)]_{n,m \in [N]}$$

$$= \sum_{t=1}^{T} a_t^{(k)} \left( q_t^{(k)} \boldsymbol{P} \mathrm{diag}(\boldsymbol{\mu}) + (1 - q_t^{(k)}) \boldsymbol{\mu}\boldsymbol{\mu}^\top \right)$$

$$= \left\langle \boldsymbol{q}^{(k)}, \boldsymbol{a}^{(k)} \right\rangle \boldsymbol{P} \mathrm{diag}(\boldsymbol{\mu}) + \left(1 - \left\langle \boldsymbol{q}^{(k)}, \boldsymbol{a}^{(k)} \right\rangle \right) \boldsymbol{\mu}\boldsymbol{\mu}^\top,$$

where the second line comes from Lemma B.1. Thus, for $\nabla_V l$, we have

$$\mathbb{E}\, \nabla_V l = \boldsymbol{V} \sum_{k=1}^{N} \mu_k \mathbb{E}^{(k)} \left[ (\boldsymbol{X}\boldsymbol{a})(\boldsymbol{X}\boldsymbol{a})^\top \right] - \sum_{k=1}^{N} \mu_k \mathbb{E}^{(k)}[\boldsymbol{e}_{x_o}(\boldsymbol{X}\boldsymbol{a})^\top]$$

$$= \|A\|_\mu^2 \, \boldsymbol{V} \mathrm{diag}(\boldsymbol{\mu}) + \left(1 - \|A\|_\mu^2\right) \boldsymbol{V}\boldsymbol{\mu}\boldsymbol{\mu}^\top - \langle \boldsymbol{Q}, \boldsymbol{A} \rangle_\mu \, \boldsymbol{P} \mathrm{diag}(\boldsymbol{\mu}) - \left(1 - \langle \boldsymbol{Q}, \boldsymbol{A} \rangle_\mu \right) \boldsymbol{\mu}\boldsymbol{\mu}^\top.$$

Now, consider $\nabla_{\boldsymbol{a}^{(k)}} l = \mathbb{1}\{x_{T+1} = k\}(\boldsymbol{V}\boldsymbol{X})^\top \left(\boldsymbol{V}\boldsymbol{X}\boldsymbol{a} - \boldsymbol{e}_{x_o}\right)$ and compute $\mathbb{E}^{(k)}(\boldsymbol{V}\boldsymbol{X})^\top (\boldsymbol{V}\boldsymbol{X})$ and $\mathbb{E}^{(k)}(\boldsymbol{V}\boldsymbol{X})^\top \boldsymbol{e}_{x_i}$. By Lemma B.5, for each $s, t \in [T]$, we have

$$\boldsymbol{e}_s^\top \mathbb{E}^{(k)}[(\boldsymbol{V}\boldsymbol{X})^\top (\boldsymbol{V}\boldsymbol{X})]\boldsymbol{e}_t = \mathbb{E}^{(k)}[(\boldsymbol{V}\boldsymbol{e}_{x_s})^\top \boldsymbol{V}\boldsymbol{e}_{x_t}] = \sum_{n=1}^{N} \mathbb{E}\, V_{n,x_s} V_{n,x_t} = \begin{cases} \|V\|_\mu^2, & s = t, \\ \|\boldsymbol{\mu}\|^2, & s \neq t. \end{cases}$$

In matrix form, this is

$$\mathbb{E}^{(k)}(\boldsymbol{V}\boldsymbol{X})^\top \boldsymbol{V}\boldsymbol{X} = \left( \|V\|_\mu^2 - \|\boldsymbol{\mu}\|^2 \right) \boldsymbol{I} + \mathbf{1}\mathbf{1}^\top \|\boldsymbol{\mu}\|^2.$$

Then, by Lemma B.6, for each $t \in [T]$, we have

$$\boldsymbol{e}_t^\top \, \mathbb{E}[(\boldsymbol{V}\boldsymbol{X})^\top \boldsymbol{e}_{x_o}] = \mathbb{E}^{(k)}[(\boldsymbol{V}\boldsymbol{e}_{x_t})^\top \boldsymbol{e}_{x_o}] = \mathbb{E}^{(k)} V_{x_o, x_t}$$

$$= q_t^{(k)} \langle \boldsymbol{V}, \boldsymbol{P} \rangle_\mu + \left(1 - q_t^{(k)}\right) \|\boldsymbol{\mu}\|^2.$$

In matrix for, this is

$$\mathbb{E}[(\boldsymbol{V}\boldsymbol{X})^\top \boldsymbol{e}_{x_o}] = \boldsymbol{q}^{(k)} \langle \boldsymbol{V}, \boldsymbol{P} \rangle_\mu + \left(\mathbf{1} - \boldsymbol{q}^{(k)}\right) \|\boldsymbol{\mu}\|^2.$$

Combine these together, and we obtain

$$\mu_k^{-1} \, \mathbb{E}\, \nabla_{\boldsymbol{a}^{(k)}} l = \mathbb{E}^{(k)} \left[ (\boldsymbol{V}\boldsymbol{X})^\top \boldsymbol{V}\boldsymbol{X} \right] \boldsymbol{a}^{(k)} - \mathbb{E}^{(k)} \left[ (\boldsymbol{V}\boldsymbol{X})^\top \boldsymbol{e}_{x_o} \right]$$

$$= \left( \|V\|_\mu^2 - \|\boldsymbol{\mu}\|^2 \right) \boldsymbol{a}^{(k)} + \mathbf{1} \|\boldsymbol{\mu}\|^2 - \boldsymbol{q}^{(k)} \langle \boldsymbol{V}, \boldsymbol{P} \rangle_\mu - \left(\mathbf{1} - \boldsymbol{q}^{(k)}\right) \|\boldsymbol{\mu}\|^2.$$

$\square$

*Proof of Lemma B.8.* Recall from Lemma B.7 that

$$\mathbb{E}\, \nabla_V l = \|A\|_\mu^2 \, \boldsymbol{V} \mathrm{diag}(\boldsymbol{\mu}) + \left(1 - \|A\|_\mu^2\right) \boldsymbol{V}\boldsymbol{\mu}\boldsymbol{\mu}^\top$$

$$- \langle \boldsymbol{Q}, \boldsymbol{A} \rangle_\mu \, \boldsymbol{P} \mathrm{diag}(\boldsymbol{\mu}) - \left(1 - \langle \boldsymbol{Q}, \boldsymbol{A} \rangle_\mu \right) \boldsymbol{\mu}\boldsymbol{\mu}^\top,$$

$$\mathbb{E}\, \nabla_{\boldsymbol{a}^{(k)}} l = \mu_k \left( \|V\|_\mu^2 - \|\boldsymbol{\mu}\|^2 \right) \boldsymbol{a}^{(k)} + \mu_k \mathbf{1} \|\boldsymbol{\mu}\|^2$$

$$- \mu_k \boldsymbol{q}^{(k)} \langle \boldsymbol{V}, \boldsymbol{P} \rangle_\mu - \mu_k \left(\mathbf{1} - \boldsymbol{q}^{(k)}\right) \|\boldsymbol{\mu}\|^2.$$

For $\nabla_V$, we have

$$V\mathrm{diag}(\boldsymbol{\mu})\mathrm{diag}(1/\boldsymbol{\mu})\left(I - \frac{\boldsymbol{\mu}\boldsymbol{\mu}^\top}{\|\boldsymbol{\mu}\|^2}\right) = V - \frac{\boldsymbol{\mu}\boldsymbol{\mu}^\top}{\|\boldsymbol{\mu}\|^2},$$

$$P\mathrm{diag}(\boldsymbol{\mu})\mathrm{diag}(1/\boldsymbol{\mu})\left(I - \frac{\boldsymbol{\mu}\boldsymbol{\mu}^\top}{\|\boldsymbol{\mu}\|^2}\right) = P - \frac{\boldsymbol{\mu}\boldsymbol{\mu}^\top}{\|\boldsymbol{\mu}\|^2},$$

$$\boldsymbol{\mu}\boldsymbol{\mu}^\top\mathrm{diag}(1/\boldsymbol{\mu})\left(I - \frac{\boldsymbol{\mu}\boldsymbol{\mu}^\top}{\|\boldsymbol{\mu}\|^2}\right) = \boldsymbol{\mu}\mathbf{1}^\top - \frac{\boldsymbol{\mu}\boldsymbol{\mu}^\top}{\|\boldsymbol{\mu}\|^2}.$$

In particular, note that the $\boldsymbol{\mu}\boldsymbol{\mu}^\top/\|\boldsymbol{\mu}\|^2$ terms will cancel with each other. Thus, we have

$$\mathbb{E}\,\hat{\nabla}_V l = \|A\|_\mu^2\, V + \left(1 - \|A\|_\mu^2\right)\boldsymbol{\mu}\mathbf{1}^\top - \langle Q, A\rangle_\mu\, P - \left(1 - \langle Q, A\rangle_\mu\right)\boldsymbol{\mu}\mathbf{1}^\top$$

$$= \|A\|_\mu^2\left(V - \boldsymbol{\mu}\mathbf{1}^\top\right) - \langle Q, A\rangle_\mu\left(P - \boldsymbol{\mu}\mathbf{1}^\top\right).$$

For $\hat{\nabla}_{\boldsymbol{a}^{(k)}} l$, we have

$$\mathbb{E}\,\hat{\nabla}_{\boldsymbol{a}^{(k)}} l = \left(\|V\|_\mu^2 - \|\boldsymbol{\mu}\|^2\right)\left(I - \frac{\mathbf{1}\mathbf{1}^\top}{T}\right)\boldsymbol{a}^{(k)} - \left(I - \frac{\mathbf{1}\mathbf{1}^\top}{T}\right)\boldsymbol{q}^{(k)}\,\langle V, P\rangle_\mu$$

$$+ \left(I - \frac{\mathbf{1}\mathbf{1}^\top}{T}\right)\boldsymbol{q}^{(k)}\,\|\boldsymbol{\mu}\|^2$$

$$= \left(\|V\|_\mu^2 - \|\boldsymbol{\mu}\|^2\right)\left(\boldsymbol{a}^{(k)} - \frac{\mathbf{1}}{T}\right) - \left(\boldsymbol{q}^{(k)} - \frac{\mathbf{1}}{T}\right)\langle V, P\rangle_\mu + \left(\boldsymbol{q}^{(k)} - \frac{\mathbf{1}}{T}\right)\|\boldsymbol{\mu}\|^2$$

$$= \left(\|V\|_\mu^2 - \|\boldsymbol{\mu}\|^2\right)\left(\boldsymbol{a}^{(k)} - \frac{\mathbf{1}}{T}\right) - \left(\langle V, P\rangle_\mu - \|\boldsymbol{\mu}\|^2\right)\left(\boldsymbol{q}^{(k)} - \frac{\mathbf{1}}{T}\right).$$

$\square$

## B.4  Concentration

In this section, we provide concentration inequalities for the gradients of the loss function. The concentration is applied on the gradient noise term

$$\boldsymbol{h}_{V,\tau} := \left(\hat{\nabla}_V^{(B)} l - \mathbb{E}\,\hat{\nabla}_V l\right)$$

$$\boldsymbol{h}_{A,\tau} := \left(\hat{\nabla}_A^{(B)} l - \mathbb{E}\,\hat{\nabla}_A l\right)$$

where $\hat{\nabla}_V^{(B)} l$ and $\hat{\nabla}_A^{(B)} l$ are the preconditioned empirical gradients computed from a batch of size $B$. Here, we first consider the concentration of the original gradients:

$$\nabla_V l = \left(VX\boldsymbol{a} - \boldsymbol{e}_{x_o}\right)(X\boldsymbol{a})^\top,$$

$$\nabla_{\boldsymbol{a}^{(k)}} l = \mathbb{1}\{x_{T+1} = k\}(VX)^\top\left(VX\boldsymbol{a} - \boldsymbol{e}_{x_o}\right)$$

and then consider the concentration of the preconditioned gradients. In this paper, we focus on $\|\cdot\|_\mu$ as the mostly used metric for the gradient matrices.

First we prove a naive concentration w.r.t. any random vector $\boldsymbol{y}$ with bounded second moment with any $\|\cdot\|$.

**Lemma B.9.** *Fix $\delta, \varepsilon > 0$. Let $\boldsymbol{y}$ be a $D$-dimensional random vector with $\mathbb{E}\,\|\boldsymbol{y}\|^2 \le G$. Define $\boldsymbol{y}^{(B)} := B^{-1}\sum_{i=1}^B \boldsymbol{y}_i$ where $(\boldsymbol{y}_i)_i$ are i.i.d. versions of $\boldsymbol{y}$. If*

$$B \ge \frac{G}{\delta\varepsilon^2},$$

*then with probability at least $1 - \delta$, we have $\|\boldsymbol{y} - \mathbb{E}\,\boldsymbol{y}\| \le \varepsilon$.*

*Proof of Lemma B.9.* Assume w.l.o.g. that $\mathbb{E}\,\boldsymbol{y} = 0$. First, note that

$$\mathbb{E}\left\|\boldsymbol{y}^{(B)}\right\|^2 = \frac{1}{B^2}\sum_{i,j=1}^N \mathbb{E}\,\langle\boldsymbol{y}_i, \boldsymbol{y}_j\rangle = \frac{1}{B^2}\sum_{i=1}^N \mathbb{E}\,\|\boldsymbol{y}_i\|^2 \le \frac{G}{B}.$$

Hence, by the Markov inequality, we have

$$\mathbb{P}\left(\left\|\boldsymbol{y}^{(B)}\right\| \geq \varepsilon\right) = \mathbb{P}\left(\left\|\boldsymbol{y}^{(B)}\right\|^2 \geq \varepsilon^2\right) \leq \frac{\mathbb{E}\left\|\boldsymbol{y}^{(B)}\right\|^2}{\varepsilon^2} \leq \frac{G}{B\varepsilon^2}.$$

Thus, for fixed $\varepsilon, \delta \in (0, 1)$, if we choose $B = G/(\delta\varepsilon^2)$, then we have with probability at least $1 - \delta$, $\left\|\boldsymbol{y}^{(B)}\right\| \leq \varepsilon$. $\qquad\square$

Now we upper bound the infinity norm of the preconditioned gradients to apply concentration.

**Lemma B.10.** *Suppose that* $\|\text{Vec}\,\boldsymbol{V}\|_\infty = O(1), \boldsymbol{V} = \tilde{\boldsymbol{V}} + \boldsymbol{\Delta}_V, \boldsymbol{A} = \tilde{\boldsymbol{A}} + \boldsymbol{\Delta}_A,$ *where* $\tilde{\boldsymbol{V}} \in \mathbb{R}^{N \times N}$ *is a transition probability matrix, and in the attention matrix* $\tilde{\boldsymbol{A}}$ *each column* $\tilde{\boldsymbol{a}}^{(k)}$ *is a probability vector. Moreover,* $\|\boldsymbol{\Delta}_A\|_F^2, \|\boldsymbol{\Delta}_V\|_F^2 \leq O(1/T)$. *Then, we have*

$$\left\|\hat{\nabla}_{\boldsymbol{a}^{(k)}} l\right\|_\infty \leq O(N), \quad \left\|\text{Vec}\,\hat{\nabla}_V l\right\|_\infty \leq O(N)$$

*Proof.* We first consider the infinity norm of the original gradient. Recall that the gradient for $\boldsymbol{V}$ and $\boldsymbol{A}$ are

$$\nabla_{\boldsymbol{a}^{(k)}} l = \mathbb{1}\{x_{T+1} = k\}(\boldsymbol{V}\boldsymbol{X})^\top (\boldsymbol{V}\boldsymbol{X}\boldsymbol{a}^{(k)} - \boldsymbol{e}_{x_o})$$
$$\nabla_V l = \left(\boldsymbol{V}\boldsymbol{X}\boldsymbol{a} - \boldsymbol{e}_{x_o}\right)(\boldsymbol{X}\boldsymbol{a})^\top$$

and the preconditioned gradient is:

$$\hat{\nabla}_V^{(B_\tau)} l := \left(\boldsymbol{I}_N - \frac{\boldsymbol{1}_N \boldsymbol{1}_N^\top}{N}\right)\left(\nabla_V^{(B_\tau)} l\right) \text{diag}(1/\boldsymbol{\mu})\left(\boldsymbol{I}_N - \frac{\boldsymbol{\mu}\boldsymbol{\mu}^\top}{\|\boldsymbol{\mu}\|^2}\right),$$

$$\hat{\nabla}_{\boldsymbol{a}^{(k)}}^{(B_\tau)} l := \frac{1}{\mu_k}\left(\boldsymbol{I}_T - \frac{\boldsymbol{1}_T \boldsymbol{1}_T^\top}{T}\right)\left(\nabla_{\boldsymbol{a}^{(k)}}^{(B_\tau)} l\right).$$

We first consider the maximum absolute value in the original gradients. For $\nabla_{\boldsymbol{a}^{(k)}} l$, we have

$$\left\|\mathbb{1}\{x_{T+1} = k\}(\boldsymbol{V}\boldsymbol{X})^\top (\boldsymbol{V}\boldsymbol{X}\boldsymbol{a}^{(k)} - \boldsymbol{e}_{x_o})\right\|_\infty$$

$$\leq \left\|\mathbb{1}\{x_{T+1} = k\}(\boldsymbol{V}\boldsymbol{X})^\top (\boldsymbol{V}\boldsymbol{X}\boldsymbol{a}^{(k)})\right\|_\infty + \left\|\mathbb{1}\{x_{T+1} = k\}(\boldsymbol{V}\boldsymbol{X})^\top (\boldsymbol{e}_{x_o})\right\|_\infty$$

$$\leq \max_{s \in [T]}\left|\sum_{t=1}^T a_t (\boldsymbol{V}\boldsymbol{e}_{x_s})^\top \boldsymbol{V}\boldsymbol{e}_{x_t}\right| + \max_{s \in [T]}\left|(\boldsymbol{V}\boldsymbol{e}_{x_s})^\top \boldsymbol{e}_{x_o}\right|$$

The first term can be upper-bounded in the following way:

$$\max_{s \in [T]}\left|\sum_{t=1}^T a_t (\boldsymbol{V}\boldsymbol{e}_{x_s})^\top \boldsymbol{V}\boldsymbol{e}_{x_t}\right| = \max_{s \in [T]}\left|\sum_{t=1}^T \tilde{a}_t (\boldsymbol{V}\boldsymbol{e}_{x_s})^\top \boldsymbol{V}\boldsymbol{e}_{x_t} + \sum_{t=1}^T \Delta_{\boldsymbol{a},t}(\boldsymbol{V}\boldsymbol{e}_{x_s})^\top \boldsymbol{V}\boldsymbol{e}_{x_t}\right|$$

$$\leq \left(1 + \sum_{t=1}^T \|\boldsymbol{\Delta}_{\boldsymbol{a},t}\|_1\right)\max_{s,t}\left|(\boldsymbol{V}\boldsymbol{e}_{x_s})^\top \boldsymbol{V}\boldsymbol{e}_{x_t}\right|$$

$$\leq (1 + \sqrt{T}\|\boldsymbol{\Delta}_{\boldsymbol{a},t}\|_2)\max_{s,t}\left|(\boldsymbol{V}\boldsymbol{e}_{x_s})^\top \boldsymbol{V}\boldsymbol{e}_{x_t}\right|$$

Since $\boldsymbol{V} = \tilde{\boldsymbol{V}} + \boldsymbol{\Delta}_V$ and $\|\boldsymbol{\Delta}_V\|_F^2, \|\boldsymbol{\Delta}_A\|_F^2 \leq O(1/T)$, we have $\max_{s,t}\left|(\boldsymbol{V}\boldsymbol{e}_{x_s})^\top \boldsymbol{V}\boldsymbol{e}_{x_t}\right|$ upper bounded by $O(1)$. Therefore

$$\max_{s \in [T]}\left|\sum_{t=1}^T a_t (\boldsymbol{V}\boldsymbol{e}_{x_s})^\top \boldsymbol{V}\boldsymbol{e}_{x_t}\right| \leq O(1)$$

And similarly, the second term $\max_{s \in [T]}\left|(\boldsymbol{V}\boldsymbol{e}_{x_s})^\top \boldsymbol{e}_{x_o}\right|$ can be bounded by $O(1)$ because the infinity norm of $\boldsymbol{V}$ is also upper bounded by $O(1)$. Therefore, we know $\|\nabla_{\boldsymbol{a}^{(k)}} l\|_\infty \leq O(1)$.

Now we consider the preconditioned gradient $\hat{\nabla}_{\boldsymbol{a}^{(k)}}^{(B_\tau)} l$:

$$\|\hat{\nabla}_{\boldsymbol{a}^{(k)}}^{(B_\tau)} l\|_\infty = \left\|\frac{1}{\mu_k}\left(\boldsymbol{I}_T - \frac{\boldsymbol{1}_T \boldsymbol{1}_T^\top}{T}\right)\left(\nabla_{\boldsymbol{a}^{(k)}}^{(B_\tau)} l\right)\right\|_\infty$$

$$\leq \left\| \frac{1}{\mu_k} \boldsymbol{I}_T \left( \nabla_{\boldsymbol{a}^{(k)}}^{(B_\tau)} l \right) \right\|_\infty + \left\| \frac{1}{\mu_k} \left( \frac{\mathbf{1}_T \mathbf{1}_T^\top}{T} \right) \left( \nabla_{\boldsymbol{a}^{(k)}}^{(B_\tau)} l \right) \right\|_\infty \leq O(N)$$

since $\mu_k \geq \frac{c}{N}$ for all $k \in [N]$.

We use similar technique on $\hat{\nabla}_{\boldsymbol{V}}^{(B_\tau)} l$. First, we prove the infinity norm upper bound on the original gradient.

$$\begin{aligned}
\|\text{Vec} \nabla_{\boldsymbol{V}} l\|_\infty &= \left\| \text{Vec} \left( \boldsymbol{V} \boldsymbol{X} \boldsymbol{a} - \boldsymbol{e}_{x_o} \right) (\boldsymbol{X} \boldsymbol{a})^\top \right\|_\infty \\
&= \left\| \text{Vec} \left( \boldsymbol{V} \boldsymbol{X} \boldsymbol{a} \right) (\boldsymbol{X} \boldsymbol{a})^\top \right\|_\infty + \left\| \text{Vec} \left( \boldsymbol{e}_{x_o} (\boldsymbol{X} \boldsymbol{a})^\top \right) \right\|_\infty \\
&= \max_{s,t \in [T]} \left| (\boldsymbol{V} \boldsymbol{X} \boldsymbol{a})_s (\boldsymbol{X} \boldsymbol{a})_t^\top \right| + \left\| \sum_{t=1}^T a_t \text{Vec} \left( \boldsymbol{e}_{x_o} \boldsymbol{e}_{x_t}^\top \right) \right\|_\infty \\
&\leq \| (\boldsymbol{V} \boldsymbol{X} \boldsymbol{a}) \|_\infty \left\| (\boldsymbol{X} \boldsymbol{a})_t^\top \right\|_\infty + \left\| \sum_{t=1}^T a_t \text{Vec} \left( \boldsymbol{e}_{x_o} \boldsymbol{e}_{x_t}^\top \right) \right\|_\infty \\
&= \sum_{s,t=1}^T (\tilde{a}_s + \|\Delta_{\boldsymbol{A},s}\|_1)(\tilde{a}_t + \|\Delta_{\boldsymbol{A},t}\|_1) \left\| \boldsymbol{V} \boldsymbol{e}_{x_s} \right\|_\infty \left\| \boldsymbol{e}_{x_t} \right\|_\infty \\
&\quad + \sum_{t=1}^T (\tilde{a}_t + \|\Delta_{\boldsymbol{A},t}\|_1) \left\| \text{Vec} \left( \boldsymbol{e}_{x_o} \boldsymbol{e}_{x_t}^\top \right) \right\|_\infty \leq \Theta(1).
\end{aligned}$$

And therefore, the preconditioned gradient can also be bounded.

$$\begin{aligned}
\|\hat{\nabla}_{\boldsymbol{V}}^{(B_\tau)} l\|_\infty &= \left\| \left( \boldsymbol{I}_N - \frac{\mathbf{1}_N \mathbf{1}_N^\top}{N} \right) \left( \nabla_{\boldsymbol{V}}^{(B_\tau)} l \right) \text{diag}(1/\boldsymbol{\mu}) \left( \boldsymbol{I}_N - \frac{\boldsymbol{\mu} \boldsymbol{\mu}^\top}{\|\boldsymbol{\mu}\|^2} \right) \right\|_\infty \\
&\leq \left\| \left( \nabla_{\boldsymbol{V}}^{(B_\tau)} l \right) \text{diag}(1/\boldsymbol{\mu}) \left( \boldsymbol{I}_N - \frac{\boldsymbol{\mu} \boldsymbol{\mu}^\top}{\|\boldsymbol{\mu}\|^2} \right) \right\|_\infty \\
&\quad + \left\| \frac{\mathbf{1}_N \mathbf{1}_N^\top}{N} \left( \nabla_{\boldsymbol{V}}^{(B_\tau)} l \right) \text{diag}(1/\boldsymbol{\mu}) \left( \boldsymbol{I}_N - \frac{\boldsymbol{\mu} \boldsymbol{\mu}^\top}{\|\boldsymbol{\mu}\|^2} \right) \right\|_\infty \\
&\leq O(N)
\end{aligned}$$

since $\mu_k \geq \frac{c}{N}$ for all $k \in [N]$. Now we finished the proof. $\qquad \square$

With the upper bound of the infinity norm, we have the following upper bound on the second order moments of the preconditioned gradients of $\boldsymbol{A}$ and $\boldsymbol{V}$.

**Corollary B.11.** *With the same setting in Lemma B.10 and $\left\| \hat{\nabla}_{\boldsymbol{a}^{(k)}} l \right\|_\infty \leq O(N), \left\| \text{Vec} \hat{\nabla}_{\boldsymbol{V}} l \right\|_\infty \leq O(N)$. Moreover, $\|\Delta_{\boldsymbol{A}}\|_F^2, \|\Delta_{\boldsymbol{V}}\|_F^2 \leq O(1/T)$. Then, we have*

$$\mathbb{E} \left\| \hat{\nabla}_{\boldsymbol{A}} l \right\|_\mu^2 \leq O(TN^2), \quad \mathbb{E} \left\| \hat{\nabla}_{\boldsymbol{V}} l \right\|_\mu^2 \leq O(N^3)$$

*Proof.* We directly upper bound $\left\| \hat{\nabla}_{\boldsymbol{A}} l \right\|_\mu^2$ and $\left\| \hat{\nabla}_{\boldsymbol{V}} l \right\|_\mu^2$ using the upper bound on infinity norm. Since $\|\Delta_{\boldsymbol{A}}\|_F^2, \|\Delta_{\boldsymbol{V}}\|_F^2 \leq O(1/T)$, we have the infinity norm be upper bounded by

$$\left\| \hat{\nabla}_{\boldsymbol{a}^{(k)}} l \right\|_\infty \leq O(N), \quad \left\| \text{Vec} \hat{\nabla}_{\boldsymbol{V}} l \right\|_\infty \leq O(N)$$

Then, we can first bound the Frobenius norm $\|\hat{\nabla}_{\boldsymbol{A}} l\|_F^2, \|\hat{\nabla}_{\boldsymbol{V}} l\|_F^2$. We have $\boldsymbol{V} \in \mathbb{R}^{N \times N}, \boldsymbol{A} \in \mathbb{R}^{T \times N}$, so

$$\|\hat{\nabla}_{\boldsymbol{V}} l\|_F^2 \leq N^2 \left\| \hat{\nabla}_{\boldsymbol{V}} l \right\|_\infty^2 = O(N^4), \quad \|\hat{\nabla}_{\boldsymbol{A}} l\|_F^2 \leq NT \left\| \text{Vec} \hat{\nabla}_{\boldsymbol{A}} l \right\|_\infty^2 = O(N^3 T).$$

That leads to:

$$\left\| \hat{\nabla}_{\boldsymbol{V}} l \right\|_\mu^2 = \langle \hat{\nabla}_{\boldsymbol{V}} l, \hat{\nabla}_{\boldsymbol{V}} l \, \text{diag}(\boldsymbol{\mu}) \rangle \leq O\left( \frac{1}{N} \right) \|\hat{\nabla}_{\boldsymbol{V}} l\|_F^2 \leq O(N^3)$$

$$\left\| \hat{\nabla}_{\boldsymbol{A}} l \right\|_\mu^2 = \langle \hat{\nabla}_{\boldsymbol{A}} l, \hat{\nabla}_{\boldsymbol{A}} l \, \text{diag}(\boldsymbol{\mu}) \rangle \leq O\left( \frac{1}{N} \right) \|\hat{\nabla}_{\boldsymbol{A}} l\|_F^2 \leq O(N^2 T)$$

where the second inequality comes from the assumption that $\boldsymbol{\mu} \sim \Theta(1/N)$. $\qquad \square$

Now with the upper bound of the second moments of the gradients, we begin to prove the concentration of the gradients. We first consider the first-order terms that need to be bounded in the signal dynamics:

$$\frac{\langle \boldsymbol{h}_V, \boldsymbol{P} \rangle_\mu}{K_P K_Q}, \quad \frac{\langle \boldsymbol{h}_A, \boldsymbol{Q} \rangle_\mu}{K_P K_Q}$$

**Lemma B.12.** *Fix $\varepsilon, \delta > 0$. Under Assumption 2.1, suppose $\left\|\hat{\nabla}_{\boldsymbol{a}^{(k)}} l\right\|_\infty \leq \Theta(N), \left\|\hat{\nabla}_V l\right\|_\infty \leq \Theta(N)$. If $B \geq \Theta(1) \max\left(N^4, Q^2 N^2\right) \frac{N^2 \log \frac{4}{\delta}}{\epsilon^2 K_P^2 K_Q^2}$, then with probability at least $1 - \delta$, we have:*

$$\left|\frac{\langle \boldsymbol{h}_V, \boldsymbol{P} \rangle_\mu}{K_P K_Q}\right| \leq \varepsilon, \quad \left|\frac{\langle \boldsymbol{h}_A, \boldsymbol{Q} \rangle_\mu}{K_P K_Q}\right| \leq \varepsilon.$$

*Proof.* Note $\boldsymbol{h}_A = \frac{1}{B} \sum_i \hat{\nabla}_{\boldsymbol{a}^{(k_i)}} l(i) - \mathbb{E} \hat{\nabla}_{\boldsymbol{a}^{(k)}} l$, thus we have the upper bound for each coordinate of the gradient error bounded by $\Theta(N)$. Similarly, we have the upper bound for each coordinate of the gradient error of $\boldsymbol{h}_V$ bounded by $\Theta(N)$.

Then, we can bound the infinity norm of $\langle \hat{\nabla}_V l(i), \boldsymbol{P} \rangle_\mu$ and $\langle \hat{\nabla}_A l(i), \boldsymbol{P} \rangle_\mu$:

$$
\begin{aligned}
\left|\frac{\langle \hat{\nabla}_V l(i), \boldsymbol{P} \rangle_\mu}{K_P K_Q}\right| &= \left|\frac{\sum_{n,m=1}^N \hat{\nabla}_V l(i)_{n,m} \boldsymbol{P}_{n,m}}{K_P K_Q}\right| \\
&\leq \frac{N^2 \left\|\hat{\nabla}_V l(i)\right\|_\infty \|\boldsymbol{P}\|_\infty}{K_P K_Q} \\
&\leq \frac{\Theta(1) N^3}{K_P K_Q}. && (\|\boldsymbol{P}\|_\infty \leq 1.) \\
\left|\frac{\langle \hat{\nabla}_A l(i), \boldsymbol{Q} \rangle_\mu}{K_P K_Q}\right| &= \left|\frac{\sum_{n=1}^T \sum_{m=1}^N \hat{\nabla}_A l(i)_{n,m} \boldsymbol{Q}_{n,m}}{K_P K_Q}\right| \\
&\leq \frac{QN \left\|\hat{\nabla}_A l(i)\right\|_\infty \|\boldsymbol{Q}\|_\infty}{K_P K_Q} && (\boldsymbol{Q} \text{ is Q-sparse.}) \\
&\leq \frac{\Theta(1) Q N^2}{K_P K_Q}. && (\|\boldsymbol{Q}\|_\infty \leq 1.)
\end{aligned}
$$

Note that $\mathbb{E}\left[\nabla_{\boldsymbol{a}^{(k_i)}} l(i) - \mathbb{E} \nabla_{\boldsymbol{a}^{(k)}} l\right] = 0, \mathbb{E}\left[\nabla_V l(i) - \mathbb{E} \nabla_V l\right] = 0$, which means the two terms above have expectation $0$. Since $\boldsymbol{h}_A, \boldsymbol{h}_V$ are both averages of $B$ gradients of a single sample, we use Hoeffding Inequality:

$$\mathbb{P}\left(\left|\frac{\langle \boldsymbol{h}_V, \boldsymbol{P} \rangle_\mu}{K_P K_Q}\right| \geq \varepsilon\right) \leq 2 \exp\left(\frac{-B\varepsilon^2 K_P^2 K_Q^2}{N^6}\right)$$

$$\mathbb{P}\left(\left|\frac{\langle \boldsymbol{h}_A, \boldsymbol{Q} \rangle_\mu}{K_P K_Q}\right| \geq \varepsilon\right) \leq 2 \exp\left(\frac{-B\varepsilon^2 K_P^2 K_Q^2}{N^4 Q^2}\right)$$

By union bound, if $B \geq \max\left(N^2, Q^2\right) \frac{N^4 \log \frac{4}{\delta}}{\epsilon^2 K_P^2 K_Q^2}$ it has at least $1 - \delta$ probability, s.t.

$$\left|\frac{\langle \boldsymbol{h}_V, \boldsymbol{P} \rangle_\mu}{K_P K_Q}\right| \leq \varepsilon, \quad \left|\frac{\langle \boldsymbol{h}_A, \boldsymbol{Q} \rangle_\mu}{K_P K_Q}\right| \leq \varepsilon.$$

$\square$

Then we finish this section with the concentration of the second order terms $\|\boldsymbol{h}_A\|^2$ and $\|\boldsymbol{h}_V\|^2$, which need to be bounded in the error evolution.

**Lemma B.13.** *Fix $\varepsilon, \delta > 0$. Under Assumption 2.1, if $\mathbb{E} \|\hat{\nabla}_V l\|_\mu^2 = \texttt{Tmp}_3 = O(N^3), \mathbb{E} \|\hat{\nabla}_A l\|_\mu^2 = \texttt{Tmp}_4 = O(TN^2), B \geq \frac{\Theta(TN^2)}{\delta \epsilon^2}$, then with probability at least $1 - \delta$, we have:*

$$\|\boldsymbol{h}_A\|_\mu \leq \varepsilon, \quad \|\boldsymbol{h}_V\|_\mu \leq \varepsilon.$$

*Proof.* Similar to Lemma B.12, $\mathbb{E}\,\boldsymbol{h}_V = 0, \mathbb{E}\,\boldsymbol{h}_A = 0$. By Lemma B.9, when we pick $B \geq \frac{\Theta(TN^2)}{\delta\epsilon^2}$ we have $\|\boldsymbol{h}_A\|_\mu \leq \varepsilon, \|\boldsymbol{h}_V\|_\mu \leq \varepsilon$ with probability at least $1 - \delta$. $\qquad\square$

# C   The population projected process

In this section, we define the projection of the true SGD process onto the "space of population trajectories". Then, we derive formulas for the dynamics of projected process and the distance of the true SGD process to the space of population trajectories. All proofs — except for those short ones — are deferred to the end of this section.

## C.1   Definition of the population projection

The main reason we analyze the population process first is that on the population trajectory, both layers possess special structures. Recall that

$$\mathbb{E}\,\hat{\nabla}_V l = \|A\|_\mu^2 \left(V - \mu\mathbf{1}^\top\right) - \langle Q, A\rangle_\mu \left(P - \mu\mathbf{1}^\top\right),$$

$$\mathbb{E}\,\hat{\nabla}_A l = \left(\|V\|_\mu^2 - \|\mu\|^2\right)\left(A - \frac{\mathbf{1}_T\mathbf{1}_N^\top}{T}\right) - \left(\langle V, P\rangle_\mu - \|\mu\|^2\right)\left(Q - \frac{\mathbf{1}_T\mathbf{1}_N^\top}{T}\right),$$

and we initialize $V_0 - \mu\mathbf{1}^\top = 0$ and $A - \mathbf{1}\mathbf{1}^\top/T = 0$. Note that for any $(z_\tau)_\tau$, if $z_0 = 0$ and $\dot{z} = Az + Bz_*$, then $z_\tau \propto z_*$ for all $\tau \geq 0$. In other words, $z_\tau$ moves only along the direction $z_*$ and therefore, can be characterized by a single real number. This is exactly the same case of $V - \mu\mathbf{1}^\top$ and $A - \mathbf{1}\mathbf{1}^\top/T$ in the population case. Hence, in the population case, $V$ stays on the line crossing $\mu\mathbf{1}^\top$ and $P$, and $A$ stays on the line crossing $\mathbf{1}\mathbf{1}^\top/T$ and $Q$.

Unfortunately, mini-batch SGD does not stay exactly on the population trajectory. We can still, however, look at the projection of SGD onto the "population trajectories". Formally, for any $V, A$ satisfying $\mathbf{1}_N^\top V = \mathbf{1}_N^\top, \mathbf{1}_T^\top A = \mathbf{1}_N^\top$, and $V\mu = \mu$, we define

$$\alpha_V := \underset{\alpha\in\mathbb{R}}{\operatorname{argmin}} \left\|\alpha P + (1 - \alpha)\mu\mathbf{1}_N^\top - V\right\|_\mu^2,$$

$$\alpha_A := \underset{\alpha\in\mathbb{R}}{\operatorname{argmin}} \left\|\alpha Q + (1 - \alpha)\frac{\mathbf{1}_T\mathbf{1}_N^\top}{T} - A\right\|_\mu^2.$$

By setting the derivative to be 0, we can obtain the following closed-form formulas for $\alpha_V$ and $\alpha_A$.

**Note:** Without specification, we drop the the time subscript $\tau$ and consider $\alpha_V := \alpha_{V,\tau}$ for similicity.

**Lemma C.1.** *For any $V, A$ satisfying $\mathbf{1}_N^\top V = \mathbf{1}_N^\top, \mathbf{1}_T^\top A = \mathbf{1}_N^\top$, and $V\mu = \mu$, we have*

$$\alpha_V = K_{VP}/K_P \quad and \quad \alpha_A = K_{AQ}/K_Q,$$

*where $K_P = \|P\|_\mu^2 - \|\mu\|^2$, $K_{VP} = \langle V, P\rangle_\mu - \|\mu\|^2$, $K_Q = \|Q\|_\mu^2 - 1/T$, $K_{AQ} = \langle A, Q\rangle_\mu - 1/T$.*

For notational simplicity, we define $\beta_V = 1 - \alpha_V$, $\beta_A = 1 - \alpha_A$,

$$\tilde{V} = \alpha_V P + \beta_V \mu\mathbf{1}^\top \quad and \quad \tilde{A} = \alpha_A Q + \beta_A \frac{\mathbf{1}\mathbf{1}^\top}{T}.$$

Then, define $\Delta_V = V - \tilde{V}$ and $\Delta_A = A - \tilde{A}$ so that we can decompose $V = \tilde{V} + \Delta_V$ and $A = \tilde{A} + \Delta_A$. By our construction, we have $\tilde{V} - \mu\mathbf{1}^\top \perp \Delta_V$ and similarly for $\Delta_A$. We will now show that we can in fact drop $\mu\mathbf{1}^\top$.

**Lemma C.2.** *For any $V' = \theta P + (1 - \theta)\mu\mathbf{1}^\top$ and $A' = \theta Q + (1 - \theta)\mathbf{1}\mathbf{1}^\top/T$ with $\theta \in \mathbb{R}$, we have $\langle\Delta_V, V'\rangle_\mu = 0$ and $\langle\Delta_A, A'\rangle_\mu = 0$. In particular, we have $\langle\Delta_V, P\rangle_\mu = \langle\Delta_V, \tilde{V}\rangle_\mu = 0$ and $\langle\Delta_A, Q\rangle_\mu = \langle\Delta_A, \tilde{A}\rangle_\mu = 0$.*

*Proof.* Note that $\langle\mu\mathbf{1}^\top, \Delta_V\rangle_\mu = \langle\mu, (V - \tilde{V})\mu\rangle = 0$. Hence, $\langle\Delta_V, V'\rangle = \langle\Delta_V, V' - \mu\mathbf{1}^\top\rangle = 0$. For $\langle\Delta_A, A'\rangle$, it suffices to note that $\langle\Delta_A, \mathbf{1}\mathbf{1}^\top/T\rangle_\mu = \langle\mathbf{1}^\top(A - \tilde{A}), \mathbf{1}^\top/T\rangle_\mu = 0$. $\qquad\square$

The following lemma the basic definitions and results about the population projection.

**Lemma C.3** (Definitions and basic results on the population projection). *Suppose that $V$, $A$ satisfy $\mathbf{1}_N^\top V = \mathbf{1}_N^\top$, $\mathbf{1}_T^\top A = \mathbf{1}_N^\top$, and $V\mu = \mu$. We define the following:*

$$K_P = \|P\|_\mu^2 - \|\mu\|^2, \quad K_{VP} = \langle V, P \rangle_\mu - \|\mu\|^2, \quad K_V = \|V\|_\mu^2 - \|\mu\|^2,$$

$$\alpha_V = K_{VP}/K_P, \quad \beta_V = 1 - \alpha_V, \quad \tilde{V} = \alpha_V P + \beta_V \mu \mathbf{1}^\top,$$

$$\Delta_V = V - \tilde{V},$$

$$K_Q = \|Q\|_\mu^2 - 1/T, \quad K_{AQ} = \langle A, Q \rangle_\mu - 1/T, \quad K_A = \|A\|_\mu^2 - 1/T,$$

$$\alpha_A = K_{AQ}/K_Q, \quad \beta_A = 1 - \beta_A, \quad \tilde{A} = \alpha_A A + \beta_A \mathbf{1}_T \mathbf{1}_N^\top/T,$$

$$\Delta_A = A - \tilde{A}.$$

*Moreover, by Lemma C.2, the following hold.*

$$K_{VP} = \langle \tilde{V}, P \rangle_\mu - \|\mu\|^2 = \alpha_V K_P,$$

$$K_V = \left\| \tilde{V} \right\|_\mu^2 + \|\Delta_V\|_\mu^2 - \|\mu\|^2 = \alpha_V^2 K_P + \|\Delta_V\|_\mu^2,$$

$$K_{AQ} = \langle \tilde{A}, Q \rangle_\mu - 1/T = \alpha_A K_Q,$$

$$K_A = \left\| \tilde{A} \right\|_\mu^2 + \|\Delta_A\|_\mu^2 - 1/T = \alpha_A^2 K_Q + \|\Delta_A\|_\mu^2.$$

## C.2  Dynamics of the population projected process and the approximation error

We write

$$V_{\tau+1} = V_\tau - \eta_V \mathbb{E} \hat{\nabla}_V l - \eta_V \left( \hat{\nabla}_V^{(B)} l - \mathbb{E} \hat{\nabla}_V l \right) =: V_\tau - \eta_V \mathbb{E} \hat{\nabla}_V l - \eta_V h_{V,\tau},$$

$$A_{\tau+1} = A_\tau - \eta_A \mathbb{E} \hat{\nabla}_A l - \eta_A \left( \hat{\nabla}_A^{(B)} l - \mathbb{E} \hat{\nabla}_A l \right) =: A_\tau - \eta_A \mathbb{E} \hat{\nabla}_A l - \eta_A h_{A,\tau},$$

where the expectations are taken over the fresh samples at step $\tau$.

First, we expand the expected preconditioned gradients around the population projection.

**Lemma C.4** (Expanding the gradients).

$$\mathbb{E} \hat{\nabla}_A l = K_P \alpha_V (\alpha_V \alpha_A - 1) \left( Q - \frac{\mathbf{1}_T \mathbf{1}_N^\top}{T} \right) + K_P \alpha_V^2 \Delta_A + \|\Delta_V\|_\mu^2 \left( A - \frac{\mathbf{1}_T \mathbf{1}_N^\top}{T} \right),$$

$$\mathbb{E} \hat{\nabla}_V l = \alpha_A K_Q (\alpha_A \alpha_V - 1) (P - \mu \mathbf{1}^\top) + \frac{\alpha_V - 1}{T} (P - \mu \mathbf{1}^\top)$$

$$+ \left( \alpha_A^2 K_Q + \frac{1}{T} \right) \Delta_V + \|\Delta_A\|^2 (V - \mu \mathbf{1}^\top).$$

Then, we compute the dynamics of the projected process.

**Lemma C.5** (Dynamics of the population projection).

$$\alpha_{V,\tau+1} = \alpha_{V,\tau} + \eta_V K_Q (1 - \alpha_A \alpha_V) \alpha_A + \eta_V \frac{1 - \alpha_V}{T} - \eta_V \alpha_V \|\Delta_A\|^2 - \frac{\eta_V}{K_P} \langle h_{V,\tau}, P \rangle_\mu,$$

$$\alpha_{A,\tau+1} = \alpha_{A,\tau} + \eta_A K_P (1 - \alpha_V \alpha_A) \alpha_V - \eta_A \alpha_A \|\Delta_V\|_\mu^2 - \frac{\eta_A}{K_Q} \langle h_{A,\tau}, Q \rangle_\mu.$$

*Note that $\tilde{V} = \mu \mathbf{1}^\top + \alpha_V (P - \mu \mathbf{1}^\top)$ and $\tilde{A} = \mathbf{1}_T \mathbf{1}_N^\top/T + \alpha_A (Q - \mathbf{1}_T \mathbf{1}_N^\top/T)$. Hence, this also gives formulas for $\tilde{V}_{\tau+1}$ and $\tilde{A}_{\tau+1}$.*

Now, we consider the dynamics of the errors.

**Lemma C.6** (Dynamics of the errors).

$$\left\| \Delta_{A,\tau+1} \right\|_\mu^2 = \left( 1 - \eta_A K_P \alpha_V^2 - \eta_A \|\Delta_V\|_\mu^2 \right)^2 \|\Delta_A\|_\mu^2$$

$$- 2\eta_A \left( 1 - \eta_A K_P \alpha_V^2 - \eta_A \|\Delta_V\|_\mu^2 \right) \langle \Delta_A, h_A \rangle_\mu$$

$$- \frac{\eta_A^2}{K_Q} \langle h_{A,\tau}, Q \rangle_\mu^2 + \eta_A^2 \|h_A\|_\mu^2,$$

$$\|\Delta_{V,\tau+1}\|_\mu^2 = \left(1 - \eta_V \left(\alpha_A^2 K_Q + \frac{1}{T}\right) - \eta_V \|\Delta_A\|^2\right)^2 \|\Delta_V\|_\mu^2$$

$$+ 2\eta_V \left(1 - \eta_V \left(\alpha_A^2 K_Q + \frac{1}{T}\right) - \eta_V \|\Delta_A\|^2\right) \langle \Delta_V, h_V \rangle_\mu$$

$$- \frac{\eta_V^2}{K_P} \langle P, h_V \rangle_\mu^2 + \eta_V^2 \|h_V\|_\mu^2.$$

### C.3 Omitted proofs in this section

*Proof of Lemma C.1.* We compute

$$\frac{1}{2}\partial_\alpha \left\|\alpha P + (1 - \alpha)\mu \mathbf{1}^\top - V\right\|_\mu^2 = \left\langle \alpha P + (1 - \alpha)\mu \mathbf{1}^\top - V, P - \mu \mathbf{1}^\top \right\rangle_\mu$$

$$= \left\langle \alpha P + (1 - \alpha)\mu \mathbf{1}^\top - V, P \right\rangle_\mu = \alpha K_P + \|\mu\|^2 - \langle V, P \rangle_\mu.$$

Set the derivative to be 0, and we get $\alpha_V = (\langle V, P \rangle_\mu - \|\mu\|^2)/K_P$. Similarly, we compute

$$\frac{1}{2}\partial_\alpha \left\|\alpha Q + (1 - \alpha)\frac{\mathbf{1}\mathbf{1}^\top}{T} - A\right\|_\mu^2 = \left\langle \alpha Q + (1 - \alpha)\frac{\mathbf{1}\mathbf{1}^\top}{T} - A, Q - \frac{\mathbf{1}\mathbf{1}^\top}{T}\right\rangle_\mu$$

$$= \left\langle \alpha Q + (1 - \alpha)\frac{\mathbf{1}\mathbf{1}^\top}{T} - A, Q \right\rangle_\mu$$

$$= \alpha \left(\|Q\|_\mu^2 - 1/T\right) - \left(\langle A, Q \rangle_\mu - 1/T\right).$$

Again, set the derivative to be 0, and we get $\alpha_A = \left(\langle A, Q \rangle_\mu - 1/T\right)/(\|Q\|_\mu^2 - 1/T)$. $\qquad \square$

*Proof of Lemma C.4.* Recall from Lemma B.8 that

$$\mathbb{E}\,\hat{\nabla}_V l = \left(K_A + \frac{1}{T}\right)(V - \mu \mathbf{1}^\top) - \left(K_{AQ} + \frac{1}{T}\right)(P - \mu \mathbf{1}^\top),$$

$$\mathbb{E}\,\hat{\nabla}_A l = K_V \left(A - \frac{\mathbf{1}_T \mathbf{1}_N^\top}{T}\right) - K_{VP} \left(Q - \frac{\mathbf{1}_T \mathbf{1}^\top}{T}\right).$$

First, consider the dynamics of $A$. By Lemma C.3, we can further decompose it as

$$\mathbb{E}\,\hat{\nabla}_A l = \left(\alpha_V^2 K_P + \|\Delta_V\|_\mu^2\right)\left(A - \frac{\mathbf{1}_T \mathbf{1}_N^\top}{T}\right) - \alpha_V K_P \left(Q - \frac{\mathbf{1}_T \mathbf{1}^\top}{T}\right)$$

$$= K_P \alpha_V \left(\alpha_V \left(A - \frac{\mathbf{1}_T \mathbf{1}_N^\top}{T}\right) - \left(Q - \frac{\mathbf{1}_T \mathbf{1}^\top}{T}\right)\right) + \|\Delta_V\|_\mu^2 \left(A - \frac{\mathbf{1}_T \mathbf{1}_N^\top}{T}\right)$$

$$= K_P \alpha_V \left(\alpha_V \left(\tilde{A} - \frac{\mathbf{1}_T \mathbf{1}_N^\top}{T}\right) - \left(Q - \frac{\mathbf{1}_T \mathbf{1}^\top}{T}\right)\right) + K_P \alpha_V^2 \Delta_A + \|\Delta_V\|_\mu^2 \left(A - \frac{\mathbf{1}_T \mathbf{1}_N^\top}{T}\right)$$

$$= K_P \alpha_V (\alpha_V \alpha_A - 1)\left(Q - \frac{\mathbf{1}_T \mathbf{1}_N^\top}{T}\right) + K_P \alpha_V^2 \Delta_A + \|\Delta_V\|_\mu^2 \left(A - \frac{\mathbf{1}_T \mathbf{1}_N^\top}{T}\right).$$

Similarly, we can rewrite the expected preconditioned gradient of $V$ as

$$\mathbb{E}\,\hat{\nabla}_V l = \left(\alpha_A^2 K_Q + \frac{1}{T}\right)(V - \mu \mathbf{1}^\top) - \left(\alpha_A K_Q + \frac{1}{T}\right)(P - \mu \mathbf{1}^\top) + \|\Delta_A\|^2 (V - \mu \mathbf{1}^\top)$$

$$= \left(\alpha_A^2 K_Q + \frac{1}{T}\right)(\tilde{V} - \mu \mathbf{1}^\top) - \left(\alpha_A K_Q + \frac{1}{T}\right)(P - \mu \mathbf{1}^\top)$$

$$+ \left(\alpha_A^2 K_Q + \frac{1}{T}\right)\Delta_V + \|\Delta_A\|^2 (V - \mu \mathbf{1}^\top)$$

$$= \alpha_A K_Q \left(\alpha_A \alpha_V - 1\right) \left(\boldsymbol{P} - \boldsymbol{\mu} \boldsymbol{1}^\top\right) + \frac{\alpha_V - 1}{T} \left(\boldsymbol{P} - \boldsymbol{\mu} \boldsymbol{1}^\top\right)$$
$$+ \left(\alpha_A^2 K_Q + \frac{1}{T}\right) \boldsymbol{\Delta}_V + \|\boldsymbol{\Delta}_A\|^2 \left(\boldsymbol{V} - \boldsymbol{\mu} \boldsymbol{1}^\top\right).$$

$\square$

*Proof of Lemma C.5.* Recall that $\alpha_V = \langle \boldsymbol{V}, \boldsymbol{P}\rangle_\mu / K_P$ and $\alpha_A = \langle \boldsymbol{A}, \boldsymbol{Q}\rangle_\mu / K_Q$. First, consider the dynamics of $\boldsymbol{V}$. By Lemma C.4 and Lemma C.3, we have

$$\alpha_{V,\tau+1} = \alpha_{V,\tau} - \eta_V \frac{\left\langle \hat{\nabla}_V \mathcal{L} + \boldsymbol{h}_{V,\tau}, \boldsymbol{P}\right\rangle_\mu}{K_P}$$

$$= \alpha_{V,\tau} - \frac{\eta_V}{K_P} \alpha_A K_Q \left(\alpha_A \alpha_V - 1\right) \left\langle \boldsymbol{P} - \boldsymbol{\mu} \boldsymbol{1}^\top, \boldsymbol{P}\right\rangle_\mu - \frac{\eta_V}{K_P} \frac{\alpha_V - 1}{T} \left\langle \boldsymbol{P} - \boldsymbol{\mu} \boldsymbol{1}^\top, \boldsymbol{P}\right\rangle_\mu$$

$$- \frac{\eta_V}{K_P} \|\boldsymbol{\Delta}_A\|^2 \left\langle \boldsymbol{V} - \boldsymbol{\mu} \boldsymbol{1}^\top, \boldsymbol{P}\right\rangle_\mu - \eta_V \frac{\left\langle \boldsymbol{h}_{V,\tau}, \boldsymbol{P}\right\rangle_\mu}{K_P}$$

$$= \alpha_{V,\tau} + \eta_V K_Q \left(1 - \alpha_A \alpha_V\right) \alpha_A + \eta_V \frac{1 - \alpha_V}{T} - \eta_V \alpha_V \|\boldsymbol{\Delta}_A\|^2 - \frac{\eta_V}{K_P} \left\langle \boldsymbol{h}_{V,\tau}, \boldsymbol{P}\right\rangle_\mu.$$

Similarly, for $\boldsymbol{V}$, we have

$$\alpha_{A,\tau+1} = \alpha_{A,\tau} - \eta_A \frac{\left\langle \hat{\nabla}_A \mathcal{L} + \boldsymbol{h}_{A,\tau}, \boldsymbol{Q}\right\rangle_\mu}{K_Q}$$

$$= \alpha_{A,\tau} - \frac{\eta_A}{K_Q} K_P \alpha_V \left(\alpha_V \alpha_A - 1\right) \left\langle \boldsymbol{Q} - \frac{\boldsymbol{1}_T \boldsymbol{1}_N^\top}{T}, \boldsymbol{Q}\right\rangle_\mu$$

$$- \frac{\eta_A}{K_Q} \|\boldsymbol{\Delta}_V\|_\mu^2 \left\langle \boldsymbol{A} - \frac{\boldsymbol{1}_T \boldsymbol{1}_N^\top}{T}, \boldsymbol{Q}\right\rangle_\mu - \eta_A \frac{\left\langle \boldsymbol{h}_{A,\tau}, \boldsymbol{Q}\right\rangle_\mu}{K_Q}$$

$$= \alpha_{A,\tau} + \eta_A K_P \left(1 - \alpha_V \alpha_A\right) \alpha_V - \eta_A \alpha_A \|\boldsymbol{\Delta}_V\|_\mu^2 - \frac{\eta_A}{K_Q} \left\langle \boldsymbol{h}_{A,\tau}, \boldsymbol{Q}\right\rangle_\mu.$$

$\square$

*Proof of Lemma C.6.* First, consider the dynamics of $\boldsymbol{\Delta}_A$, which is given by

$$\boldsymbol{\Delta}_{A,\tau+1} = \boldsymbol{\Delta}_A - \eta_A K_P \alpha_V^2 \boldsymbol{\Delta}_A - \eta_A \|\boldsymbol{\Delta}_V\|_\mu^2 \left(\boldsymbol{A} - \frac{\boldsymbol{1}_T \boldsymbol{1}_N^\top}{T}\right) - \eta_A \boldsymbol{h}_A$$

$$+ \left(\eta_A \alpha_A \|\boldsymbol{\Delta}_V\|_\mu^2 + \frac{\eta_A}{K_Q} \left\langle \boldsymbol{h}_{A,\tau}, \boldsymbol{Q}\right\rangle_\mu\right) \left(\boldsymbol{Q} - \frac{\boldsymbol{1}_T \boldsymbol{1}_N^\top}{T}\right).$$

Decompose $\boldsymbol{A}$ into $\tilde{\boldsymbol{A}} + \boldsymbol{\Delta}_A$, rearrange terms, and we obtain

$$\boldsymbol{\Delta}_{A,\tau+1} = \boldsymbol{\Delta}_A - \eta_A K_P \alpha_V^2 \boldsymbol{\Delta}_A - \eta_A \|\boldsymbol{\Delta}_V\|_\mu^2 \boldsymbol{\Delta}_A$$

$$- \eta_A \|\boldsymbol{\Delta}_V\|_\mu^2 \left(\tilde{\boldsymbol{A}} - \frac{\boldsymbol{1}_T \boldsymbol{1}_N^\top}{T}\right) + \eta_A \alpha_A \|\boldsymbol{\Delta}_V\|_\mu^2 \left(\boldsymbol{Q} - \frac{\boldsymbol{1}_T \boldsymbol{1}_N^\top}{T}\right)$$

$$+ \frac{\eta_A}{K_Q} \left\langle \boldsymbol{h}_{A,\tau}, \boldsymbol{Q}\right\rangle_\mu \left(\boldsymbol{Q} - \frac{\boldsymbol{1}_T \boldsymbol{1}_N^\top}{T}\right) - \eta_A \boldsymbol{h}_A.$$

Note that $\tilde{\boldsymbol{A}} - \boldsymbol{1}\boldsymbol{1}^\top/T = \alpha_A \boldsymbol{Q} + (1 - \alpha_A)\boldsymbol{1}\boldsymbol{1}^\top/T - \boldsymbol{1}\boldsymbol{1}^\top/T = \alpha_A(\boldsymbol{Q} - \boldsymbol{1}\boldsymbol{1}^\top/T)$. Hence, we have

$$\boldsymbol{\Delta}_{A,\tau+1} = \boldsymbol{\Delta}_A - \eta_A K_P \alpha_V^2 \boldsymbol{\Delta}_A - \eta_A \|\boldsymbol{\Delta}_V\|_\mu^2 \boldsymbol{\Delta}_A$$

$$- \eta_A \alpha_A \|\boldsymbol{\Delta}_V\|_\mu^2 \left(\boldsymbol{Q} - \frac{\boldsymbol{1}_T \boldsymbol{1}_N^\top}{T}\right) + \eta_A \alpha_A \|\boldsymbol{\Delta}_V\|_\mu^2 \left(\boldsymbol{Q} - \frac{\boldsymbol{1}_T \boldsymbol{1}_N^\top}{T}\right)$$

$$+ \frac{\eta_A}{K_Q} \left\langle \boldsymbol{h}_{A,\tau}, \boldsymbol{Q}\right\rangle_\mu \left(\boldsymbol{Q} - \frac{\boldsymbol{1}_T \boldsymbol{1}_N^\top}{T}\right) - \eta_A \boldsymbol{h}_A$$

$$= \left(1 - \eta_A K_P \alpha_V^2 - \eta_A \|\Delta_V\|_\mu^2 \Delta_A\right) \Delta_A + \frac{\eta_A}{K_Q} \langle h_{A,\tau}, Q \rangle_\mu \left(Q - \frac{\mathbf{1}_T \mathbf{1}_N^\top}{T}\right) - \eta_A h_A.$$

Recall that $\langle \Delta_A, Q - \mathbf{1}\mathbf{1}^\top/T \rangle_\mu = 0$, $\|Q - \mathbf{1}\mathbf{1}^\top/T\|_\mu^2 = K_Q$, and $\langle \mathbf{1}_T \mathbf{1}_N^\top/T, h_A \rangle_\mu = 0$. Hence,

$$\|\Delta_{A,\tau+1}\|_\mu^2 = \left\|\left(1 - \eta_A K_P \alpha_V^2 - \eta_A \|\Delta_V\|_\mu^2\right) \Delta_A + \frac{\eta_A}{K_Q} \langle h_{A,\tau}, Q \rangle_\mu \left(Q - \frac{\mathbf{1}_T \mathbf{1}_N^\top}{T}\right) - \eta_A h_A \right\|_\mu^2$$

$$= \left(1 - \eta_A K_P \alpha_V^2 - \eta_A \|\Delta_V\|_\mu^2\right)^2 \|\Delta_A\|_\mu^2 + \left(\frac{\eta_A}{K_Q} \langle h_{A,\tau}, Q \rangle_\mu\right)^2 K_Q + \eta_A^2 \|h_A\|_\mu^2$$

$$\quad - 2\eta_A \left(1 - \eta_A K_P \alpha_V^2 - \eta_A \|\Delta_V\|_\mu^2\right) \langle \Delta_A, h_A \rangle_\mu$$

$$\quad - 2\frac{\eta_A^2}{K_Q} \langle h_{A,\tau}, Q \rangle_\mu \langle Q, h_A \rangle_\mu$$

$$= \left(1 - \eta_A K_P \alpha_V^2 - \eta_A \|\Delta_V\|_\mu^2\right)^2 \|\Delta_A\|_\mu^2 - 2\eta_A \left(1 - \eta_A K_P \alpha_V^2 - \eta_A \|\Delta_V\|_\mu^2\right) \langle \Delta_A, h_A \rangle_\mu$$

$$\quad + \eta_A^2 \|h_A\|_\mu^2 - \frac{\eta_A^2}{K_Q} \langle h_{A,\tau}, Q \rangle_\mu^2.$$

Now, consider $\Delta_V$. Similar to the previous calculation, we have

$$\Delta_{V,\tau+1} = \Delta_V - \eta_V \left(\alpha_A^2 K_Q + \frac{1}{T}\right) \Delta_V - \eta_V \|\Delta_A\|^2 \Delta_V - \eta_V h_V$$

$$\quad - \eta_V \frac{\alpha_V - 1}{T} \left(P - \mu \mathbf{1}^\top\right) - \eta_V \|\Delta_A\|^2 \left(\tilde{V} - \mu \mathbf{1}^\top\right)$$

$$\quad - \left(\eta_V \frac{1 - \alpha_V}{T} - \eta_V \alpha_V \|\Delta_A\|^2 - \frac{\eta_V}{K_P} \langle h_{V,\tau}, P \rangle_\mu\right) \left(P - \mu \mathbf{1}^\top\right)$$

$$= \left(1 - \eta_V \left(\alpha_A^2 K_Q + \frac{1}{T}\right) - \eta_V \|\Delta_A\|^2\right) \Delta_V$$

$$\quad + \frac{\eta_V}{K_P} \langle h_{V,\tau}, P \rangle_\mu \left(P - \mu \mathbf{1}^\top\right) - \eta_V h_V.$$

Again, note that $\Delta_V \perp_\mu P - \mu \mathbf{1}^\top$, $\|P - \mu \mathbf{1}^\top\|_\mu^2 = K_P$, and $\langle \mu \mathbf{1}^\top, h_V \rangle_\mu = 0$. Hence, we have

$$\|\Delta_{V,\tau+1}\|_\mu^2 = \left(1 - \eta_V \left(\alpha_A^2 K_Q + \frac{1}{T}\right) - \eta_V \|\Delta_A\|^2\right)^2 \|\Delta_V\|_\mu^2$$

$$\quad + 2\eta_V \left(1 - \eta_V \left(\alpha_A^2 K_Q + \frac{1}{T}\right) - \eta_V \|\Delta_A\|^2\right) \langle \Delta_V, h_V \rangle_\mu$$

$$\quad - \frac{\eta_V^2}{K_P} \langle P, h_V \rangle_\mu^2 + \eta_V^2 \|h_V\|_\mu^2.$$

$\square$

## D   Stage 1: signal boosting

In this section, we assume both $\alpha_V$ and $\alpha_A$ are close to 0. In this case, we can approximate Lemma C.5 with

$$\alpha_{V,\tau+1} \approx \alpha_{V,\tau} + \eta \alpha_A + \eta \frac{1}{T K_Q} - \eta \alpha_V \frac{\|\Delta_A\|_\mu^2}{K_Q} - \eta \frac{\langle h_V, P \rangle_\mu}{K_Q K_P},$$

$$\alpha_{A,\tau+1} \approx \alpha_{A,\tau} + \eta \alpha_V - \eta \alpha_A \frac{\|\Delta_V\|_\mu^2}{K_P} - \eta \frac{\langle h_A, Q \rangle_\mu}{K_P K_Q}.$$

We can also write this matrix form as

$$\begin{bmatrix} \alpha_{V,\tau+1} \\ \alpha_{A,\tau+1} \end{bmatrix} \approx \begin{bmatrix} \alpha_{V,\tau} \\ \alpha_{A,\tau} \end{bmatrix} + \eta \begin{bmatrix} -\|\Delta_A\|_\mu^2 / K_Q & 1 \\ 1 & -\|\Delta_V\|_\mu^2 / K_P \end{bmatrix} \begin{bmatrix} \alpha_{V,\tau} \\ \alpha_{A,\tau} \end{bmatrix}$$

$$+ \eta \begin{bmatrix} 1/(TK_Q) \\ 0 \end{bmatrix} - \eta \begin{bmatrix} \alpha_A \|\Delta_V\|_\mu^2 /(K_P K_Q) \\ \langle \boldsymbol{h}_A, \boldsymbol{Q} \rangle_\mu /(K_P K_Q). \end{bmatrix}$$

Suppose that $\|\Delta_A\|_\mu^2 / K_Q$ and $\|\Delta_V\|_\mu^2 / K_P$ are both bounded by $\delta^2$. Then, we have

$$\alpha_{V,\tau+1} + \alpha_{A,\tau+1} \gtrsim \left(1 + \eta - 2\delta^2 \eta\right) \left(\alpha_{V,\tau} + \alpha_{A,\tau}\right) + \eta \frac{1}{TK_Q}$$
$$- \eta \frac{\langle \boldsymbol{h}_V, \boldsymbol{P} \rangle_\mu}{K_Q K_P} - \eta \frac{\langle \boldsymbol{h}_A, \boldsymbol{Q} \rangle_\mu}{K_P K_Q}. \tag{11}$$

As long as $\delta \ll 1$ and we choose a sufficiently large batch size so that the second line is bounded by $\eta/(2TK_Q)$, $\alpha_V + \alpha_A$ grows exponentially fast. Similarly, one can also bound the difference between $\alpha_V$ and $\alpha_A$. Formally, we have the following lemma.

**Lemma D.1** (Main result of Stage 1). *Define the end of Stage 1 as*

$$\mathcal{T}_1 := \min \left\{ \tau \geq 0 \; : \; \max\{\alpha_{V,\tau}, \alpha_{A,\tau}\} \geq \min \left\{ \frac{1}{2}, \frac{\Theta(1)}{QN} \right\} \right\}.$$

*Suppose $\eta \leq 1/10, \eta_{V,\tau} = \eta/K_Q$, and $\eta_{A,\tau} = \eta/K_P$ for some $\eta \leq \min\{K_P, K_Q\}$. Let $B_\tau > 0$ be the number fresh samples we use at step $\tau$. Suppose that $B_\tau \geq \tilde{O}\left( \frac{T^2 Q^4 N^5}{\min\{K_P^3, K_Q^3\}} \right)$ are chosen s.t. with probability $1 - \delta_\tau$*

$$\max \left\{ \left| \frac{\langle \boldsymbol{h}_V, \boldsymbol{P} \rangle_\mu}{K_Q K_P} \right|, \left| \frac{\langle \boldsymbol{h}_A, \boldsymbol{Q} \rangle_\mu}{K_P K_Q} \right| \right\} \leq \frac{1}{4TK_Q}, \tag{12}$$

$$\max \left\{ \|\boldsymbol{h}_{V,\tau}\|_\mu, \|\boldsymbol{h}_{A,\tau}\|_\mu \right\} \leq \frac{\Theta(1) \min \left\{ K_Q^{1/2}, K_P^{1/2}, \frac{1}{\sqrt{QN^{1/2}}} \right\} \min\{K_Q, K_P\}}{T \log TK_Q} \tag{13}$$

*Then, the following hold with probability at least $1 - \delta_P$:*

*(a) $\mathcal{T}_1 \leq \Theta(1) \log(TK_Q)/\eta$.*

*(b) Throughout Stage 1, $\|\Delta_A\|_\mu^2 \leq \Delta/T$ and $\|\Delta_V\|_\mu^2 \leq \Delta/T$, where*

$$\Delta := \min \left\{ \frac{1}{4} K_Q, \frac{1}{4} K_P, \frac{\Theta(1)}{Q^4 N^3} \right\}.$$

*(c) At $\mathcal{T}_1$, we have $\alpha_V + \alpha_A = \Theta(\frac{1}{QN})$ and $|\alpha_V - \alpha_A| \leq \frac{\Theta(1)}{TK_Q}$.*

*(d) For all $\tau \leq \mathcal{T}_1$, we have $\|V\|_\infty \leq \Theta(1), \|A\|_\infty \leq \Theta(1/Q)$.*

The proof of this lemma is a large induction argument. We will first assume the bounds on $\|\Delta_A\|_\mu$ and $\|\Delta_V\|_\mu$ are true, so that the approximation (11) is valid. This will give us an upper bound on the length of Stage 1. Then, we show that within this many steps, the errors cannot exceed the given maximum values. Thus, the induction hypotheses are true and Lemma D.1 can be established.

**Part I of the proof of Lemma D.1: Signal growth rate**

*Proof.* Since $\alpha_V \alpha_A \leq \frac{1}{4}$ by definition of Stage I, we can rewrite Lemma C.5 as

$$\alpha_{V,\tau+1} + \alpha_{A,\tau+1} \geq \left(1 + 3\eta/4\right) \left(\alpha_{V,\tau} + \alpha_{A,\tau}\right) + \eta \frac{3}{4TK_Q}$$
$$- \eta \alpha_V \frac{\|\Delta_A\|_\mu^2}{K_Q} - \eta \alpha_A \frac{\|\Delta_V\|_\mu^2}{K_P} - \eta \frac{\langle \boldsymbol{h}_V, \boldsymbol{P} \rangle_\mu}{K_Q K_P} - \eta \frac{\langle \boldsymbol{h}_A, \boldsymbol{Q} \rangle_\mu}{K_P K_Q}$$
$$\geq \left(1 + 3\eta/4\right) \left(\alpha_{V,\tau} + \alpha_{A,\tau}\right) + \eta \frac{3}{4TK_Q} - \frac{1}{4} \eta \alpha_V - \frac{1}{4} \eta \alpha_A - \frac{1}{4TK_Q} \times 2$$

$$\geq \left(1 + \frac{\eta}{2}\right)\left(\alpha_{V,\tau} + \alpha_{A,\tau}\right) + \eta \frac{1}{4TK_Q},$$

where the second line comes from induction hypothesis (b) and (12). Recursively expand the RHS, and we obtain

$$\alpha_{V,\tau} + \alpha_{A,\tau} \geq \left(1 + \frac{\eta}{2}\right)^{\tau}\left(\alpha_{V,0} + \alpha_{A,0} + \frac{1}{2TK_Q}\right) - \frac{1}{2TK_Q} = \left(\left(1 + \frac{\eta}{2}\right)^{\tau} - 1\right)\frac{1}{2TK_Q}.$$

Since the RHS is upper bounded by 1 by definition of stage I, and we obtain

$$\mathcal{T}_I \leq \Theta(1)\frac{\log(TK_Q)}{\eta}.$$

$\square$

**Part II of the proof of Lemma D.1: Upper bounds on $\|\Delta\|_\mu$**

*Proof.* Recall from Lemma C.6 that

$$\left\|\Delta_{V,\tau+1}\right\|_\mu^2 \leq \left\|\Delta_{V,\tau}\right\|_\mu^2 + 2\eta_V \left\|\Delta_{V,\tau}\right\|_\mu \left\|h_{V,\tau}\right\|_\mu + \eta_V^2 \left\|h_{V,\tau}\right\|_\mu^2,$$

$$\left\|\Delta_{A,\tau+1}\right\|_\mu^2 \leq \left\|\Delta_{A,\tau}\right\|_\mu^2 + 2\eta_A \left\|\Delta_{A,\tau}\right\|_\mu \left\|h_{A,\tau}\right\|_\mu + \eta_A^2 \left\|h_{A,\tau}\right\|_\mu^2.$$

By part I of the proof, Stage 1 takes at most $\Theta(1)\log(TK_Q)/\eta$ steps. Hence, it suffices to bound the increase of these $\|\Delta\|_\mu^2$ in this many steps. Recall the notation $\Delta = \min\left\{\frac{1}{4}K_Q, \frac{1}{4}K_P, \frac{\Theta(1)}{QN^{1/2}}\right\}$. By induction hypothesis (b), we have for all $\tau$,

$$\left\|\Delta_{V,\tau+1}\right\|_\mu^2 \leq \left\|\Delta_{V,\tau}\right\|_\mu^2 + 2\eta_V \left\|\Delta_{V,\tau}\right\|_\mu \left\|h_{V,\tau}\right\|_\mu + \eta_V^2 \left\|h_{V,\tau}\right\|_\mu^2$$

$$\leq \left\|\Delta_{V,\tau}\right\|_\mu^2 + 2\eta_V \sqrt{\Delta/T} \left\|h_{V,\tau}\right\|_\mu + \eta_V^2 \left\|h_{V,\tau}\right\|_\mu^2$$

$$\leq \left\|\Delta_{V,\tau}\right\|_\mu^2 + \frac{\Theta(1)\eta\sqrt{\Delta/T}}{K_Q} \cdot \frac{\sqrt{\Delta/T}\min\{K_P, K_Q\}}{\log(TK_Q)} + \eta^2 \frac{\Theta(1)\Delta \cdot \min\{K_P^2, K_Q^2\}}{TK_Q^2 \log(TK_Q)^2}$$

$$\left\|\Delta_{A,\tau+1}\right\|_\mu^2 \leq \left\|\Delta_{A,\tau}\right\|_\mu^2 + 2\eta_A \left\|\Delta_{A,\tau}\right\|_\mu \left\|h_{A,\tau}\right\|_\mu + \eta_A^2 \left\|h_{A,\tau}\right\|_\mu^2$$

$$\leq \left\|\Delta_{A,\tau}\right\|_\mu^2 + 2\eta_A \sqrt{\Delta/T} \left\|h_{A,\tau}\right\|_\mu + \eta_A^2 \left\|h_{A,\tau}\right\|_\mu^2$$

$$\leq \left\|\Delta_{A,\tau}\right\|_\mu^2 + \frac{\Theta(1)\eta\sqrt{\Delta/T}}{K_P} \cdot \frac{\sqrt{\Delta/T}\min\{K_P, K_Q\}}{\log(TK_Q)} + \eta^2 \frac{\Theta(1)\Delta \cdot \min\{K_P^2, K_Q^2\}}{TK_P^2 \log(TK_Q)^2}$$

Since Stage I at most takes $\mathcal{T}_I = \Theta(1)\log(TK_Q)/\eta$ steps, the increase of $\left\|\Delta_{A,\tau}\right\|_\mu^2$ and $\left\|\Delta_{A,\tau}\right\|_\mu^2$ in $\tau \leq \mathcal{T}_I$ are at most (since $\Delta_{A,0} = 0, \Delta_{V,0} = 0$):

$$\left\|\Delta_{V,t}\right\|_\mu^2 \leq \left(\frac{\Theta(1)\eta\sqrt{\Delta/T}}{K_Q} \cdot \frac{\sqrt{\Delta/T}\min\{K_P, K_Q\}}{\log(TK_Q)} + \eta^2 \frac{\Theta(1)\Delta \cdot \min\{K_P^2, K_Q^2\}}{TK_Q^2 \log(TK_Q)}\right)t$$

$$\leq \left(\frac{\Theta(1)\eta\sqrt{\Delta/T}}{K_Q} \cdot \frac{\sqrt{\Delta/T}\min\{K_P, K_Q\}}{\log(TK_Q)^2} + \eta^2 \frac{\Theta(1)\Delta \cdot \min\{K_P^2, K_Q^2\}}{TK_Q^2 \log(TK_Q)^2}\right)\mathcal{T}_I \leq \Delta/T,$$

$$\left\|\Delta_{A,t}\right\|_\mu^2 \leq \left(\frac{\Theta(1)\eta\sqrt{\Delta/T}}{K_P} \cdot \frac{\sqrt{\Delta/T}\min\{K_P, K_Q\}}{\log(TK_Q)} + \eta^2 \frac{\Theta(1)\Delta \cdot \min\{K_P^2, K_Q^2\}}{TK_P^2 \log(TK_Q)^2}\right)t$$

$$\leq \left(\frac{\Theta(1)\eta\sqrt{\Delta/T}}{K_P} \cdot \frac{\sqrt{\Delta/T}\min\{K_P, K_Q\}}{\log(TK_Q)} + \eta^2 \frac{\Theta(1)\Delta \cdot \min\{K_P^2, K_Q^2\}}{TK_P^2 \log(TK_Q)^2}\right)\mathcal{T}_I \leq \Delta/T.$$

Therefore, we completed the induction for the error terms. $\square$

**Part III of the proof of Lemma D.1: Ending state**

*Proof.* First, consider the distance between $\alpha_V$ and $\alpha_A$. Similar to the part I of the proof, we rewrite Lemma C.5 as

$$\alpha_{V,\tau+1} - \alpha_{A,\tau+1} = -\left(1 - \eta\left(1 - \alpha_V\alpha_A\right)\right)\left(\alpha_{V,\tau} - \alpha_{A,\tau}\right) + \eta\frac{1 - \alpha_V}{TK_Q}$$

$$- \eta\alpha_V\frac{\|\Delta_A\|_\mu^2}{K_Q} + \eta\alpha_A\frac{\|\Delta_V\|_\mu^2}{K_P} - \eta\frac{\langle h_V, P\rangle_\mu}{K_QK_P} + \eta\frac{\langle h_A, Q\rangle_\mu}{K_PK_Q}$$

$$= -\left(1 - \eta/2\right)\left(\alpha_{V,\tau} - \alpha_{A,\tau}\right) \pm O\left(\eta\frac{1}{TK_Q}\right).$$

Thus, whenever $\alpha_{V,\tau} - \alpha_{A,\tau} \geq \Omega(1/(TK_Q))$, it will start to decrease. Since the amount of increase at each step is also upper bounded by $O(1/(TK_Q))$, this implies $|\alpha_{V,\tau} - \alpha_{A,\tau}| \leq O(1/(TK_Q))$. The other direction can be proved in the same way.

Finally, we bound the possible amount of overshot. By the part I of the proof, we can also upper bound the signal term growth

$$\alpha_{V,\tau+1} + \alpha_{V,\tau+1} \leq (1 + 2\eta)\left(\alpha_{V,\tau} + \alpha_{A,\tau}\right) + O\left(\frac{\eta}{TK_Q}\right).$$

Since $\alpha_{V,\tau} + \alpha_{A,\tau} \leq 1$ for all $\tau \leq \mathcal{T}_1$, we have $\alpha_V + \alpha_A = \Theta(\frac{1}{QN})$ at time $\mathcal{T}_1$. $\qquad\square$

**Part IV of the proof of Lemma D.1: Upper bound on Infinity norm of $V$ and $A$**

Here we consider the upper bound of the weights $V$ and $A$, which can be used in the concentration section below.

*Proof.* First, we upper bound the infinity norm of $V_\tau$.

$$\|V_\tau\|_\infty = \|\tilde{V}_\tau + \Delta_V\|_\infty \leq \|\tilde{V}_\tau\|_\infty + \|\Delta_V\|_\infty \leq \|\tilde{V}_\tau\|_\infty + \|\Delta_V\|_F.$$

and we can upper bound $\|\Delta_V\|_F$ by its $\mu$-norm:

$$\|\Delta_V\|_\mu^2 = \langle\Delta_V, \Delta_V\operatorname{diag}(\mu)\rangle \geq \frac{c}{N}\|\Delta_V\|_F^2.$$

Thus we have $\|V_\tau\|_\infty \leq \|\tilde{V}_\tau\|_\infty + \Theta(1)\sqrt{N}\|\Delta_V\|_\mu \leq \|\tilde{V}_\tau\|_\infty + \Theta(\frac{1}{Q^2N})$ by Induction hypothesis (b). And we can further bound $\|\tilde{V}_\tau\|_\infty$:

$$\|\tilde{V}_\tau\|_\infty = \|\alpha_{V,\tau}P + (1 - \alpha_{V,\tau})\mu\mathbf{1}^\top\|_\infty \leq \|P\|_\infty + \|\mu\mathbf{1}^\top\|_\infty \leq \Theta(1).$$

Therefore, we have $\|V_\tau\|_\infty \leq \Theta(1)$.

Similarly, for $A$ we have (since $\|\tilde{A}_\tau\|_\infty = \|\mathbf{1}_T\mathbf{1}_N^\top/T + \alpha_{A,\tau}(Q - \mathbf{1}_T\mathbf{1}_N^\top/T)\|_\infty \leq C/Q$.)

$$\|A_\tau\|_\infty = \|\tilde{A}_\tau + \Delta_A\|_\infty \leq \|\tilde{A}_\tau\|_\infty + \|\Delta_A\|_\infty \leq \|\tilde{A}_\tau\|_\infty + \|\Delta_A\|_F \leq \Theta(1/Q).$$

$\qquad\square$

**Part V of the proof of Lemma D.1: Concentration**

Finally, we need to ensure that with high probability, all the error terms $h$ cannot exceed the given bounds throughout $t \leq \mathcal{T}_1$. We use Lemma B.12 and Lemma B.13 to bound the concentration of the error terms.

By Lemma B.13 and union bound, we have that if $B_\tau \geq \frac{\Theta(1)\mathcal{T}_1 \cdot T^2Q^4N^5\mathcal{T}_1\log^2(TK_Q)}{\delta_\tau\min\{K_P^3, K_Q^3, 1\}}$, then with probability at least $1 - \delta_\tau/2$, the following holds for all $t \leq \mathcal{T}_1$:

$$\max\left\{\|h_{V,\tau}\|_\mu, \|h_{A,\tau}\|_\mu\right\} \leq \frac{\Theta(1)\min\left\{K_Q^{1/2}, K_P^{1/2}, \frac{1}{Q^2N^{3/2}}\right\}\min\{K_Q, K_P\}}{\sqrt{T}\log TK_Q}$$

By Lemma B.12 and union bound, we have that if $B_\tau \geq \dfrac{\Theta(1)\mathcal{T}_1 \cdot T^2 \max\{N^2, Q^2\} N^4 \log\left(\frac{16 \log\left(TK_Q\right)}{\delta_\tau \eta}\right)}{K_P^2}$, then

with probability at least $1 - \delta_\tau/2$, the following holds for all $t \leq \mathcal{T}_1$:

$$\left| \frac{\langle \boldsymbol{h}_V, \boldsymbol{P} \rangle_\mu}{K_P K_Q} \right| \leq \frac{1}{4TK_Q}, \quad \left| \frac{\langle \boldsymbol{h}_A, \boldsymbol{Q} \rangle_\mu}{K_P K_Q} \right| \leq \frac{1}{4TK_Q}.$$

And by union bound, we have that with probability at least $1 - \delta_\tau$, all the bounds above hold for all $t \leq \mathcal{T}_1$. Therefore, we conclude the proof of Lemma D.1.

# E  Stage 2: learning the model

In this Stage 2, we use a positive $\lambda$ for the $\ell_1$-regularization. In this case, we can write the update rule of each $a^{(k)}$ as

$$a_{\tau+1}^{(k,')} = a_\tau^{(k)} - \frac{\eta_A}{\mu_k} \nabla_{a^{(k)}}^{(B_\tau)} l, \qquad \text{(gradient descent step)},$$

$$a_{\tau+1,t}^{(k,'')} = \begin{cases} a_{\tau+1,t}^{(k,')} - \lambda, & \text{if } a_{\tau+1,t}^{(k,')} \geq \lambda, \\ 0, & \text{if } \left| a_{\tau+1,t}^{(k,')} \right| \leq \lambda, \end{cases} \qquad \text{(proximal step)}, \qquad (14)$$

$$a_{\tau+1}^{(k)} = a_{\tau+1}^{(k,'')} + \left( 1 - \mathbf{1}^\top a_{\tau+1}^{(k,'')} \right) \frac{\mathbf{1}}{T}, \qquad \text{(projection step)}.$$

For notational simplicity, we define

$$g_\tau^{(k)} := -\eta_\tau^{-1} (a_{\tau+1}^{(k)} - a_\tau^{(k)}).$$

We further define $G_{\lambda,\tau} = -\eta_A^{-1}(A_{\tau+1} - A_\tau)$ as the full gradient of the matrix $A$.

First, we will show that with appropriate rounding at the beginning of Stage 2, we can ensure $g_\tau^{(k)} \approx \eta \, \mathbb{E} \, \hat{\nabla}_{a^{(k)}} l$ using only $B_\tau \propto \log(T)$ fresh samples at each step (Section E.1).

## E.1  Rounding and gradient denoising

Recall that the first step of the Stage 2 is rounding each $a^{(k)}$ by setting all small coordinates to 0 and then projecting it back to the affine space the probability simplex lies. Since after Stage 1, there will be a separation between $a_t^{(k)}$ with $t \in q^{(k)}$ and $t \notin q^{(k)}$, this rounding step makes all $a_t^{(k)}$ with $t \notin q^{(k)}$ have the same small value.

Also recall that we use $\ell_1$-regularization and proximal gradients in the Stage 2. Effectively, the $\ell_1$-regularization ensures those useless $a_t^{(k)}$ are always 0 (before projection). Then, similar to the first rounding step, projection will again make then have the same small value.

In this subsection, we formalize the above argument. We show that the rounding step can recover the support of $q^{(k)}$, analyze its influence on $\alpha_A$ and the distance to the population subspace. Then, we analyze the effect of use of the proximal gradients and show that with $\text{poly}(N, Q) \log(T)$ fresh samples at each step, we can make sure the difference between update and the population update is small with high probability.

**Lemma E.1** (Separation between noise and signal). *Assume that at the beginning of Stage 2, we have*

$$\alpha_V \geq \Theta \left( \frac{Q}{T} + Q\sqrt{N} \|\Delta_A\|_\mu \right).$$

*Then, we can choose a threshold $\theta_{\text{Tmp}} = \Theta(\alpha_V/Q)$ s.t. $a_t^{(k)} \leq \theta_{\text{Tmp}}$ iff $q_t^{(k)} = 0$.*

*Proof.* Note that there exists some universal constant $c > 0$ such that $q_t^{(k)} \geq c/Q$ for all nonzero $q_t^{(k)}$ and $\mu_k \geq c/N$ for all $k \in [N]$. Recall that the population process $\tilde{A} = \alpha_A Q + (1 - \alpha_A)\mathbf{1}_T \mathbf{1}_N^\top / T$. Hence, for all $k \in [N]$ and $t \in [T]$,

$$\tilde{a}_t^{(k)} \leq 1/T, \quad \text{if } t \notin q^{(k)},$$

$$\tilde{a}_t^{(k)} \geq c\alpha_V/Q, \quad \text{if } t \in q^{(k)}.$$

Then, note that

$$\|\Delta_A\|_\mu^2 = \sum_{t=1}^T \sum_{k=1}^N \left( a_t^{(k)} - \tilde{a}_t^{(k)} \right)^2 \mu_k$$

$$\geq \frac{c}{N} \sum_{t=1}^T \sum_{k=1}^N \left( a_t^{(k)} - \tilde{a}_t^{(k)} \right)^2 \geq \frac{c}{N} \left\| \text{Vec}(\tilde{A} - A) \right\|_2^2 \geq \frac{c}{N} \left\| \text{Vec}(\tilde{A} - A) \right\|_\infty^2.$$

Hence, for any $k \in [N]$ and $t \in [T]$, we have $|a_t^{(k)} - \tilde{a}_t^{(k)}| \leq \|\Delta_A\|_\mu \sqrt{N/c}$. Combine these together, and we obtain

$$a_t^{(k)} \leq 1/T + \|\Delta_A\|_\mu \sqrt{N/c}, \quad \text{if } t \notin q^{(k)},$$
$$a_t^{(k)} \geq c\alpha_V/Q - \|\Delta_A\|_\mu \sqrt{N/c}, \quad \text{if } t \in q^{(k)}.$$

Hence, in order to get a separation,

$$1/T + \|\Delta_A\|_\mu \sqrt{N/c} \leq \frac{1}{2}\left(c\alpha_V/Q - \|\Delta_A\|_\mu \sqrt{N/c}\right) \quad \Leftarrow \quad \frac{2Q}{cT} + \frac{3Q\sqrt{N}}{c^{1.5}}\|\Delta_A\|_\mu \leq \alpha_V.$$

$\square$

**Lemma E.2** (Effect of rounding). *Under the conditions of Lemma E.1, choose $\lambda$ as in Lemma E.1, and set*

$$a^{(k)} \leftarrow \left(I - \frac{\mathbf{1}\mathbf{1}^\top}{T}\right)a^{(k)} \odot \left(\mathbb{1}\{a_t^{(k)} \geq \lambda\}\right)_{t=1}^T + \frac{\mathbf{1}}{T}$$

$$= a^{(k)} \odot \left(\mathbb{1}\{a_t^{(k)} \geq \lambda\}\right)_{t=1}^T + \left(1 - \sum_{t \in q^{(k)}} a_t^{(k)}\right)\frac{\mathbf{1}}{T}.$$

*We have $\alpha_A \leftarrow \alpha_A + O(1)/T$ and $\|\Delta_A\|_\mu^2 \leftarrow \|\Delta_A\|_\mu^2 + O(1)/T$.*

*Proof.* For notational simplicity, put $b^{(k)} = a^{(k)} \odot (\mathbb{1}\{a_t^{(k)} \geq \lambda\})_{t=1}^T$. By Lemma E.1, we know $b^{(k)}$ is supported within $q^{(k)}$. Set $b^{(k)} = \sum_{t=1}^T b_t^{(k)}$. Then, we can write

$$a^{(k)} \leftarrow b^{(k)} + \left(1 - b^{(k)}\right)\frac{\mathbf{1}}{T}.$$

Recall that $\alpha_A = K_{AQ}/K_Q$ where $K_{AQ} = \langle A, Q \rangle_\mu = \sum_{k=1}^N \mu_k \langle a^{(k)}, q^{(k)} \rangle$. Hence, for each $k \in [N]$, we have

$$\left\langle a^{(k)}, q^{(k)} \right\rangle \leftarrow \left\langle b^{(k)}, q^{(k)} \right\rangle + \left(1 - b^{(k)}\right)\left\langle \frac{\mathbf{1}}{T}, q^{(k)} \right\rangle$$

$$\leftarrow \left\langle a^{(k)}, q^{(k)} \right\rangle + \left(1 - b^{(k)}\right)\frac{1}{T}.$$

As a result,

$$\alpha_A \leftarrow \alpha_A + \frac{1}{T}\sum_{k=1}^N \mu_k\left(1 - b^{(k)}\right) = \alpha_A + \frac{O(1)}{T}.$$

Now, consider the effect of projection on the distance to the population subspace. We will use subscript new to indicate values after rounding and use notations such as $a^{(k)}$ to denote the values before rounding. Recall that $\|\Delta_A\|_\mu^2 = \sum_{k=1}^N \mu_k \|a^{(k)} - \tilde{a}^{(k)}\|^2$. We have

$$\left\|a_{\text{new}}^{(k)} - \tilde{a}_{\text{new}}^{(k)}\right\|^2 = \left\|b^{(k)} + (1 - b^{(k)})\frac{\mathbf{1}}{T} - \frac{\mathbf{1}}{T} - \alpha_{A,\text{new}}\left(q^{(k)} - \frac{\mathbf{1}}{T}\right)\right\|^2$$

$$= \left\|b^{(k)} - \alpha_{A,\text{new}}q^{(k)} - \left(b^{(k)} - \alpha_{A,\text{new}}\right)\frac{\mathbf{1}}{T}\right\|^2$$

$$= \left\|b^{(k)} - \alpha_{A,\text{new}}q^{(k)}\right\|^2 + \left(b^{(k)} - \alpha_{A,\text{new}}\right)^2 \frac{1}{T}$$

$$\quad - 2\left(b^{(k)} - \alpha_{A,\text{new}}\right)\left\langle b^{(k)} - \alpha_{A,\text{new}}q^{(k)}, \frac{\mathbf{1}}{T} \right\rangle$$

$$= \left\|b^{(k)} - (\alpha_A + O(1)/T)q^{(k)}\right\|^2 - \left(b^{(k)} - \alpha_{A,\text{new}}\right)^2 \frac{1}{T}.$$

Note that

$$\left\|a^{(k)} - \tilde{a}^{(k)}\right\|^2 = \left\|a^{(k)} - \alpha_A q^{(k)} - (1 - \alpha_A)\frac{\mathbf{1}}{T}\right\|^2 = \left\|a^{(k)} - \alpha_A q^{(k)}\right\|^2 - (1 - \alpha_A)^2\frac{1}{T}$$

$$\geq \left\| \boldsymbol{b}^{(k)} - \alpha_A \boldsymbol{q}^{(k)} \right\|^2 - (1 - \alpha_A)^2 \frac{1}{T}.$$

Hence,

$$\left\| \boldsymbol{a}^{(k)}_{\text{new}} - \tilde{\boldsymbol{a}}^{(k)}_{\text{new}} \right\|^2 - \left\| \boldsymbol{a}^{(k)} - \tilde{\boldsymbol{a}}^{(k)} \right\|^2$$

$$\leq \left\| \boldsymbol{b}^{(k)} - \alpha_A \boldsymbol{q}^{(k)} - \frac{O(1)}{T} \boldsymbol{q}^{(k)} \right\|^2 - \left\| \boldsymbol{b}^{(k)} - \alpha_A \boldsymbol{q}^{(k)} \right\|^2 + \frac{O(1)}{T} \leq \frac{O(1)}{T}.$$

Thus, $\|\Delta_A\|^2_\mu \leftarrow \|\Delta_A\|^2_\mu + O(1)/T$. $\qquad\qquad\qquad\qquad\qquad\qquad\qquad\qquad\qquad\qquad\square$

As we have seen in Stage 1, the norm of $\nabla_{\boldsymbol{a}^{(k)}} l$ can scale linearly with $T$. Hence, in order to make $\boldsymbol{h}_{\boldsymbol{a}^{(k)}, \tau}$ has size $O(1)$ in terms of $\|\cdot\|_2$, it is necessary to use poly$(T)$ samples, which is undesirable. However, note that all entries of $\nabla_{\boldsymbol{a}^{(k)}} l$ are bounded, and therefore are subgaussian. Hence, for each entry of $\nabla_{\boldsymbol{a}^{(k)}} l$, with $O(\log T)$ samples, we can make sure the relative error is small with probability at least $1 - 1/\text{poly}(T)$. By union bound, this means the $\|\cdot\|_\infty$ error can be made small using only $\log(T)$ samples. Note that there is a separation between the signal and noise parts of $\mathbb{E} \nabla_{\boldsymbol{a}^{(k)}} l$. This implies that we can distinguish them using $\log(T)$ samples and directly remove the noise part. Formally, we have the following lemma.

**Lemma E.3** (Gradient denoising). *Given* $\varepsilon \in \left( \Theta\left( \frac{Q^2 N^3}{\sqrt{T}} \right), 0.1 \right)$. *Suppose that for any* $k, m, n \in [N]$, $|a^{(k)}_t| \leq 1/T$ *if* $t \notin \boldsymbol{q}^{(k)}$, $|V_{n,m}| \leq O(1)$, *and* $\|\Delta_A\|_\mu \leq c(1-\alpha_V)/(\sqrt{N}Q)$ *and* $\|\Delta_V\|^2_\mu \leq c(1-\alpha_V)K_P$ *for some small constant* $c > 0$. *For the target accuracy* $\varepsilon_{\text{Tmp}} = \Theta(\frac{\varepsilon}{QN^2})$, *we have with probability* $1 - \delta_\tau$. *If we choose*

$$B_\tau \geq C \frac{N^8 Q^4}{\varepsilon^2 \alpha_V^2 K_P^2} \log \left( \frac{CNT}{\delta_\tau} \right),$$

*for some large constant* $C > 0$. *Then, for each* $k \in [N]$, *with probability at least* $1 - \delta_{\text{Tmp}}$, *we have*

$$\partial^{(B_\tau)}_{a^{(k)}_t} l = \left( 1 \pm \varepsilon_{\text{Tmp}} \right) \mathbb{E} \, \partial_{a^{(k)}_t} l \geq \Theta \left( \frac{\alpha_V K_P}{NQ} \right) \qquad \forall t \in \boldsymbol{q}^{(k)},$$

$$\partial^{(B_\tau)}_{a^{(k)}_t} l = O \left( \frac{1}{T} + \varepsilon_{\text{Tmp}} \frac{\alpha_V K_P}{NQ} \right) = O \left( \frac{\varepsilon K_P}{N^3 Q^2} \right) \qquad \forall t \notin \boldsymbol{q}^{(k)}.$$

*Proof.* For $s \in [T]$, recall that

$$\partial_{a^{(k)}_s} l = \mathbb{1}\{x_{T+1} = k\}(V \boldsymbol{e}_{x_s})^\top \left( VX\boldsymbol{a} - \boldsymbol{e}_{x_o} \right),$$

$$\mathbb{E} \, \partial_{a^{(k)}_s} l = \mu_k K_V a_s - \mu_k K_{VP} q_s.$$

First, consider the expectations. If $s \notin \boldsymbol{q}^{(k)}$, then by our assumption, we have $|\mathbb{E} \, \partial_{a^{(k)}_s} l| = \mu_k K_V a_s = O(K_V/(NT))$. Meanwhile, for $s \in \boldsymbol{q}^{(k)}$, we have

$$- \mathbb{E} \, \partial_{a^{(k)}_s} l \geq \mu_k K_{VP}(q_s - a_s) + \mu_k (K_V - K_{VP}) a_s$$

$$\geq \mu_k \left( \alpha_V (1 - \alpha_V) K_P q_s - \alpha_V K_P \|\Delta_A\|_\mu \sqrt{N} - \|\Delta_V\|^2_\mu a_s \right).$$

As a result, we have for any $k \in [N]$ and $s \in \boldsymbol{q}^{(k)}$,

$$- \mathbb{E} \, \partial_{a^{(k)}_s} l \geq c \frac{\alpha_V K_P}{NQ} \gg O \left( \frac{K_V}{NT} \right) \quad \Longleftarrow \quad \begin{cases} \|\Delta_A\|_\mu \leq c \dfrac{1 - \alpha_V}{Q\sqrt{N}}, \\[2mm] \|\Delta_V\|^2_\mu \leq c(1 - \alpha_V)K_P, \end{cases}$$

for some small constant $c > 0$. Then, for the size of each entry, we have

$$\left| \partial_{a^{(k)}_s} l \right| \leq \left| (V \boldsymbol{e}_{x_s})^\top VX\boldsymbol{a} \right| + \left| (V \boldsymbol{e}_{x_s})^\top \boldsymbol{e}_{x_o} \right| \leq \sum_{t \in \boldsymbol{q}^{(k)}} a_t \sum_{n=1}^N |V_{n,x_s} V_{n,x_t}| + |V_{x_o,x_s}| \leq N.$$

Therefore, $\partial_{a_s^{(k)}} l$ is $N^2 K_{\text{Tmp}}^4$-subgaussian, whence $\partial_{a_s^{(k)}}^{(B_\tau)} l$ is $N^2 K_{\text{Tmp}}^4 / B_\tau$-subgaussian. As a result, for each $\xi > 0$, we have

$$\mathbb{P}\left[\left|\partial_{a_s^{(k)}}^{(B_\tau)} l - \mathbb{E}\,\partial_{a_s^{(k)}} l\right| \geq \xi\right] \leq 2\exp\left(\frac{-\xi^2 B_\tau}{N^2}\right).$$

Apply union bound and we get

$$\mathbb{P}\left[\|\nabla_A l - \mathbb{E}\,\nabla_A l\|_\infty \geq \xi\right] \leq 2TN\exp\left(\frac{-\xi^2 B_\tau}{N^2}\right) = \exp\left(\log(2NT) - \frac{\xi^2 B_\tau}{N^2}\right).$$

Recall that the separation between the expectations is $c\alpha_V K_P/(NQ)$. Hence, it suffices to choose $\xi = c\alpha_V K_P/(2NQ)$. Then, to make the failure probability at most $\delta_{\text{Tmp}}$, we can choose $B_\tau$ as follows:

$$\exp\left(\log(2NT) - \frac{\xi^2 B_\tau}{N^2}\right) \leq \delta_\tau \quad \Leftarrow \quad B_\tau \geq C\frac{N^4 Q^2}{\alpha_V^2 K_P^2}\log\left(\frac{CNT}{\delta_\tau}\right),$$

for some large constant $C > 0$. Then, to boost the accuracy from $1/2$ to $\varepsilon_{\text{Tmp}}$, it suffices to increase the batch size to $C\frac{N^4 Q^2}{\varepsilon_{\text{Tmp}}^2 \alpha_V^2 K_P^2}\log\left(\frac{CNT}{\delta_\tau}\right)$. $\qquad\square$

Using this lemma, we can pick $\lambda = \Theta(\frac{\varepsilon K_P}{Q^2 N^3})$ to sparsify our proximal gradient. Notice that our proximal gradient is a biased estimate of the true preconditioned gradient, but the separation guarantees that it is possible to make the error controllable. The following lemma calculates the error each proximal step introduces. Here we define $\hat{h}_{A,\tau} = G_{\lambda,\tau}^{(k)} - \mathbb{E}\,\hat{\nabla}_A l$ instead of $h$ because of the bias introduced by the proximal gradient.

**Lemma E.4.** *Under the same setting of Lemma E.3, if the batch size*

$$B_\tau \geq \max\left\{C\frac{N^8 Q^4}{\varepsilon^2 \alpha_V^2 K_P^2}\log\left(\frac{2CNT}{\delta_\tau}\right), \frac{\Theta(N^6 Q^3)}{\delta \varepsilon^2}\right\},$$

*the noise attention score $|a_t^{(k)}| \leq O(1/T)$ for all $t \notin q^{(k)}$ and all $k \in [N]$, and $\lambda = \Theta\left(\frac{\varepsilon K_P}{Q^2 N^3}\right)$, then with probability $1 - \delta_\tau$, the gradient error at iteration $\tau$*

$$\|\hat{h}_{A,\tau}\|_\mu := \|G_{\lambda,\tau} - \mathbb{E}\,\hat{\nabla}_A l\|_\mu \leq O\left(\frac{\varepsilon}{QN^2}\right).$$

*Proof.* For notational simplicity, we drop the superscript $k$. The goal is to estimate the difference between $\mathbb{E}\,\hat{\nabla}_a l$ and $g_{\lambda,\tau}$ by calculating the magnitude of the bias of the proximal gradient together with the concentration error.

We consider the population gradient first. Since we have $\{a_{t,\tau}\}_{t \notin q}$ are all the same in Stage 2, we can write

$$a_\tau = [a_\tau]_q + \frac{(1 - b_\tau)\mathbf{1}_{q^c}}{T - Q} \quad \text{where} \quad b_\tau = \sum_{t \in q} a_t,$$

and recall the projected preconditioned gradient for $A$.

$$\mathbb{E}\,\hat{\nabla}_a l = \left(\|V\|_\mu^2 - \|\mu\|^2\right)\left(a^{(k)} - \frac{1}{T}\right) - \left(\langle V, P\rangle_\mu - \|\mu\|^2\right)\left(q^{(k)} - \frac{1}{T}\right).$$

Therefore, we have

$$\mathbb{E}\,\hat{\nabla}_a l = \left[\mathbb{E}\,\hat{\nabla}_a l\right]_q + K_V\frac{(1 - b_\tau)\mathbf{1}_{q^c}}{T - Q} + (\|V\|_\mu^2 - \langle V, P\rangle_\mu)\frac{\mathbf{1}_{q^c}}{T}.$$

Now, we consider $g$ by calculating the expression of $a_{\tau+1}$. First, note that by Lemma E.3, we have $a_{t,\tau+1}'' = \left(a_{t,\tau+1}' - \lambda\right)\mathbb{1}\{t \in q\}$. This implies that $\{a_{t,\tau+1}\}_{t \notin q}$ are all the same and the value is at most $1/T$. Moreover, it also implies that it suffices to focus on $[a_{\tau+1}'']_q$, for which we have

$$[a_{\tau+1}'']_q = [a_{\tau+1}']_q - \lambda\mathbf{1}_q = [a_\tau]_q - \frac{\eta}{\mu_k}\left[\nabla_a^{(B_\tau)} l\right]_q - \lambda\mathbf{1}_q.$$

Therefore, $\sum_{t \in q} a'_{t,\tau+1} = b_\tau - \frac{\eta}{\mu_k} \sum_{t \in q} \partial_{a_t}^{(B_\tau)} l - Q\lambda$ and

$$a_{\tau+1} = [a_\tau]_q - \frac{\eta}{\mu_k} \left[ \nabla_a^{(B_\tau)} l \right]_q - \lambda \mathbf{1}_q + \left( 1 - b_\tau + \frac{\eta}{\mu_k} \sum_{t \in q} \partial_{a_t}^{(B_\tau)} l + Q\lambda \right) \frac{1}{T}$$

$$= a_\tau - \frac{(1 - b_\tau)\mathbf{1}_{q^c}}{T - Q} - \frac{\eta}{\mu_k} \left[ \nabla_a^{(B_\tau)} l \right]_q - \lambda \mathbf{1}_q + \left( 1 - b_\tau + \frac{\eta}{\mu_k} \sum_{t \in q} \partial_{a_t}^{(B_\tau)} l + Q\lambda \right) \frac{1}{T}.$$

Thus, we can write an explicit update

$$g_{\lambda,\tau} = \frac{1}{\mu_k} \left[ \nabla_a^{(B_\tau)} l \right]_q + \frac{\lambda}{\eta} \mathbf{1}_q - \left( 1 - b_\tau + \frac{\eta}{\mu_k} \sum_{t \in q} \partial_{a_t}^{(B_\tau)} l + Q\lambda \right) \frac{1}{\eta T} + \frac{(1 - b_\tau)\mathbf{1}_{q^c}}{\eta(T - Q)}$$

$$= \left[ \hat{\nabla}_a^{(B_\tau)} l \right]_q + \frac{\mathbf{1}_q \mathbf{1}^\top \nabla_a^{(B_\tau)} l}{\mu_k T} - \left( 1 - b_\tau + \frac{\eta}{\mu_k} \sum_{t \in q} \partial_{a_t}^{(B_\tau)} l + Q\lambda \right) \frac{1}{\eta T}$$

$$+ \frac{(1 - b_\tau)\mathbf{1}_{q^c}}{\eta(T - Q)} + \frac{\lambda}{\eta} \mathbf{1}_q$$

Then the gradient error at step $\tau$ can be decomposed into:

$$g_{\lambda,\tau} - \mathbb{E}\,\hat{\nabla}_a l = \left[ \hat{\nabla}_a^{(B_\tau)} l - \mathbb{E}\,\hat{\nabla}_a l \right]_q \qquad\qquad \text{(Concentration error)}$$

$$+ \frac{\mathbf{1}_q \mathbf{1}^\top \nabla_a^{(B_\tau)} l}{\mu_k T} - \left( 1 - b_\tau + \frac{\eta}{\mu_k} \sum_{t \in q} \partial_{a_t}^{(B_\tau)} l + Q\lambda \right) \frac{1}{\eta T} + \frac{\lambda}{\eta} \mathbf{1}_q$$

$$+ \frac{(1 - b_\tau)\mathbf{1}_{q^c}}{\eta(T - Q)} - K_V \frac{(1 - b_\tau)\mathbf{1}_{q^c}}{T - Q} - (\|V\|_\mu^2 - \langle V, P \rangle_\mu) \frac{\mathbf{1}_{q^c}}{T} \quad \text{(Gradient bias error)}$$

Here the gradient bias error can be further simplified to

$$\left[ \frac{1}{\mu_k T} \sum_{t \notin q} \partial_{a_t}^{(B_\tau)} l - \frac{1 - b_\tau + Q\lambda}{\eta T} + \frac{\lambda}{\eta} \right] \mathbf{1}_q \qquad\qquad (1^*)$$

$$+ \left[ \frac{Q(1 - b_\tau - Q\lambda) - Q\lambda T - \eta T K_V}{\eta T(T - Q)} - \frac{\sum_{t \in q} \partial_{a_t}^{(B_\tau)} l}{\mu_k T} - \frac{\|V\|_\mu^2 - \langle V, P \rangle_\mu}{T} \right] \mathbf{1}_{q^c} \qquad (2^*)$$

First, we estimate the concentration error. Similar to Lemma B.10, we first upper bound the infinity norm of the gradient. Consider the maximum absolute value in the original gradients. For $\nabla_{a^{(k)}} l$, we have

$$\left\| \mathbb{1}\{x_{T+1} = k\}(VX)^\top (VX a^{(k)} - e_{x_o}) \right\|_\infty$$

$$\leq \left\| \mathbb{1}\{x_{T+1} = k\}(VX)^\top (VX a^{(k)}) \right\|_\infty + \left\| \mathbb{1}\{x_{T+1} = k\}(VX)^\top (e_{x_o}) \right\|_\infty$$

$$\leq \max_{s \in [T]} \left| \sum_{t=1}^{T} a_t (Ve_{x_s})^\top Ve_{x_t} \right| + \max_{s \in [T]} \left| (Ve_{x_s})^\top e_{x_o} \right|$$

The first term can be upper-bounded in the following way:

$$\max_{s \in [T]} \left| \sum_{t=1}^{T} a_t (Ve_{x_s})^\top Ve_{x_t} \right| \leq \left( \sum_{t \in q} |a_t| + \sum_{t \notin q} |a_t| \right) \max_{s,t} \left| (Ve_{x_s})^\top Ve_{x_t} \right|$$

$$\leq \Theta(1) \max_{s,t} \left| (Ve_{x_s})^\top Ve_{x_t} \right|$$

The second inequality is due to $|a_t^{(k)}| \leq O(1/T)$ for $t \notin q$. Since $V_{n,m} \leq O(1)$, we have $\max_{s,t} \left| (Ve_{x_s})^\top Ve_{x_t} \right|$ upper bounded by $O(1)$. Therefore

$$\max_{s \in [T]} \left| \sum_{t=1}^{T} a_t (Ve_{x_s})^\top Ve_{x_t} \right| \leq O(1)$$

And similarly, the second term $\max_{s \in [T]} \left| (\boldsymbol{V}\boldsymbol{e}_{x_s})^\top \boldsymbol{e}_{x_o} \right|$ can be bounded by $O(1)$ because the infinity norm of $\boldsymbol{V}$ is also upper bounded by $O(1)$. Therefore, we know $\|\nabla_{\boldsymbol{a}^{(k)}} l\|_\infty \leq O(1)$.

Now we consider the preconditioned gradient $\hat{\nabla}_{\boldsymbol{a}}^{(B_\tau)} l$:

$$\|\hat{\nabla}_{\boldsymbol{a}}^{(B_\tau)} l\|_\infty = \left\| \frac{1}{\mu_k} \left( \boldsymbol{I}_T - \frac{\mathbf{1}_T \mathbf{1}_T^\top}{T} \right) \left( \nabla_{\boldsymbol{a}^{(k)}}^{(B_\tau)} l \right) \right\|_\infty$$

$$\leq \left\| \frac{1}{\mu_k} \boldsymbol{I}_T \left( \nabla_{\boldsymbol{a}^{(k)}}^{(B_\tau)} l \right) \right\|_\infty + \left\| \frac{1}{\mu_k} \left( \frac{\mathbf{1}_T \mathbf{1}_T^\top}{T} \right) \left( \nabla_{\boldsymbol{a}^{(k)}}^{(B_\tau)} l \right) \right\|_\infty \leq O(N)$$

since $\mu_k \geq \frac{c}{N}$ for all $k \in [N]$. Now since $\left[ \hat{\nabla}_{\boldsymbol{a}}^{(B_\tau)} l \right]_{\boldsymbol{q}}$ is $Q$-sparse, we have $\mathbb{E} \left\| \left[ \hat{\nabla}_A l \right]_{\boldsymbol{q}} \right\|_\mu^2 \leq O(QN^2)$. By Lemma B.9, when $B_\tau \geq \frac{\Theta(N^6 Q^3)}{\delta \varepsilon^2}$, with probability $1 - \frac{\delta_\tau}{2}$,

$$\left\| \left[ \hat{\nabla}_{\boldsymbol{a}}^{(B_\tau)} l - \mathbb{E} \hat{\nabla}_{\boldsymbol{a}} l \right]_{\boldsymbol{q}} \right\| \leq O\left( \frac{\varepsilon}{QN^2} \right)$$

Then, consider the gradient bias term. With the selected $\lambda = \Theta\left( \frac{\varepsilon K_P}{Q^2 N^3} \right)$ and $\left\| \partial_{a_t}^{(B_\tau)} l \right\| \leq O\left( \frac{\varepsilon K_P}{N^3 Q^2} \right)$ for $t \notin \boldsymbol{q}$, with probability $1 - \delta_\tau/2$ we have the $\mu$-norm of first term (since $K_P \leq O(N)$):

$$\|(1^*)\| \leq \left( \left\| \frac{1}{\mu_k T} \sum_{t \notin \boldsymbol{q}} \partial_{a_t}^{(B_\tau)} l \right\| + \frac{Q\lambda}{T} + \lambda \right) \cdot \sqrt{Q}$$

$$\leq O\left( \frac{\varepsilon}{QN^2} \right) + O\left( \frac{\varepsilon}{TN^2 \sqrt{Q}} \right) + O\left( \frac{\varepsilon}{Q^{3/2} N^2} \right) \leq O\left( \frac{\varepsilon}{QN^2} \right).$$

and the second term can be upper-bounded by

$$\|(2^*)\| \leq \left( O\left( \frac{QN}{T} \right) + O\left( \frac{QN}{T} \right) + O\left( \frac{N}{T} \right) \right) \cdot \sqrt{T} \leq O\left( \frac{\varepsilon}{QN^2} \right).$$

since $\frac{1}{\sqrt{T}} \leq O\left( \frac{\varepsilon}{Q^2 N^3} \right)$, $\partial_{a_t}^{(B_\tau)} l = O(1)$, and $\|\boldsymbol{V}\|_\mu^2 \leq N$.

Combine all three terms and by union bound, we have with probability $1 - \delta_\tau$

$$\left\| \hat{\boldsymbol{h}}_{A,\tau} \right\| = \left\| \boldsymbol{G}_{\lambda,\tau}^{(k)} - \mathbb{E} \hat{\nabla}_A l \right\|_\mu$$

$$= \sqrt{\sum_{k=1}^{N} \mu_k \|\boldsymbol{g}_{\lambda,\tau}^{(k)} - \mathbb{E} \hat{\nabla}_{\boldsymbol{a}^{(k)}} l\|^2} \leq O\left( \frac{\varepsilon}{QN^2} \right)$$

since $\mu_k = \Theta(1/N)$. $\qquad\square$

## E.2 Model aligning and the decrease of the errors

First, we show that the signal will continue to grow and approximation error will decrease. This decouples the error at the end of Stage 1 and the final error. In particular, we show that eventually we will have $\alpha_V + \alpha_A \approx 2$, $\|\Delta_A\|_\mu^2 \approx 0$ and $\|\Delta_V\|_\mu^2 \approx 0$[5].

For notational simplicity, define $\delta_A^2 = \|\Delta_A\|_\mu^2 / K_Q$ and $\delta_V^2 = \|\Delta_V\|_\mu^2 / K_P$. Recall Lemma C.5 and Lemma C.6 and that we choose $\eta_V = \eta/K_Q$, $\eta_A = \eta/K_P$. The dynamics of the signals and the errors can be described using[6]

$$\alpha_{V,\tau+1} = \alpha_{V,\tau} + \eta \left( 1 - \alpha_A \alpha_V \right) \alpha_A + \frac{\eta}{K_Q} \frac{1 - \alpha_V}{T} - \eta \alpha_V \delta_A^2 - \eta \frac{\langle \boldsymbol{h}_{V,\tau}, \boldsymbol{P} \rangle_\mu}{K_P K_Q},$$

---

[5]However, we cannot ensure $\alpha_A \approx \alpha_V$ since $\alpha_A - \alpha_V$ is not contractive toward the end of training due to the magnitude of gradient noise. This issue can be fixed by a final rounding stage (See Appendix F).

[6]Here $\boldsymbol{h}_{A,\tau} \to \hat{\boldsymbol{h}}_{A,\tau}$ because each gradient step is changed to the $\ell_1$-regularized gradient. It does not change the main parts in the population process.

$$\alpha_{A,\tau+1} = \alpha_{A,\tau} + \eta \left(1 - \alpha_V \alpha_A\right) \alpha_V - \eta \alpha_A \delta_V^2 - \eta \frac{\langle \hat{h}_{A,\tau}, Q \rangle_\mu}{K_P K_Q},$$

and

$$\delta_{A,\tau+1}^2 = \left(1 - \eta \alpha_V^2 - \eta \delta_A^2\right)^2 \delta_A^2 - 2\eta \left(1 - \eta \alpha_V^2 - \eta \delta_V^2\right) \frac{\langle \Delta_A, \hat{h}_{A,\tau} \rangle_\mu}{K_P K_Q}$$

$$- \eta^2 \frac{\langle \hat{h}_{A,\tau}, Q \rangle_\mu^2}{K_P^2 K_Q^2} + \eta^2 \frac{\left\| \hat{h}_{A,\tau} \right\|_\mu^2}{K_P^2 K_Q},$$

$$\delta_{V,\tau+1}^2 = \left(1 - \eta \left(\alpha_A^2 + \frac{1}{K_Q T} + \delta_A^2\right)\right)^2 \delta_V^2 + 2\eta \left(1 - \eta \left(\alpha_A^2 + \frac{1}{K_Q T} + \delta_A^2\right)\right) \frac{\langle \Delta_V, h_V \rangle_\mu}{K_P K_Q}$$

$$- \eta^2 \frac{\langle P, h_V \rangle_\mu^2}{K_Q^2 K_P^2} + \eta^2 \frac{\| h_V \|_\mu^2}{K_P K_Q^2}.$$

### E.2.1 Lemmas for the dynamics

Before we come to the final convergence analysis, we first simplify the dynamics with some basic lemmas.

**Lemma E.5** (Dynamics of the errors). *For the errors, we have*

$$\delta_{A,\tau+1}^2 \le \exp\left(-2\eta \alpha_V^2\right) \delta_{A,\tau}^2 + 3\eta \left(\delta_A + \eta \frac{\left\| \hat{h}_A \right\|_\mu}{K_P \sqrt{K_Q}}\right) \frac{\left\| \hat{h}_A \right\|_\mu}{K_P \sqrt{K_Q}},$$

$$\delta_{V,\tau+1}^2 \le \exp\left(-2\eta \alpha_A^2\right) \delta_{V,\tau}^2 + 3\eta \left(\delta_V + \eta \frac{\| h_V \|_\mu}{\sqrt{K_P} K_Q}\right) \frac{\| h_V \|_\mu}{\sqrt{K_P} K_Q}.$$

*Proof.* First, we write

$$\delta_{A,\tau+1}^2 = \left(1 - \eta \alpha_V^2 - \eta \delta_A^2\right)^2 \delta_A^2 - 2\eta \left(1 - \eta \alpha_V^2 - \eta \delta_V^2\right) \frac{\langle \Delta_A, \hat{h}_A \rangle_\mu}{K_P K_Q}$$

$$- \eta^2 \frac{\langle \hat{h}_{A,\tau}, Q \rangle_\mu^2}{K_P^2 K_Q^2} + \eta^2 \frac{\left\| \hat{h}_A \right\|_\mu^2}{K_P^2 K_Q}$$

$$\le \left(1 - \eta \alpha_V^2 - \eta \delta_A^2\right)^2 \delta_A^2 + 2\eta \frac{\delta_A \left\| \hat{h}_A \right\|_\mu}{K_P \sqrt{K_Q}} + 3\eta^2 \frac{\left\| \hat{h}_{A,\tau} \right\|_\mu^2}{K_P^2 K_Q}$$

$$\le \left(1 - \eta \alpha_V^2 - \eta \delta_A^2\right)^2 \delta_A^2 + 3\eta \left(\delta_A + \frac{\eta}{K_P} \frac{\left\| \hat{h}_A \right\|_\mu}{\sqrt{K_Q}}\right) \frac{\left\| \hat{h}_A \right\|_\mu}{K_P \sqrt{K_Q}}.$$

For the first term, we have

$$\left(1 - \eta \alpha_V^2 - \eta \delta_A^2\right)^2 \le \exp\left(-\eta \alpha_V^2 - \eta \delta_A^2\right)^2 \le \exp\left(-2\eta \alpha_V^2\right).$$

Thus, for $\delta_A$, we have

$$\delta_{A,\tau+1}^2 \le \exp\left(-2\eta \alpha_V^2\right) \delta_{A,\tau}^2 + 3\eta \left(\delta_A + \eta \frac{\left\| \hat{h}_A \right\|_\mu}{K_P \sqrt{K_Q}}\right) \frac{\left\| \hat{h}_A \right\|_\mu}{K_P \sqrt{K_Q}}.$$

Similarly, for $\delta_V$, we have

$$\delta_{V,\tau+1}^2 \leq \left(1 - \eta\left(\alpha_A^2 + \frac{1}{K_Q T} + \delta_A^2\right)\right)^2 \delta_V^2 + 3\eta \frac{\delta_V \|\boldsymbol{h}_V\|_\mu}{\sqrt{K_P} K_Q} + 3\eta^2 \frac{\|\boldsymbol{h}_V\|_\mu^2}{K_P K_Q^2}$$

$$\leq \exp\left(-2\eta\alpha_A^2\right)\delta_V^2 + 3\eta\left(\delta_V + \eta \frac{\|\boldsymbol{h}_V\|_\mu}{\sqrt{K_P} K_Q}\right)\frac{\|\boldsymbol{h}_V\|_\mu}{\sqrt{K_P} K_Q}$$

$\square$

**Lemma E.6** (Dynamics of $\alpha_A - \alpha_V$). *The difference between the signals evolves as follows*

$$(\alpha_{V,\tau+1} - \alpha_{A,\tau+1})^2 \leq \exp\left(-2\eta\left(1 - \alpha_A\alpha_V\right)\right)(\alpha_V - \alpha_A)^2$$

$$+ 8\eta\left(\frac{1}{K_Q T} + \delta_A^2 + \delta_V^2 + \frac{\|\boldsymbol{h}_{V,\tau}\|_\mu}{\sqrt{K_P} K_Q} + \frac{\|\hat{\boldsymbol{h}}_{A,\tau}\|_\mu}{K_P \sqrt{K_Q}}\right).$$

*Proof.* First, we write

$$\alpha_{V,\tau+1} - \alpha_{A,\tau+1} = \alpha_{V,\tau} - \alpha_{A,\tau} - \eta\left(1 - \alpha_A\alpha_V\right)(\alpha_V - \alpha_A)$$

$$+ \frac{\eta}{K_Q}\frac{1 - \alpha_V}{T} - \eta\alpha_V\delta_A^2 - \eta\frac{\langle \boldsymbol{h}_{V,\tau}, \boldsymbol{P}\rangle_\mu}{K_P K_Q} + \eta\alpha_A\delta_V^2 + \eta\frac{\langle \hat{\boldsymbol{h}}_{A,\tau}, \boldsymbol{Q}\rangle_\mu}{K_P K_Q}$$

$$=: \left(1 - \eta\left(1 - \alpha_A\alpha_V\right)\right)(\alpha_V - \alpha_A) + \texttt{Tmp}.$$

Therefore, we have

$$(\alpha_{V,\tau+1} - \alpha_{A,\tau+1})^2 = \left(\left(1 - \eta\left(1 - \alpha_A\alpha_V\right)\right)(\alpha_V - \alpha_A) + \texttt{Tmp}\right)^2$$

$$= \left(1 - \eta\left(1 - \alpha_A\alpha_V\right)\right)^2 (\alpha_V - \alpha_A)^2$$

$$+ 2\left(1 - \eta\left(1 - \alpha_A\alpha_V\right)\right)(\alpha_V - \alpha_A)\texttt{Tmp} + \texttt{Tmp}^2$$

$$\leq \exp\left(-2\eta\left(1 - \alpha_A\alpha_V\right)\right)(\alpha_V - \alpha_A)^2$$

$$+ 3|\alpha_V - \alpha_A||\texttt{Tmp}| + \texttt{Tmp}^2.$$

Then, for $\texttt{Tmp}$, we compute

$$0.5\eta^{-1}|\texttt{Tmp}| \leq \frac{1}{K_Q T} + \delta_A^2 + \delta_V^2 + \frac{\|\boldsymbol{h}_{V,\tau}\|_\mu}{\sqrt{K_P} K_Q} + \frac{\|\hat{\boldsymbol{h}}_{A,\tau}\|_\mu}{K_P \sqrt{K_Q}}.$$

In particular, this implies $|\texttt{Tmp}| \leq 1$. Thus, we have

$$(\alpha_{V,\tau+1} - \alpha_{A,\tau+1})^2 \leq \exp\left(-2\eta\left(1 - \alpha_A\alpha_V\right)\right)(\alpha_V - \alpha_A)^2 + 4|\texttt{Tmp}|$$

$$\leq \exp\left(-2\eta\left(1 - \alpha_A\alpha_V\right)\right)(\alpha_V - \alpha_A)^2$$

$$+ 8\eta\left(\frac{1}{K_Q T} + \delta_A^2 + \delta_V^2 + \frac{\|\boldsymbol{h}_{V,\tau}\|_\mu}{\sqrt{K_P} K_Q} + \frac{\|\hat{\boldsymbol{h}}_{A,\tau}\|_\mu}{K_P \sqrt{K_Q}}\right).$$

$\square$

**Lemma E.7.** *Suppose that both $\alpha_V, \alpha_A$ are at most 1. Then, we have*

$$1 - \frac{\alpha_{V,\tau+1} + \alpha_{A,\tau+1}}{2} \leq \left(1 - \eta\frac{\alpha_V + \alpha_A}{2}\right)\left(1 - \frac{\alpha_V + \alpha_A}{2}\right) + \eta\frac{\alpha_V + \alpha_A}{2}\left(\delta_A^2 + \delta_V^2\right)$$

$$+ \frac{\eta}{2}\frac{\|\boldsymbol{h}_{V,\tau}\|_\mu}{\sqrt{K_P} K_Q} + \frac{\eta}{2}\frac{\|\hat{\boldsymbol{h}}_{A,\tau}\|_\mu}{K_P \sqrt{K_Q}}.$$

*Proof.* First, we write

$$\alpha_{V,\tau+1} + \alpha_{A,\tau+1} = \alpha_{V,\tau} + \alpha_{A,\tau} + \eta\left(1 - \alpha_A\alpha_V\right)\left(\alpha_V + \alpha_A\right)$$

$$+ \frac{\eta}{K_Q}\frac{1-\alpha_V}{T} - \eta\alpha_V\delta_A^2 - \eta\alpha_A\delta_V^2 - \eta\frac{\left\langle h_{V,\tau}, P\right\rangle_\mu}{K_P K_Q} - \eta\frac{\left\langle \hat{h}_{A,\tau}, Q\right\rangle_\mu}{K_P K_Q}$$

$$=: \alpha_{V,\tau} + \alpha_{A,\tau} + \eta\mathrm{Tmp}_{\mathrm{sig}} + \eta\mathrm{Tmp}_{\mathrm{err}}.$$

For the signal growth, note that $\alpha_A\alpha_V \le (\alpha_V^2 + \alpha_A^2)/2 \le (\alpha_V + \alpha_A)/2$. Hence,

$$\left(1 - \alpha_A\alpha_V\right)\left(\alpha_V + \alpha_A\right) \ge \frac{1}{2}\left(\alpha_V + \alpha_A\right)\left(2 - \alpha_V - \alpha_A\right).$$

For the error terms, we have

$$\left|\frac{\left\langle h_{V,\tau}, P\right\rangle_\mu}{K_P K_Q} + \frac{\left\langle \hat{h}_{A,\tau}, Q\right\rangle_\mu}{K_P K_Q}\right| \le \frac{\left\|h_{V,\tau}\right\|_\mu}{\sqrt{K_P}K_Q} + \frac{\left\|\hat{h}_{A,\tau}\right\|_\mu}{K_P\sqrt{K_Q}}.$$

Thus, we have

$$\alpha_{V,\tau+1} + \alpha_{A,\tau+1} \ge \alpha_{V,\tau} + \alpha_{A,\tau} + \eta\left(\alpha_V + \alpha_A\right)\left(1 - \frac{\alpha_V + \alpha_A}{2} - \delta_A^2 - \delta_V^2\right)$$

$$- \eta\frac{\left\|h_{V,\tau}\right\|_\mu}{\sqrt{K_P}K_Q} - \eta\frac{\left\|\hat{h}_{A,\tau}\right\|_\mu}{K_P\sqrt{K_Q}},$$

and, therefore,

$$1 - \frac{\alpha_{V,\tau+1} + \alpha_{A,\tau+1}}{2} \le \left(1 - \eta\frac{\alpha_V + \alpha_A}{2}\right)\left(1 - \frac{\alpha_V + \alpha_A}{2}\right) + \eta\frac{\alpha_V + \alpha_A}{2}\left(\delta_A^2 + \delta_V^2\right)$$

$$+ \frac{\eta}{2}\frac{\left\|h_{V,\tau}\right\|_\mu}{\sqrt{K_P}K_Q} + \frac{\eta}{2}\frac{\left\|\hat{h}_{A,\tau}\right\|_\mu}{K_P\sqrt{K_Q}}.$$

$\square$

**Lemma E.8.** *Put $\varepsilon_A = 1 - \alpha_A$ and $\varepsilon_V = 1 - \alpha_V$. When $|\varepsilon_A|, |\varepsilon_V| \le 1/2$, we have*

$$\left(\varepsilon_{A,\tau+1} + \varepsilon_{V,\tau+1}\right)^2 \le \exp\left(-2\eta\right)\left(\varepsilon_A + \varepsilon_V\right)^2 + 8\eta|\varepsilon_A + \varepsilon_V|\varepsilon_A\varepsilon_V + 16\eta^2\varepsilon_A^2\varepsilon_V^2$$

$$+ 8\eta|\varepsilon_A + \varepsilon_V|\left(\frac{1}{K_Q T} + \delta_A^2 + \delta_V^2 + \frac{\left\|h_{V,\tau}\right\|_\mu}{\sqrt{K_P}K_Q} + \frac{\left\|\hat{h}_{A,\tau}\right\|_\mu}{K_P\sqrt{K_Q}}\right).$$

*Proof.* Similar to the proof of the previous lemma, we write

$$\alpha_{V,\tau+1} + \alpha_{A,\tau+1} = \alpha_{V,\tau} + \alpha_{A,\tau} + \eta\left(1 - \alpha_A\alpha_V\right)\left(\alpha_V + \alpha_A\right)$$

$$+ \frac{\eta}{K_Q}\frac{1-\alpha_V}{T} - \eta\alpha_V\delta_A^2 - \eta\alpha_A\delta_V^2 - \eta\frac{\left\langle h_{V,\tau}, P\right\rangle_\mu}{K_P K_Q} - \eta\frac{\left\langle \hat{h}_{A,\tau}, Q\right\rangle_\mu}{K_P K_Q}$$

$$=: \alpha_{V,\tau} + \alpha_{A,\tau} + \eta\mathrm{Tmp}_{\mathrm{sig}} + \eta\mathrm{Tmp}_{\mathrm{err}}.$$

For the signal term, we have

$$\mathrm{Tmp}_{\mathrm{sig}} = \left(1 - (1 - \varepsilon_A)(1 - \varepsilon_V)\right)\left(2 - \varepsilon_A - \varepsilon_V\right)$$

$$= \left(2 - \varepsilon_A - \varepsilon_V\right)\left(\varepsilon_A + \varepsilon_V\right) - \varepsilon_A\varepsilon_V\left(2 - \varepsilon_A - \varepsilon_V\right).$$

For the error term, we have

$$|\mathrm{Tmp}_{\mathrm{err}}| \le \frac{1}{K_Q T} + 2\delta_A^2 + 2\delta_V^2 + \frac{2\left\|h_{V,\tau}\right\|_\mu}{\sqrt{K_P}K_Q} + \frac{2\left\|\hat{h}_{A,\tau}\right\|_\mu}{K_P\sqrt{K_Q}}.$$

Combine these together, and we obtain

$$\varepsilon_{A,\tau+1} + \varepsilon_{V,\tau+1} = \varepsilon_{A,\tau} + \varepsilon_{V,\tau} - \eta \left(2 - \varepsilon_A - \varepsilon_V\right) \left(\varepsilon_A + \varepsilon_V\right)$$

$$+ \eta \varepsilon_A \varepsilon_V \left(2 - \varepsilon_A - \varepsilon_V\right) \pm 2\eta \left(\frac{1}{K_Q T} + \delta_A^2 + \delta_V^2 + \frac{\left\|\boldsymbol{h}_{V,\tau}\right\|_\mu}{\sqrt{K_P} K_Q} + \frac{\left\|\hat{\boldsymbol{h}}_{A,\tau}\right\|_\mu}{K_P \sqrt{K_Q}}\right)$$

$$=: \left(1 - \eta \left(2 - \varepsilon_A - \varepsilon_V\right)\right) \left(\varepsilon_A + \varepsilon_V\right) + \mathtt{Tmp}.$$

Thus,

$$\left(\varepsilon_{A,\tau+1} + \varepsilon_{V,\tau+1}\right)^2 = \left(\left(1 - \eta \left(2 - \varepsilon_A - \varepsilon_V\right)\right) \left(\varepsilon_A + \varepsilon_V\right) + \mathtt{Tmp}\right)^2$$

$$\leq \exp\left(-2\eta \left(2 - \varepsilon_A - \varepsilon_V\right)\right) \left(\varepsilon_A + \varepsilon_V\right)^2 + 2|\varepsilon_A + \varepsilon_V||\mathtt{Tmp}| + \mathtt{Tmp}^2.$$

Note that

$$|\mathtt{Tmp}| \leq 4\eta \left(\varepsilon_A \varepsilon_V + \frac{1}{K_Q T} + \delta_A^2 + \delta_V^2 + \frac{\left\|\boldsymbol{h}_{V,\tau}\right\|_\mu}{\sqrt{K_P} K_Q} + \frac{\left\|\hat{\boldsymbol{h}}_{A,\tau}\right\|_\mu}{K_P \sqrt{K_Q}}\right).$$

Recall $|\varepsilon_A|, |\varepsilon_V| \leq 1/2$. Thus,

$$\left(\varepsilon_{A,\tau+1} + \varepsilon_{V,\tau+1}\right)^2 \leq \exp\left(-2\eta\right) \left(\varepsilon_A + \varepsilon_V\right)^2$$

$$+ 8\eta|\varepsilon_A + \varepsilon_V| \left(\varepsilon_A \varepsilon_V + \frac{1}{K_Q T} + \delta_A^2 + \delta_V^2 + \frac{\left\|\boldsymbol{h}_{V,\tau}\right\|_\mu}{\sqrt{K_P} K_Q} + \frac{\left\|\boldsymbol{h}_{A,\tau}\right\|_\mu}{K_P \sqrt{K_Q}}\right)$$

$$+ 16\eta^2 \left(\varepsilon_A^2 \varepsilon_V^2 + \left(\frac{1}{K_Q T} + \delta_A^2 + \delta_V^2 + \frac{\left\|\boldsymbol{h}_{V,\tau}\right\|_\mu}{\sqrt{K_P} K_Q} + \frac{\left\|\boldsymbol{h}_{A,\tau}\right\|_\mu}{K_P \sqrt{K_Q}}\right)^2\right)$$

$$\leq \exp\left(-2\eta\right) \left(\varepsilon_A + \varepsilon_V\right)^2 + 8\eta|\varepsilon_A + \varepsilon_V|\varepsilon_A \varepsilon_V + 16\eta^2 \varepsilon_A^2 \varepsilon_V^2$$

$$+ 8\eta|\varepsilon_A + \varepsilon_V| \left(\frac{1}{K_Q T} + \delta_A^2 + \delta_V^2 + \frac{\left\|\boldsymbol{h}_{V,\tau}\right\|_\mu}{\sqrt{K_P} K_Q} + \frac{\left\|\boldsymbol{h}_{A,\tau}\right\|_\mu}{K_P \sqrt{K_Q}}\right).$$

$\square$

**Lemma E.9.** *Suppose that $(X_\tau)_\tau$ satisfies $X_{\tau+1} \leq e^{-A} X_\tau + B$ for some $A \in (0, 1]$, $B \geq 0$. If $X_0 \leq 2B/A$, then we have $X_\tau \leq 3B/A$ for all $\tau \geq 0$. If $X_0 \geq 2B/A$, then we have $X_\tau \leq 3B/A$ for all $\tau \geq \frac{6}{A} \log\left(\frac{X_0 A}{3B}\right)$.*

*Proof.* Since $e^{-A} \leq 1 - A/2$ for $A \in (0, 1]$, we have $X_{\tau+1} \leq X_\tau - AX_\tau/2 + B$. Hence, whenever $X_\tau \geq 2B/A$, we will have $X_{\tau+1} \leq X_\tau$. Moreover, if $X_\tau < 2B/A$, we have $X_{\tau+1} \leq 2B/A + B \leq 3B/A$. This proves the first part of the lemma.

Now, suppose that $X_0 \geq 2B/A$. When $X_\tau \geq 3B/A$, we have

$$X_{\tau+1} \leq X_\tau - AX_\tau/2 + B \leq X_\tau - AX_\tau/6 \leq e^{-A/6} X_\tau.$$

Thus, it takes at most $\frac{6}{A} \log\left(\frac{X_0 A}{3B}\right)$ steps to reduce $X_\tau$ from $X_0$ to $3B/A$. After that, the previous analysis applies. $\square$

### E.2.2 Main lemma of Stage 2

We split the analysis of Stage 2 into two substage. Let $c_\alpha \in (0, 0.05)$ be a small constant. Define

$$\mathcal{T}_{2.1} := \inf\left\{\tau \geq \mathcal{T}_1 : \left(\alpha_{V,\tau} + \alpha_{A,\tau}\right)/2 \geq 1 - c_\alpha\right\}.$$

We call $\{\tau : \tau \leq \mathcal{T}_{2.1}\}$ stage 2.1 and $\{\tau : \tau \geq \mathcal{T}_{2.1}\}$ stage 2.2. For notational simplicity, we define

$$\mathtt{Err}_{\text{small}} = \max_\tau \left\{\frac{1}{K_Q T} + \frac{\left\|\boldsymbol{h}_{V,\tau}\right\|_\mu}{\sqrt{K_P} K_Q} + \frac{\left\|\hat{\boldsymbol{h}}_{A,\tau}\right\|_\mu}{K_P \sqrt{K_Q}}\right\}.$$

Note that this can be made essentially arbitrarily small by choosing a large enough batch size.

**Lemma E.10.** *Suppose that the following hold at the beginning of Stage 2 (after thresholding and projection):*

*(a)* $\alpha_V, \alpha_A \geq \alpha^{(2)}$

*(b)* $\delta_A^2 + \delta_V^2 \leq (\delta^{(2)})^2.$

*Let $\delta_*$ be our target value for $\delta_A$ and $\delta_V$. Choose $\lambda$ as in Lemma E.4. Suppose that*

$$\max\left\{\delta^{(2)}, \text{Err}_{\text{small}}\right\} \text{Err}_{\text{small}} \leq \delta_*^2, \quad \text{Err}_{\text{small}} \leq O(\alpha^{(2)}),$$

$$\max\left\{(\alpha_{V,\mathcal{T}_1} - \alpha_{A,\mathcal{T}_1})^2, \delta^{(2)}, \text{Err}_{\text{small}}\right\} \leq O\left(\frac{1}{\log(1/\delta_*)}\right).$$

*Then, within $O(1/(\eta\alpha^{(2)}) + \log(1/\delta_*)/\eta)$ steps, we will have $\delta_A^2, \delta_V^2 \leq \delta_*^2$ and $\alpha_V, \alpha_A \in (0.9, 1.1)$.*

**Remark.** Note that our conditions on $\alpha_V, \alpha_A$ and $\delta^{(2)}$ are much weaker that what one can obtain from Stage 1. This allows us to apply the analysis here to transfer learning. ♣

The following proof should be treated as a large induction argument though we do not explicitly write down the induction as in the proof of Stage 1. In particular, we will show (by induction) that the approximation errors $\delta_A$ and $\delta_V$ are small, so that most of the naïve bounds on the entries of $\tilde{A}$ and $\tilde{V}$ can be transferred to $A$ and $V$. In particular, $|V_{n,m}| = O(1)$ for all $n, m \in [N]$ so that our bounds in Section E.1 are valid, and $|a_t^{(k)}| = O(1)$ for all $k \in [N], t \in [N]$, which implies that after the projection step, $a_t^{(k)} = O(1/T)$ for all $t \notin q^{(k)}$.

**Proof of the Lemma E.10**

**Common results for Stage 2.1 and 2.2** First, we prove some basic results that hold for both Stage 2.1 and 2.2. First, we show (by induction) that $\alpha_V, \alpha_A \geq \alpha^{(2)}$ and $\delta_V^2 + \delta_A^2 \leq \delta^{(2)}$ hold throughout Stage 2. Recall from Lemma E.5 that

$$\delta_{A,\tau+1}^2 \leq \exp\left(-2\eta\alpha_V^2\right)\delta_{A,\tau}^2 + 3\eta\left(\delta_A + \eta\frac{\left\|\hat{h}_A\right\|_\mu}{K_P\sqrt{K_Q}}\right)\frac{\left\|\hat{h}_A\right\|_\mu}{K_P\sqrt{K_Q}}$$

$$\leq \exp\left(-2\eta(\alpha^{(2)})^2\right)\delta_{A,\tau}^2 + \eta\text{Err}_{\text{small}}.$$

Hence, as long as $\text{Err}_{\text{small}} \leq O(1)\delta^{(2)}/(\alpha^{(2)})^2$, we can ensure $\delta_{A,\tau+1}^2 + \delta_{V,\tau+1}^2 \leq (\delta^{(2)})^2$ always hold.

**Stage 2.1: signal growth** By Lemma E.7, we have

$$1 - \frac{\alpha_{V,\tau+1} + \alpha_{A,\tau+1}}{2} \leq \left(1 - \eta\frac{\alpha_V + \alpha_A}{2}\right)\left(1 - \frac{\alpha_V + \alpha_A}{2}\right) + \eta\frac{\alpha_V + \alpha_A}{2}\left(\delta_A^2 + \delta_V^2\right) + \eta\text{Err}_{\text{small}}.$$

When $(\alpha_V + \alpha_A)/2 \leq 1 - c_\alpha$ and $\alpha_V, \alpha_A \geq \alpha^{(2)}$, we have

$$\eta\frac{\alpha_V + \alpha_A}{2}\left(1 - \frac{\alpha_V + \alpha_A}{2}\right) \geq \eta\frac{\alpha_V + \alpha_A}{2}c_\alpha \geq \eta\alpha^{(2)}c_\alpha.$$

Hence, as long as $\text{Err}_{\text{small}} \leq \alpha^{(2)}c_\alpha/2$ and $\delta^{(2)} \leq c_\alpha/2$, we have

$$1 - \frac{\alpha_{V,\tau+1} + \alpha_{A,\tau+1}}{2} \leq \left(1 - \frac{\eta}{2}\frac{\alpha_V + \alpha_A}{2}\right)\left(1 - \frac{\alpha_V + \alpha_A}{2}\right) \leq \exp\left(-\frac{\eta\alpha^{(2)}}{2}\right)\left(1 - \frac{\alpha_V + \alpha_A}{2}\right).$$

Thus, stage 2.1 takes at most $O(1/(\eta\alpha^{(2)}))$ steps.

**Stage 2.1: difference between the $\alpha$'s**   By Lemma E.6, we have

$$(\alpha_{V,\tau+1} - \alpha_{A,\tau+1})^2 \leq \exp\left(-2\eta\left(1 - \alpha_A\alpha_V\right)\right)(\alpha_V - \alpha_A)^2 + 8\eta\left((\delta^{(2)})^2 + \mathrm{Err}_{\mathrm{small}}\right).$$

By the AM-GM inequality, we have $\alpha_A\alpha_V \leq ((\alpha_A + \alpha_V)/2)^2 \leq (1 - c_\alpha)^2$. Therefore, $1 - \alpha_A\alpha_V \geq c_\alpha$ and the above inequality can be further rewritten as

$$(\alpha_{V,\tau+1} - \alpha_{A,\tau+1})^2 \leq \exp\left(-2c_\alpha\eta\right)(\alpha_V - \alpha_A)^2 + 8\eta\left((\delta^{(2)})^2 + \mathrm{Err}_{\mathrm{small}}\right).$$

Thus, by (the proof of) Lemma E.9, we have

$$(\alpha_{V,\tau} - \alpha_{A,\tau})^2 \leq \max\left\{2(\alpha_{V,\mathcal{T}_1} - \alpha_{A,\mathcal{T}_2})^2, \frac{4\left((\delta^{(2)})^2 + \mathrm{Err}_{\mathrm{small}}\right)}{c_\alpha}\right\}.$$

**Stage 2.2: error decrease**   By Lemma E.5, we have

$$\delta_{A,\tau+1}^2 \leq \exp\left(-2\eta\alpha_V^2\right)\delta_{A,\tau}^2 + 3\eta\max\left\{\delta^{(2)}, \mathrm{Err}_{\mathrm{small}}\right\}\mathrm{Err}_{\mathrm{small}}$$

$$\leq \exp\left(-\eta\right)\delta_{A,\tau}^2 + 3\eta\max\left\{\delta^{(2)}, \mathrm{Err}_{\mathrm{small}}\right\}\mathrm{Err}_{\mathrm{small}}.$$

Thus, by Lemma E.9, we have

$$\delta_{A,\tau}^2 \leq O(1)\max\left\{\delta^{(2)}, \mathrm{Err}_{\mathrm{small}}\right\}\mathrm{Err}_{\mathrm{small}} \leq \delta_*^2, \qquad \forall\tau \geq O\left(\frac{\log(1/\delta_*)}{\eta}\right).$$

In other words, Stage 2.2 takes at most $O\left(\frac{\log(1/\delta_*)}{\eta}\right)$ steps.

**Stage 2.2: stability of $\alpha_V \pm \alpha_A$**   Recall from Lemma E.8 and Lemma E.6 that

$$(\varepsilon_{A,\tau+1} + \varepsilon_{V,\tau+1})^2 \leq \exp\left(-2\eta\right)(\varepsilon_A + \varepsilon_V)^2 + 8\eta|\varepsilon_A + \varepsilon_V|\varepsilon_A\varepsilon_V + 16\eta^2\varepsilon_A^2\varepsilon_V^2$$

$$+ 8\eta|\varepsilon_A + \varepsilon_V|\left(\delta^{(2)} + \mathrm{Err}_{\mathrm{small}}\right),$$

$$(\alpha_{V,\tau+1} - \alpha_{A,\tau+1})^2 \leq \exp\left(-2\eta\left(1 - \alpha_A\alpha_V\right)\right)(\alpha_V - \alpha_A)^2 + 8\eta\left(\delta^{(2)} + \mathrm{Err}_{\mathrm{small}}\right).$$

We wish to maintain the induction hypotheses $|\alpha_V - \alpha_A| = |\varepsilon_V - \varepsilon_A| = \theta_-$ for some $\theta_- = o(1)$ and $(\alpha_A + \alpha_V)/2 \leq 1 + c/\log(1/\delta_*) =: 1 + \theta_+$.

First, assume these conditions are true. Then, we have

$$(\alpha_{V,\tau+1} - \alpha_{A,\tau+1})^2 \leq \exp\left(O(1)\eta/\log(1/\delta_*)\right)(\alpha_V - \alpha_A)^2 + 8\eta\left(\delta^{(2)} + \mathrm{Err}_{\mathrm{small}}\right).$$

Since Stage 2.2 takes at most $O(\log(1/\delta_*)/\eta)$ steps, when the constant $c$ is small, we have

$$(\alpha_{V,\tau} - \alpha_{A,\tau})^2 \leq 2\left((\alpha_{V,\mathcal{T}_{2.1}} - \alpha_{A,\mathcal{T}_{2.1}})^2, +8\left(\delta^{(2)} + \mathrm{Err}_{\mathrm{small}}\right)\right)$$

$$\leq O(1)\max\left\{(\alpha_{V,\mathcal{T}_{2.1}} - \alpha_{A,\mathcal{T}_{2.1}})^2, \delta^{(2)}, \mathrm{Err}_{\mathrm{small}}\right\} =: \theta_-.$$

We can choose the parameters appropriately so that $\theta_- < o(\theta_+)$. Then, when $\varepsilon_A + \varepsilon_V \in (-2\theta_+, -\theta_+)$, we have, for some large universal constant $C > 0$,

$$(\varepsilon_{A,\tau+1} + \varepsilon_{V,\tau+1})^2 \leq \exp\left(-2\eta\right)\theta_+^2 + C\eta\theta_+^3 + C\eta^2\theta_+^4 + C\eta\theta_+\left(\delta^{(2)} + \mathrm{Err}_{\mathrm{small}}\right)$$

$$\leq \exp\left(-2\eta\right)\theta_+^2 + C\eta\theta_+^3 + C\eta\theta_+\theta_-$$

$$\leq \theta_+^2.$$

This establishes the induction hypotheses on $\alpha_V \pm \alpha_A$.

# F   Stage 3: Final Convergence

As mentioned last section, we cannot ensure $\alpha_A \approx \alpha_V$ since $\alpha_A - \alpha_V$ is not contractive toward the end of training. However, we can add a final rounding step and then continue train the model to recover the ground-truth with $\varepsilon$-error.

First, we formally define our rounding procedure. Let $c > 0$ be a small constant (cf. the proof of Lemma F.1). For each $k \in [N]$, define

$$\hat{a}^{(k)} := \left[ a_t^{(k)} \mathbb{1}\left\{ a_t^{(k)} \geq c/Q \right\} \right]_{t \in [T]} \quad \text{and} \quad a^{(*,k)} := \frac{\hat{a}^{(k)}}{\mathbf{1}^\top \hat{a}^{(k)}}.$$

This $A^* := [a^{(*,k)}]_{k \in [N]}$ is our rounded version of $A$. For the error between $A^*$ and $Q$, we have the following lemma.

**Lemma F.1** (Rounding $A$). *Let $\varepsilon \in \left( \Theta(Q^2/T^2), 0.1 \right)$ be our target accuracy. Suppose that $\alpha_A = 1 - \varepsilon_A$ for some $|\varepsilon_A| \leq 0.1$ and $\|\Delta_A\|_\mu^2 \leq \delta_A^2$ for some $0 < \delta_A \ll 1/(Q\sqrt{N})$. Then, after rounding, we have*

$$\|A_* - Q\|_\mu^2 \leq O\left( \frac{Q^2}{T^2} + \delta_A^2 NQ \right).$$

*In particular, to achieve $\varepsilon$ accuracy in terms of $\|\cdot\|_\mu$, we only need $|\varepsilon_A| \leq 0.1$ and $\delta_A^2 \leq O(\varepsilon/\sqrt{NQ})$.*

**Remark**. In particular, this lemma implies that as long as $\varepsilon_A$ is not too large, after rounding, the error depends solely on $\|\Delta_A\|_\mu^2$. ♣

*Proof.* For notational simplicity, we omit the superscript $k$ for now. Write

$$a = \tilde{a} + \Delta_a = \alpha_V q + (1 - \alpha_V)\frac{\mathbf{1}}{T} + \Delta_a = (1 - \varepsilon_A)q + \varepsilon_A \frac{\mathbf{1}}{T} + \Delta_a.$$

Note that $\|\Delta_a\|_\infty \leq \|\Delta_a\|_2 \leq O(1)\sqrt{N}\|\Delta_A\|_\mu \leq O(\sqrt{N}\delta_A)$. Since all nonzero $q_s$ are lower bounded by $\Omega(1/Q)$, we have, for any $s \in q$ and $t \notin q$,

$$a_s \geq \frac{\Omega(1)}{Q} - \frac{|\varepsilon_A|}{T} - O\left( \sqrt{N}\delta_A \right) = \frac{\Omega(1)}{Q},$$

$$|a_t| \leq \frac{|\varepsilon_A|}{T} + O\left( \sqrt{N}\delta_A \right) \ll \frac{1}{Q}.$$

Hence, we can choose a small constant $c > 0$, so that

$$\hat{a} := [a_t \mathbb{1}\{|a_t| \geq c/Q\}]_{t \in [T]} = (1 - \varepsilon_A)q + \varepsilon_A \frac{\mathbf{1}_q}{T} + [\Delta_a]_q,$$

where for $v \in \mathbb{R}^T$, $v_q \in \mathbb{R}^T$ is defined as $[v_t \mathbb{1}\{t \in q\}]_{t \in [T]}$ here. Now, consider the difference between $q$ and $\hat{a}/\mathbf{1}^\top \hat{a}$. We have

$$\mathbf{1}^\top \hat{a} = 1 - \varepsilon_A + \frac{\varepsilon_A Q}{T} \pm O\left( Q\sqrt{N}\delta_A \right),$$

and therefore,

$$a_* := \frac{\hat{a}}{\mathbf{1}^\top \hat{a}} = \frac{(1 - \varepsilon_A)q + \varepsilon_A \mathbf{1}_q/T + [\Delta_a]_q}{(1 - \varepsilon_A) + \varepsilon_A Q/T \pm O\left( Q\sqrt{N}\delta_A \right)}$$

$$= \frac{(1 - \varepsilon_A)q + \varepsilon_A \mathbf{1}_q/T + [\Delta_a]_q}{(1 - \varepsilon_A)} \left( 1 \pm O\left( \varepsilon_A Q/T + Q\sqrt{N}\delta_A \right) \right)$$

$$= \left( q \pm O\left( \varepsilon_A \mathbf{1}_q/T + [\Delta_a]_q \right) \right) \left( 1 \pm O\left( \varepsilon_A Q/T + Q\sqrt{N}\delta_A \right) \right)$$

$$= q \pm O_2 \left( \varepsilon_A Q/T + \sqrt{QN}\delta_A \right).$$

Thus,

$$\|A_* - Q\|_\mu^2 = \sum_{k=1}^N \mu_k \left\| a_*^{(k)} - q^{(k)} \right\|^2 = O\left( \frac{\varepsilon_A^2 Q^2}{T^2} + \delta_A^2 NQ \right).$$

$\square$

**Lemma F.2.** *Let $\varepsilon \in \left(\Theta(1)/(K_Q T), 0.1\right)$ be our target accuracy. Suppose that $\|A - Q\|_\mu \leq \delta_{A,*} \leq 0.01\varepsilon$ and $\|\Delta_V\|_\mu \leq \delta_V \leq 0.01$ at the beginning of Stage 3, and $\|h_V\|_\mu \leq c\varepsilon\sqrt{K_P}K_Q$ for all $\tau$ and a sufficiently small constant $c$. Then, we have $\|V - P\|_\mu^2 \leq \varepsilon$ for all $\tau \geq \mathcal{T}_2 + \Theta(\log(1/\varepsilon)/\eta)$.*

*Proof.* Under the condition $\|A - Q\|_\mu \leq \delta_{A,*}$, we have $\|\Delta_A\|_\mu \leq \|A - Q\|_\mu \leq \delta_{A,*}$ and

$$\alpha_A = \frac{1}{K_Q}\left(\langle A, Q\rangle_\mu - \frac{1}{T}\right) = \frac{1}{K_Q}\left(K_Q + \langle \Delta_A, Q\rangle_\mu\right) = 1 \pm O\left(\sqrt{Q}\delta_{A,*}\right).$$

Recall that we only train $V$ in Stage 3. Note that by Lemma C.3,

$$\|V - P\|_\mu^2 = \left\|\tilde{V} - P\right\|_\mu^2 + \|\Delta_V\|_\mu^2 = (1 - \alpha_V)^2 K_P + \|\Delta_V\|_\mu^2 \leq (1 - \alpha_V)^2 + \|\Delta_V\|_\mu^2.$$

Hence, to get $\varepsilon$ accuracy, it suffices to have $(1 - \alpha_V)^2 \leq \varepsilon/2$ and $\|\Delta_V\|_\mu^2 \leq \varepsilon/2$.

First, for $\alpha_V$, by Lemma C.5 and $\eta_V = \eta/K_Q$, we have

$$\alpha_{V,\tau+1} = \alpha_{V,\tau} + \eta_V K_Q \left(1 - \alpha_V\alpha_A\right)\alpha_V + \eta_V\frac{1 - \alpha_V}{T} - \eta_V\alpha_V\|\Delta_A\|_\mu^2 - \frac{\eta_V}{K_P}\left\langle h_{V,\tau}, P\right\rangle_\mu$$

$$= \alpha_{V,\tau} + \eta\left(1 - \alpha_V\right)\alpha_V \pm \eta O\left(\frac{1}{K_Q T} + \delta_{A,*}\alpha_V\right) \pm \eta O\left(\frac{\left\langle h_{V,\tau}, P\right\rangle_\mu}{K_P K_Q}\right).$$

Hence,

$$\left(1 - \alpha_{V,\tau+1}\right)^2 = \left(\left(1 - \eta\alpha_V\right)\left(1 - \alpha_{V,\tau}\right) \pm \eta O\left(\frac{1}{K_Q T} + \delta_{A,*}\alpha_V\right) \pm \eta O\left(\frac{\left\langle h_{V,\tau}, P\right\rangle_\mu}{K_P K_Q}\right)\right)^2$$

$$\leq \exp\left(-2\eta\alpha_V\right)\left(1 - \alpha_{V,\tau}\right)^2 + \eta O\left(\frac{1}{K_Q T} + \delta_{A,*}\alpha_V\right) + \eta O\left(\frac{\left\langle h_{V,\tau}, P\right\rangle_\mu}{K_P K_Q}\right)$$

$$\leq \exp\left(-\eta\right)\left(1 - \alpha_{V,\tau}\right)^2 + 0.1\eta\varepsilon.$$

For $\|\Delta_A\|_\mu^2$, by (the proof of) Lemma E.5, we have

$$\left\|\Delta_{V,\tau+1}\right\|_\mu^2 \leq \exp\left(-2\eta\alpha_A^2\right)\left\|\Delta_{V,\tau}\right\|_\mu^2 + 3\eta\left(\left\|\Delta_{V,\tau}\right\|_\mu + \eta\frac{\|h_V\|_\mu}{\sqrt{K_P}K_Q}\right)\frac{\|h_V\|_\mu}{\sqrt{K_P}K_Q}$$

$$\leq \exp\left(-\eta\right)\left\|\Delta_{V,\tau}\right\|_\mu^2 + 0.1\eta\varepsilon.$$

Thus, by Lemma E.9, we have $(1 - \alpha_V)^2 \leq \varepsilon/2$ and $\|\Delta_V\|_\mu^2 \leq \varepsilon/2$ for all $\tau \geq \mathcal{T}_2 + \Theta\left(\log(1/\varepsilon)/\eta\right)$. $\square$

**Corollary F.3.** *Let $\varepsilon \in \left(\Theta(1)/(K_Q T), 0.1\right)$ be our target accuracy. Suppose that $\alpha_A \in (0.9, 1.1)$, $\|\Delta_A\|_\mu^2 \leq O(\varepsilon/(Q\sqrt{N}))$, and $\|\Delta_V\|_\mu^2 \leq 0.01$. Then with $\mathrm{poly}(N, Q, 1/\varepsilon)$ samples, we have with high probability that $\|A - Q\|_\mu^2 \leq \varepsilon$ and $\|V - P\|_\mu^2 \leq \varepsilon$ after Stage 3, which takes $O\left(\log(1/\varepsilon)/\eta\right)$ steps.*

*Proof.* It suffices to combine the previous two lemmas, the concentration results in Section B, and apply union bound. $\square$

# G   Proof of the main theorem

In this section, we combine the results from the last three sections and prove the following formal version of Theorem 3.1.

**Theorem G.1.** *Let $\varepsilon > 0$ be our target accuracy and $\mathcal{T}_1 = \min\{\tau \geq 0 : \max\{\alpha_{V,\tau}, \alpha_{A,\tau}\} \geq \Theta(1/(QN))\}$. We can choose the hyperparameters in Algorithm 1 such that within $O\left(\log(T)/\eta_1 + 1/(\eta\alpha^{(2)}) + \log(1/\varepsilon)/\eta\right)$ steps, we have $\|A - Q\|_\mu^2 \leq \varepsilon$ and $\|V - P\|_\mu^2 \leq \varepsilon$ with probability at least $1 - \delta$ and the number of samples used before and after $\mathcal{T}_1$ are $\mathrm{poly}(T, \delta)$ and $\mathrm{poly}(N, Q, 1/\varepsilon, \log T, \delta)$, respectively.*

*Proof.* The results for Stage 1 follow directly from Lemma D.1. Now, consider the results for Stage 2 and 3. First, by Corollary F.3, it suffices to make sure at time $\mathcal{T}_2$, we have $\alpha_A \in (0.9, 1.1)$, $\|\Delta_A\|_\mu^2 \leq O(\varepsilon/(Q\sqrt{N}))$ and $\|\Delta_V\|_\mu^2 \leq 0.01$ (with high probability). By Lemma D.1, we know the following hold at time $\mathcal{T}_1$ w.h.p:

$$\alpha_A, \alpha_V = \Theta(1/(QN)) \quad \text{and} \quad \|\Delta_A\|_\mu^2, \|\Delta_V\|_\mu^2 \leq O(1/T).$$

Therefore, by Lemma E.1, if we choose the threshold to be $\Theta(1/(Q^2N))$, then after thresholding and projection, we have $a_t^{(k)} = O(1/T)$ for all $t \notin \boldsymbol{q}^{(k)}$. Thus, by Lemma E.3 and Lemma E.4, with $\text{poly}(Q, N, 1/\varepsilon, \log(T))$ samples, we have with high probability that $a_t^{(k)} = O(1/T)$ for all $t \notin \boldsymbol{q}^{(k)}$ holds and $\left\|\hat{\boldsymbol{h}}_A\right\|_\mu$ satisfies the requirements in Lemma E.10 throughout Stage 2. Thus, by Lemma E.10 with $\delta_*^2 = O(\varepsilon/Q\sqrt{N})$, at the end of Stage 2, we have with high probability that $\alpha_A \in (0.9, 1.1)$, $\|\Delta_A\|_\mu^2 \leq O(\varepsilon/(Q\sqrt{N}))$ and $\|\Delta_V\|_\mu^2 \leq 0.01$. When combined with Corollary F.3, this completes the proof. $\qquad\square$

# H   Transfer learning

**Lemma H.1** (Initialization). *Suppose that we have learned $\hat{\boldsymbol{P}}$ and $\left\langle \hat{\boldsymbol{P}}, \boldsymbol{P} \right\rangle_\mu \geq 2\|\boldsymbol{\mu}\|^2$, $\|\hat{\boldsymbol{P}}\|_\mu^2 = \Theta(1)\|\boldsymbol{P}\|_\mu^2$. Let $\boldsymbol{V} = \theta\hat{\boldsymbol{P}} + (1-\theta)\boldsymbol{\mu}\boldsymbol{1}^\top$ for some $\theta \in (0, 1)$. We have*

$$\alpha_V \in \left[\Theta\left(\frac{\theta}{NK_P}\right), \Theta(\theta)\right] \quad and \quad \|\Delta_V\|_\mu^2 \leq \Theta(1)\theta^2 K_P.$$

*Proof.* First, consider $\Delta_V$. Since $\|\Delta_V\|_\mu$ is the distance to a projection, we have

$$\|\Delta_V\|_\mu^2 \leq \left\|\boldsymbol{V} - (\theta\boldsymbol{P} + (1-\theta)\boldsymbol{\mu}\boldsymbol{1}^\top)\right\|_\mu^2 = \theta^2\left\|\hat{\boldsymbol{P}} - \boldsymbol{P}\right\|_\mu^2 \leq \Theta(1)\theta^2\|\boldsymbol{P}\|_\mu^2 = \Theta(1)\theta^2 K_P.$$

For $\alpha_V$, we compute

$$\langle \boldsymbol{V}, \boldsymbol{P} \rangle_\mu - \|\boldsymbol{\mu}\|^2 = \theta\left(\left\langle \hat{\boldsymbol{P}}, \boldsymbol{P} \right\rangle_\mu - \|\boldsymbol{\mu}\|^2\right) \geq \theta\|\boldsymbol{\mu}\|^2 \geq \Theta(1)\theta/N.$$

In particular, this implies $\alpha_V \geq \Theta\left(\frac{\theta}{NK_P}\right)$. $\qquad\square$

**Lemma H.2** (First gradient step). *Let $\boldsymbol{A} = \boldsymbol{1}\boldsymbol{1}^\top/T$ and $\boldsymbol{V}$ given by Lemma H.1. Let $\varepsilon_{\text{Tmp}} \leq O(1/(NK_P))$. Run one gradient step with $\text{poly}(N, Q, \log T, 1/\varepsilon_{\text{Tmp}})$ samples and $\eta = 1$, remove all entries of $\boldsymbol{a}^{(k)}$ with $|a_t^{(k)}| \leq \Theta(\theta/(NK_P))$ and the replace $\boldsymbol{a}^{(k)}$ with the projection $(\boldsymbol{I} - \boldsymbol{1}\boldsymbol{1}^\top/T)\boldsymbol{a}^{(k)}$. With high probability, we have $\alpha_A = (1 \pm O(\varepsilon_{\text{Tmp}}))\alpha_V$ and $\|\Delta_A\|_\mu^2 \leq O\left(\frac{\varepsilon_{\text{Tmp}}^2 \alpha_V^2}{Q} + \frac{1}{T}\right)$.*

*Proof.* The proof idea is essentially the same as Lemma E.3, though the reinitialization of the first layer allows better estimations in several places. Recall that for each $s \in [T]$, we have

$$\partial_{a_s^{(k)}} l = \mathbb{1}\{x_{T+1} = k\}(\boldsymbol{V}\boldsymbol{e}_{x_s})^\top\left(\boldsymbol{V}\boldsymbol{X}\boldsymbol{1}/T - \boldsymbol{e}_{x_o}\right),$$
$$\mathbb{E}\,\partial_{a_s^{(k)}} l = \mu_k K_V/T - \mu_k K_{VP}q_s.$$

Also recall from Lemma C.3 that $K_{VP} = \alpha_V K_P$ and $K_V = \alpha_V^2 K_P + \|\Delta_V\|_\mu^2$. For any $s \in \boldsymbol{q}^{(k)}$, we have

$$-\mathbb{E}\,\partial_{a_s^{(k)}} l = \mu_k\left(\alpha_V K_P q_s - \frac{\alpha_V^2 K_P + \|\Delta_V\|_\mu^2}{T}\right) \geq \frac{\mu_k \alpha_V K_P q_s}{2}$$

where the inequality comes from $\mu_k \alpha_V K_P q_s \geq \Omega(\alpha_V/N^2/Q) \gg 1/T$. Meanwhile, for any $s \notin [T]$, we have $\mathbb{E}\,\partial_{a_s^{(k)}} l = O(1/T)$.

Meanwhile, for any $s \in [T]$ we have

$$\left| \partial_{a_s^{(k)}} l \right| \leq \left| (\boldsymbol{V} \boldsymbol{e}_{x_s})^\top \boldsymbol{V} \boldsymbol{X} \mathbf{1}/T \right| + \left| (\boldsymbol{V} \boldsymbol{e}_{x_s})^\top \boldsymbol{e}_{x_o} \right| \leq \frac{1}{T} \sum_{t=1}^{T} \left| \langle \boldsymbol{V}_{:,x_s}, \boldsymbol{V}_{:,x_t} \rangle \right| + \left| V_{x_o,x_s} \right| \leq O(N).$$

Thus, by some standard concentration argument similar to the one in Lemma E.3, we can show that with $\mathrm{poly}(N, Q, 1/\alpha_V, 1/\varepsilon_{\mathrm{Tmp}}, \log T)$ samples, we can make sure with high probability,

$$\partial_{a_t^{(k)}}^{(B_\tau)} l = \left( 1 \pm \varepsilon_{\mathrm{Tmp}} \right) \mathbb{E} \, \partial_{a_t^{(k)}} l = \left( 1 \pm \varepsilon_{\mathrm{Tmp}} \right) \mu_k \alpha_V K_P q_s \qquad \forall t \in \boldsymbol{q}^{(k)},$$

$$\partial_{a_t^{(k)}}^{(B_\tau)} l = O\left( \frac{1}{T} + \varepsilon_{\mathrm{Tmp}} \frac{\alpha_V K_P}{NQ} \right) = O\left( \varepsilon_{\mathrm{Tmp}} \frac{\alpha_V K_P}{NQ} \right) \qquad \forall t \notin \boldsymbol{q}^{(k)}.$$

Thus, after one gradient step with $\eta = 1$, we have

$$a_t^{(k)} = (1 \pm \varepsilon_{\mathrm{Tmp}}) \alpha_V q_t^{(k)} \qquad \forall t \in \boldsymbol{q}^{(k)},$$

$$a_t^{(k)} = O\left( \varepsilon_{\mathrm{Tmp}} \frac{\alpha_V}{Q} \right) \qquad \forall t \notin \boldsymbol{q}^{(k)}.$$

Recall from Lemma H.1 that $\alpha_V \in [\Theta(\theta/(NK_P)), \Theta(\theta)]$. Hence, we can choose the threshold $\lambda_0$ to be $\Theta(\theta/(NK_P))$ and $\varepsilon_{\mathrm{Tmp}} \leq 1/(NK_P)$ so that

$$\hat{\boldsymbol{a}}^{(k)} = \left[ a_t^{(k)} \mathbb{1}\{a_t^{(k)} \geq \lambda_0\} \right]_{t \in [T]} = \left[ (1 \pm \varepsilon_{\mathrm{Tmp}}) \alpha_V q_t^{(k)} \mathbb{1}\{t \in \boldsymbol{q}^{(k)}\} \right]_{t \in [T]}.$$

Now, set

$$\boldsymbol{a}^{(k)} \leftarrow (\boldsymbol{I} - \mathbf{1}\mathbf{1}^\top/T) \hat{\boldsymbol{a}}^{(k)} = (1 \pm O_\infty(\varepsilon_{\mathrm{Tmp}})) \alpha_V \boldsymbol{q}^{(k)} + O_\infty(1) \frac{\mathbf{1}}{T}.$$

Note that this implies that after the first step, we have

$$\alpha_A = \frac{\sum_k \mu_k \langle \boldsymbol{a}^{(k)}, \boldsymbol{q}^{(k)} \rangle - 1/T}{K_Q} = \frac{\alpha_V (1 \pm O(\varepsilon_{\mathrm{Tmp}})) \|\boldsymbol{Q}\|_\mu^2 + O(Q/T)}{K_Q} = (1 \pm O(\varepsilon_{\mathrm{Tmp}})) \alpha_V,$$

and

$$\|\boldsymbol{\Delta}_A\|_\mu^2 \leq \sum_{k=1}^{N} \mu_k \left\| \boldsymbol{a}^{(k)} - \left( \alpha_V \boldsymbol{q}^{(k)} + (1 - \alpha_V) \mathbf{1}/T \right) \right\|_\mu^2 \leq O\left( \frac{\varepsilon_{\mathrm{Tmp}}^2 \alpha_V^2}{Q} + \frac{1}{T} \right).$$

$\square$

**Theorem H.3** (Main theorem for transfer learning)**.** *Let $\varepsilon > 0$ be our target accuracy. Consider the same setting of Lemma H.1 and Lemma H.2. Choose $\theta = \sqrt{O(1/\log(1/\varepsilon))}$ and $\varepsilon_{\mathrm{Tmp}} = O\left( \frac{1}{\log(1/\varepsilon)} \right)$. Then, after one step of update on $\boldsymbol{A}$ as in Lemma H.2, $\boldsymbol{A}$ and $\boldsymbol{V}$ satisfies the conditions of Lemma E.10, and therefore we can learn $(\boldsymbol{P}, \boldsymbol{Q})$ to $\varepsilon$-accuracy using $\mathrm{poly}(N, Q, 1/\varepsilon, 1/\delta)$ samples with probability at least $1 - \delta$ within $\mathrm{poly}(N, Q, 1/\varepsilon, 1/\delta)$ steps.*

# I  Additional Experiment

For all our experiments, we use Numpy and run on a normal laptop which takes about 20 minutes.

**Setup.** In all our experiments, we choose $T = 5000, Q = 2, N = 3$. The architecture is

$$\boldsymbol{F}(\boldsymbol{x}, x_{T+1}; \boldsymbol{V}, \boldsymbol{A}) := \boldsymbol{V} \boldsymbol{X} \left( \boldsymbol{I}_T \boldsymbol{A} \boldsymbol{e}_{x_{T+1}} \right) =: \boldsymbol{V} \boldsymbol{X} \boldsymbol{a}^{(x_{T+1})}, \tag{15}$$

and the data model is the SCB (1) data-generating model. The batch size is $B = 64$ and the regularization hyperparameter is $\lambda = 1\mathrm{e}\text{-}5$. The total time is $\mathcal{T} = 1000$ iterations where stage 1 takes $\tau \in [0, 400]$ with learning rate $\eta_1 = 0.01$. After $\tau > 400$, we use $\eta_2 = 0.005$ for further improvement (stages 2 and 3).

**Hyperparameter selection.** Due to the limitation of computational resources, we do experiments with $N \in [3, 20]$ for real-world batched gradient experiments, and $N \in \{100, 500\}, T = 100000$ experiments by using Gaussian noise SGD simulations based on the dynamics of Lemma C.5 and C.6. As $T$ needs to scale with $N$ polynomially, it would be beyond our computation capability to

experiment with larger $N$. As for other hyperparameters, $\lambda$ is chosen based on our theoretical results (Theorem 3.1 and G.1): $\lambda \sim \Theta(\epsilon K_P/Q^2 N^3)$ in Lemma E.4. The batch size can be chosen from standard $\{64, 128, 256\}$, while smaller batch size will lead to divergence for both SGD and regularized GD. $\eta$ is chosen as the largest learning rate without divergence.

Besides the original parameters, we consider the approximation error after the normalization step (stage 3) in real-time. That is thresholding and normalizing the attention block

$$\forall k \in [n], \hat{a}^{(k)} = [a_t^{(k)} \mathbb{1}\{a_t^{(k)} \geq \Omega(1/Q)\}]_t. \ \ a^{(k)} \leftarrow \hat{a}^{(k)}/\mathbf{1}^\top \hat{a}^{(k)}$$

and we do further gradient descent to recover $V$. In this case, we will directly use the linear regression solution on population loss for $V$.

Here we report in addition: (1) original signal projection on the population process trajectory $\alpha_A$, $\alpha_V$ and the distance to the trajectory $\Delta_A, \Delta_V$. (2) The approximation error/similarity before and after normalization for both SGD and proximal gradient descent. We conclude that in all metrics proximal gradient descent performs better than the vanilla gradient descent with a small batch size (when the noise is large).

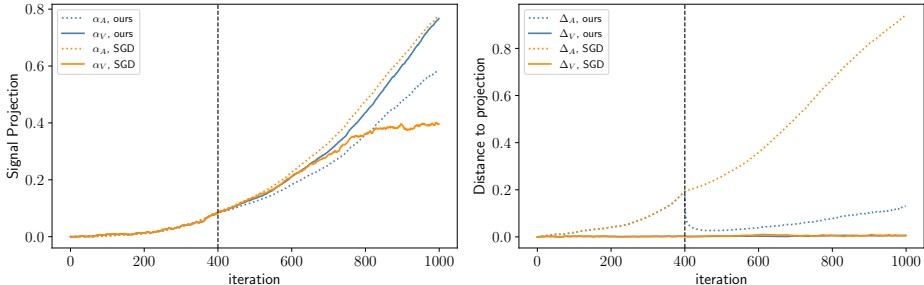

Figure 3: **Signals $\alpha_A, \alpha_V$ and the distance to population process $\Delta_A, \Delta_V$.** For the SGD, the distance to the population process of the attention matrix $A$ keeps growing and dominates the signal term. That explains the failure to learn the correct attention pattern, which leads to saturation of the signal. In comparison, our proximal methods dramatically help reduce the gradient noise and keep close to the population process. Though $\|\Delta_A\|$ eventually grows up due to the bias of the gradient estimate (the original signal growth is also slowed down), after normalization it can still approximately learn the correct pattern. Both $\|\Delta_V\|$ stay small empirically.

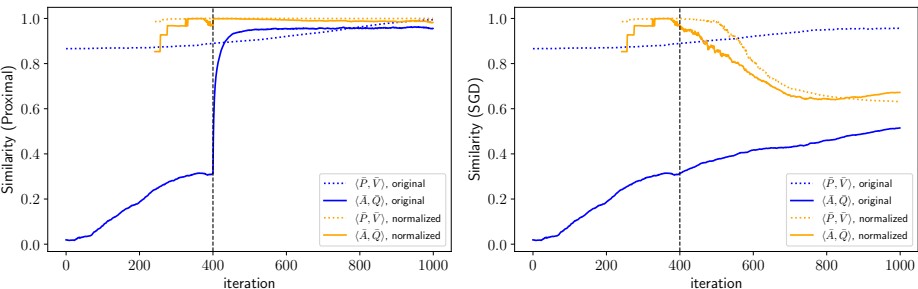

Figure 4: **Similarity with the ground-truth.** The figure shows after Stage 1, normalization helps further improve the solution of the proximal method. Meanwhile, with or without normalization, our proximal method always outperforms the vanilla SGD, which fails to recover the ground-truth.

We tried different orders of state number $N$ and show that $\ell_1$ regularization is necessary and outperforms SGD when batch size is small (gradient noise is large). Due to computation limitation, we experiment with real batched gradient on $N \leq 20, T \leq 5000$, and do SGD simulation by combining our population gradient + Gaussian noise (to mimic the batch gradient noise) for $N \leq 500, T \leq 100000$.

We also corroborate our previous experiments with the new test loss plot to show the convergence of training. Note that since there are multiple global minima for the linear attention, $\ell_1$ regularized dynamics eventually will make $A$ and $Q$ deviate from the ground-truth while representing the same function. That is why the loss converges but the distance to the ground-truth increases after some

point, making the final normalization step essential to recover the ground-truth. Another point is that according to our theory, the regularization will eventually distort the learned pattern when trained for too many iterations. Empirically, the loss also increases a little after it converges. Therefore, we must stop early and normalize before the distortion happens.

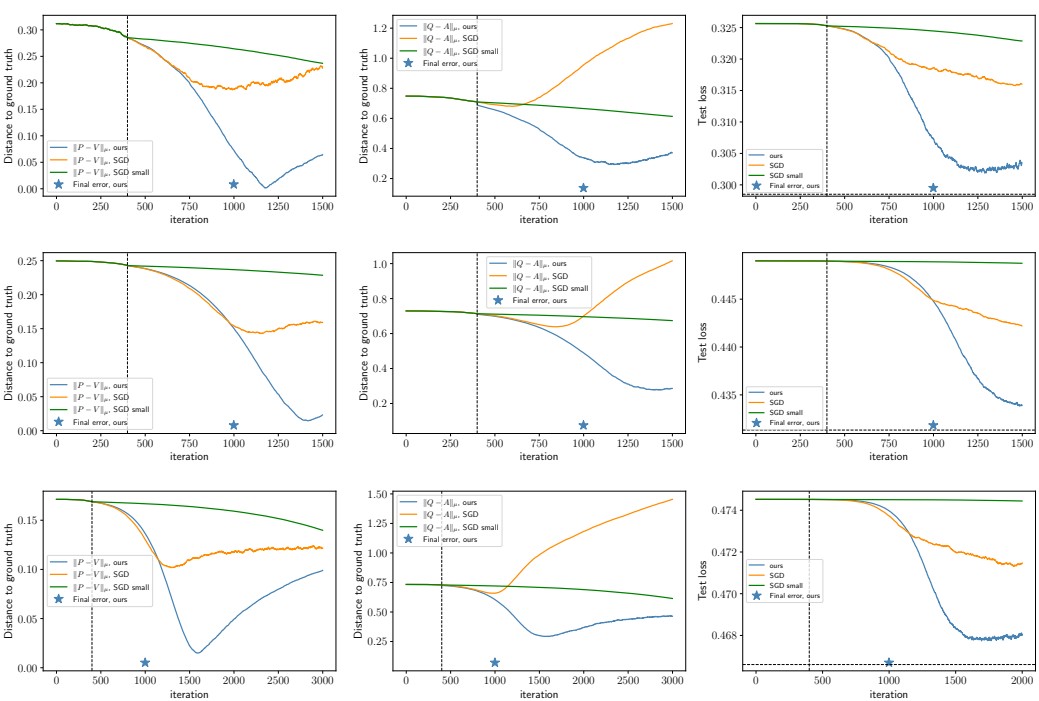

Figure 5: **Convergence analysis.** We plot the distance to the ground-truth and the test loss for $N = 3, 10, 20$ (from top to bottom). It shows that when gradient noise is large, $\ell_1$ regularized algorithm with normalization and early-stopping can almost perfectly recover the ground-truth (the star), while SGD struggles to learn the target function. Figure 3 (Appendix I) also shows when the gradient noise is large, SGD never learns ground-truth $Q$ even with normalization.

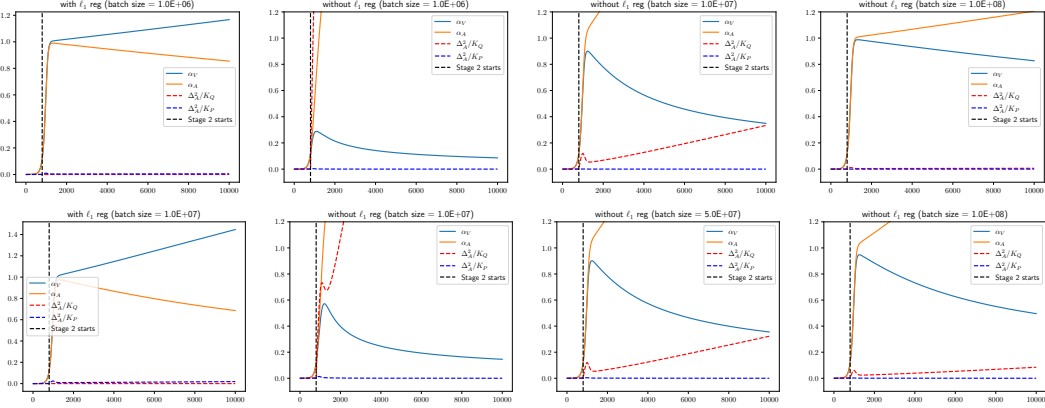

Figure 6: **Simulation with larger $N$ and $T$.** We simulate the SGD/$\ell_1$ regularized dynamics by replacing the batched noise with Gaussian noise in the dynamics formula in Lemma C.5 and C.6. The gaussian noise variance scales with the inverse of batch size. The experiments show that the conclusions drawn from the small $N$ cases still hold in those simulations: when $T = 100000$, $N = 100/500$, our $\ell_1$ regularized algorithm can recover the ground-truth since the distance to the population trajectory $(\Delta_A, \Delta_V)$ stays very small, while the error along SGD trajectories quickly increases with the same batch size.

