# OpenReview forum: "Learning and Transferring Sparse Contextual Bigrams with Linear Transformers"
_NeurIPS.cc/2024/Conference — NeurIPS 2024 poster_

### Official Review · Reviewer_9wme · 2024-07-12

**Soundness:** 4
**Presentation:** 3
**Contribution:** 2
**Rating:** 6
**Confidence:** 3

**Summary:**

The paper studies the training dynamics of a one-layer linear transformer on the sparse contextual bigram task. Using $\ell_1-$regularization and proximal gradient descent, they show that the training process goes through a sample-intensive phase and then a sample-efficient phase. They also extend the results to transfer learning.

**Strengths:**

1. The paper is well-written and organized.
2. The theory is solid. The discussion of the two phases in training dynamics is interesting.

**Weaknesses:**

1. Could the authors comment more about the choice of the initialization? Could you show, at least in experiments, the consequence of another choice of initializations?
2. The loss curve does not look like converging in Figure 1. It would be important to show the loss curve with more training rounds.

**Questions:**

1. Could the authors comment more about the choice of the setting? How does sparsity affect the results? I do not find the parameter Q to have a special effect on the results besides serving as an extra parameter.
2. Can authors give some implications of their results on the training dynamics with the complete transformer structure?

**Limitations:**

Although the theory is solid, and some findings are interesting, I do not find any important intuitions that can be gained from the author's discussion. In particular,
1. The current setting makes too many modifications to the original transformer structure and training practice. Such simplification is OK. However, the results do not facilitate understanding transformers' training under more realistic settings.
2. The paper lacks experiments under a more realistic setting with the standard transformer structure and training method.

---

> ### Author Rebuttal · Authors · 2024-08-06
>
> Thank you for your reviews. We are glad to answer your questions.
>
> ### Initialization
>
> We initialize the first layer weights with $a^{(k)}_t = 1/T$, which corresponds to the uniform attention. In practice,
> weights are initialized to be small and softmax is used. As a result, the attention pattern is also close to uniform
> at initialization. In addition, our result is stable under small perturbation at initialization. After all, since we are using
> SGD, some errors will be introduced at every step.
>
> For the second layer, we initialize $\mathbf{V} = \mathbf{\mu}\mathbf{1}^\top$. This is the trivial transition matrix with stationary
> distribution $\mathbf{\mu}$. It corresponds to the transition where the next token follows the distribution $\mathbf{\mu}$ no matter
> what the current token is. We choose this initialization to simplify the analysis. Another reasonable candidate is
> $\mathbf{V} = \mathbf{1}\mathbf{1}^\top/N$, which corresponds to choosing the next token uniformly at random. As in the situation of the
> first layer, this is what small initialization + softmax approximately implements.
>
> Since $\mathbf{\mu}$ is assumed to be well-conditioned, the difference between these two initialization schemes scales with
> $1/N$. Note that in our transfer learning result, the error also has this scale at initialization. Hence, with a similar
> argument, one can show that this uniform initialization scheme leads to the same result. The main reason behind this
> flexibility is that the population dynamics are rather stable under these type of errors (cf. line 654).
>
> ### Training curves
>
> Thanks for pointing this out. We have added the updated plots into PDF file associated with the global response.
> We can observe that with $l_1$-regularized SGD can reach a much smaller loss/distance to the ground-truth faster
> than vanilla SGD. (The distance to the ground-truth will eventually increase as the ground-truth is not the unique
> minimizer and in both cases, SGD will overshoot due to $l_1$-regularization or noise.)
>
> ### Role of the sparsity parameter $Q$
> One of the goals of the SCB model is to model the linguistic phenomenon where the next token can depend on **a few** pieces
> of the text that depend on the recent context but can potentially be far away from the current position. The sparsity
> parameter $Q$ describes the amount of long-term context on which the next token depends. We believe it is reasonable to
> assume $Q$ to be much smaller than the text length $T$, as it is hard to imagine all previous tokens can directly
> influence the next token.
>
> Then, it is natural to ask if we can obtain sample complexity that is $\mathrm{poly}(Q)$ instead of $\mathrm{poly}(T)$. Our result says
> that while it might be unlikely to obtain $\mathrm{poly}(Q)$ sample complexity when training from scratch, doing transfer
> learning can indeed reduce the sample complexity to $\mathrm{poly}(Q)$, as it allows us to skip the sample-intensive initial
> signal-boosting stage.
>
> ### Comparison with and implications on more practical architectures
> We have discussed the relationship between the training dynamics of softmax transformers and our linear model in the
> global response. In short, we expect them to be approximately equivalent, up to a change of learning rate, toward the
> beginning of training, and softmax transformers will automatically implement something similar to $\ell_1$-regularization
> in Stage~2.
>
> It might be hard to extrapolate our results to transformers with more layers rigorously. However, we want to stress that
> analyzing the simpler shallow models can be served as the first step toward understanding the more practical models,
> and this approach is widely adopted in deep learning theory and also the in the study of transformers (cf. [1-3]).
>
> Still, at a heuristic level, our transfer learning result can potentially lead to new explanation on the effectiveness
> of transfer learning and the pretraining-finetuning pattern. The usual story is during the pretraining stage, the model
> can learn representations that can be reused in downstream tasks. Our results suggest that at least in certain cases,
> this is not necessary and having some correlation between the learned and downstream features (which is much easier than
> approximately recovering the downstream features) is sufficient to greatly improve the sample complexity.
>
> [1] Alberto Bietti, Vivien Cabannes, Diane Bouchacourt, Herve Jegou, Leon Bottou. Birth of a Transformer: A Memory Viewpoint. (2023)
>
> [2] Yuchen Li, Yuanzhi Li, Andrej Risteski. How Do Transformers Learn Topic Structure: Towards a Mechanistic Understanding. (2023)
>
> [3] Ruiqi Zhang, Spencer Frei, Peter L. Bartlett. Trained Transformers Learn Linear Models In-Context. (2023)

---

> > ### Comment · Reviewer_9wme · 2024-08-10
> >
> > Thank the authors for the response. I have raised my score to 6.

---

### Official Review · Reviewer_Qmma · 2024-07-12

**Soundness:** 3
**Presentation:** 3
**Contribution:** 3
**Rating:** 7
**Confidence:** 2

**Summary:**

This paper proposes a new data model, Sparse Contextual Bigram (SCB), to study the training dynamics and sample complexity of transformers. The paper analyzes a one-layer linear transformer trained on data generated by the SCB model using a gradient-based algorithm. They prove convergence guarantees and provide bounds on the sample complexity, showing that the training process can be split into two stages: an initial sample-intensive stage and a more sample-efficient stage. The paper also includes results on transfer learning, showing that pretraining on a related task can improve sample efficiency.

**Strengths:**

* The proposed SCB model is a novel contribution that simplifies the analysis of transformer learning while still capturing essential aspects of language modeling. Especially the fact that training can be split into two stages with different data requirements and the implications for transfer learning are quite interesting.
* Although I am not familiar enough with these type of theoretical analyses to fully verify the correctness of the proofs, I found the paper relatively easy to follow.

**Weaknesses:**

* The analysis focuses on a simplified linear transformer model and a specific gradient-based algorithm. The generalizability of the results to more complex architectures and optimization methods is unclear. This limitation is acknowledged in the paper, and I don't think it's reasonable to expect a theoretical analysis without any simplifications. Hence, this might not actually be a weakness, although it is a limitation.
* The paper could benefit from more intuitive explanations of the proofs to make the theoretical results more accessible to a wider audience.

**Questions:**

* How do you think the results would change if a more complex transformer architecture (e.g., with multiple layers or non-linear activations) were used?
* Can the results on transfer learning be extended to cases where the pretraining and downstream tasks have different structures or distributions?
* Could you provide more intuition behind the proofs, either in the main text or in the appendix?

**Limitations:**

The authors have adequately addressed the limitations of their work in Appendix A. They acknowledge that the model and algorithm studied are simplified compared to those used in practice and that the empirical evaluation is limited to synthetic data. They also mention the need for further research to extend the results to more complex architectures and real-world datasets.

---

> ### Author Rebuttal · Authors · 2024-08-06
>
> Thank you for your reviews! We'd like to answer your questions as follows.
>
> ## More complex architectures
> We discuss in the global response on the relationship between our linear model and softmax transformers. At a heuristic
> level, they are approximately equivalent, up to a change of learning rate, in Stage 1, and in Stage 2, softmax
> automatically implements something similar to the $\ell_1$ regularization in our algorithm. It might be hard to extrapolate
> our results to deeper transformers rigorously. However, one can perhaps expect our message on transfer learning to hold
> at an intuitive level for deeper networks. That is, as long as pretraining can find a point that is positively
> correlated with the downstream task, which is easier than learning representations that can be directly used in downstream
> tasks, then the sample complexity can be greatly reduced.
>
> ## Transfer learning with different architectures/distributions
> One of the main benefits of our signal/correlation point of view on transfer learning is its flexibility: when the
> architecture/distribution changes, the needed representations might change as well, but in most cases, one should
> expect them to have at least a small positive correlation, and this is all we need in our theory.
>
> For example, in our setting, we only need the transition matrices of upstream and downstream tasks to be reasonably
> positively correlated and impose no correlation constraint on the $\mathbf{Q}$-matrices. In addition, our results can be
> generalized to cases where the transition matrices have different but reasonably correlated stationary distributions.
> The key here is when initializing the model as in Theorem 4.1, if both stationary distributions are well-conditioned, then
> the error should scale with $1/N$, and this amount of errors is allowed in our analysis (cf. line 654).
>
> ## Intuition on the proof
> We will discuss more on the intuition of the proof in the revision. Here is the general idea. The key equation in
> our analysis is line 654-655. If there is no noise, then it is a linear system with coefficient matrix of the linear
> term being $[[0, 1], [1, 0]]$ and drift $[1/T, 0]$. The drift term will provide the initial signal, guiding the
> dynamics toward the correct direction and then the first linear term will amplify the signal and fit the target.
> The linear term is approximately $0$ around initialization, so we have to rely on the drift term first, and to
> distinguish this $1/T$-order signal from the noise, we need $\mathrm{poly}(T)$ samples. When we have accumulated enough signals,
> the linear term will have order $1/\mathrm{poly}(Q, N)$, and we can rely on it to learn the target, which costs only $\mathrm{poly}(N, Q)$
> samples. In transfer learning, we use the pretrained model, instead of the small drift term, to get the initial
> correlation, and this allows us to skip the first sample-intensive stage.

---

> > ### Comment · Reviewer_Qmma · 2024-08-13
> >
> > Thank you for your clarifications. It'd be great if you could integrate into the next version of the paper.

---

### Official Review · Reviewer_fyHi · 2024-07-13

**Soundness:** 2
**Presentation:** 4
**Contribution:** 2
**Rating:** 5
**Confidence:** 4

**Summary:**

The topic of this paper is understanding transformers. Since this is an ambitious goal, the authors make some reasonable simplications. Specifically, they study the training of a **one-layer linear** transformer on synthetic datasets generated by a novel data-generating model, called **Sparse Contextual Bigram (SCB)**. SCB extends traditional bigram by also conditions the next token generation on prior earlier positions determined by the last token. Such design of SCB allows study on both contextual (last token conditioned early token dependence Q) and global (transition matrix P) knowledge.

Based on this data-generation model, their analysis shows the the one-layer linear transformer training process can be divided into two stages: an initial sample-intensive stage followed by a more efficient one. In transfer learning scenarios, the first stage can be skipped if there's significant correlation between the pretraining and downstream tasks.

**Strengths:**

- The paper is well-scoped. The structure is easy to follow. The authors clearly state their simplification in order to analyze the thus complicated transformer training process.
- The proposed data-generating model, SCB, is novel to me. It extends the classic bigram in a smart way so that it encompasses both contextual and global knowledge.

**Weaknesses:**

- Architectural simplification. The simplification of transformer into a one-layer linear version could make the conclusions less useful or even invalid. Particularly, the softmax over the attention weights and the layernorm before the output are known to be important for the impressive empirical performance of transformer. It would be great if the authors can explain how their simplification of architecture can impact their conclusions, either theoretically or empirically.
- Experimental setup. The experiments in Sec 5 are with a vocabulary size of N=3. Such a small vocab size is far from practical ones (~32K-250K). I doubt the conclusions on the stages would still holds across different orders of vocab size. And it is unclear to me how the hyper-parameters are chosen from line 321-line328. It would be great if the authors can explain why they use this specific setup instead of a more systematic investigation of hyper-parameters.
- Complexity. What is the complexity of the SCB data-generation model? How does it differ from a real natural language distribution?

**Questions:**

- See weakness.
- What's $I_T$ in equation 2?

**Limitations:**

They discuss the limitations in appendix A

---

> ### Author Rebuttal · Authors · 2024-08-06
>
> Thank you for your feedbacks. We'd like to address your concerns as follows.
>
> ## Meaning of $I_T$.
> $I_T$ stands for the $T$-by-$T$ identity matrix. We are using one-hot positional embeddings
> and attending only to positions here. Hence, the second $E$ is replaced by the identity matrix.
>
> ## Architectural simplification (linear vs softmax)
> We have discussed the relationship between our linear model and softmax transformers in the global response. In short,
> softmax transformers and our linear model are approximately equivalent up to a change of learning rate near
> initialization, and in Stage 2, softmax transformers implicitly implement something similar to what
> $\ell_1$-regularization does in our algorithm. In addition, analyzing the linear counterpart of the model before attacking
> the more difficult practical models is common in the development of DL theory, and in our case, this is also partially
> justified by the empirical observations in [1].
>
> ## Architectural simplification (LayerNorm)
> Indeed, LayerNorm plays an important role in the empirical success of transformers, as the pre-softmax is usually
> interpreted as the correlation of two tokens and one may wish to remove the effect of scaling via normalization. However,
> we wish to stress here that in the theoretical analysis of shallow transformers, LayerNorm is not as important as it is
> in practice and is often omitted to simplify the problem (cf. [2-4]). In particular, in our setting, the token and
> positional embeddings are fixed one-hot embeddings. They are automatically orthogonal to each other and have the same
> norm. In fact, our model is also closely related to the associative memory point of view proposed in [2], which
> interprets (the product of) the QK matrices as a dataset and the token embeddings as the key. Indeed, if the token
> embeddings are one-hot, then effectively they are accessing the corresponding entry in the QK matrix and no normalization is needed in this case.
>
> ## Experiments
> Thank you for your constructive questions on the experiments. Though the experiments is to numerically verify the theoretical results when **$T\gg N$ and the batched gradient noise is large**, we agree that a more systematic report should be added. We have added
> experimental results for different $N$ in the pdf file attached to global response.
>
> In this work, we focus on the regime where the sequence length $T\gg N$ with a relative small vocabulary size $N$, which is also used in related transformer optimization works (cf. [5]). We added new experiments for small $N\in [3,20]$, and simulation results for $N \in \{100, 500\}, T = 100000$ by using Gaussian noise SGD simulations based on the dynamics of Lemma C.5 and C.6. In all cases our theoretical conclusions still hold and $\ell_1$ regularization is necessary for this algorithm.
> As $T$ needs to scale with $N$ polynomially, it would be beyond our computation capability to experiment directly with larger $N$.
> As for other hyperparameters, $\lambda$ is chosen based on our theoretical results (Theorem 3.1 and G.1): $\lambda\sim \Theta(\epsilon K_P/Q^2N^3)$ in Lemma E.4. The batch size can be chosen from standard $\{64, 128, 256\}$, while smaller batch size will lead to divergence for both SGD and regularized GD. $\eta$ is chosen as the largest learning rate without divergence.
>
>
> ## SCB model
> I'm not sure whether I understand your meaning of the complexity of the SCB model. The purpose of this model is to
> generalize the bigram model to capture a certain linguistic phenomenon (in a simplified setting), that is, the next token can depend on a small piece
> of text that is potentially far away from it, and the location of this piece depends on the recent context, instead of providing a
> complete modeling of the natural language.
>
>
>
> [1] Kwangjun Ahn, Xiang Cheng, Minhak Song, Chulhee Yun, Ali Jadbabaie, and Suvrit Sra. Linear attention is (maybe) all you need (to understand transformer optimization). (2023)
>
> [2] Alberto Bietti, Vivien Cabannes, Diane Bouchacourt, Herve Jegou, Leon Bottou. Birth of a Transformer: A Memory Viewpoint. (2023)
>
> [3] Yuchen Li, Yuanzhi Li, Andrej Risteski. How Do Transformers Learn Topic Structure: Towards a Mechanistic Understanding. (2023)
>
> [4] Ruiqi Zhang, Spencer Frei, Peter L. Bartlett. Trained Transformers Learn Linear Models In-Context. (2023)
>
> [5] Yuandong Tian, Yiping Wang, Beidi Chen, Simon Du. Scan and Snap: Understanding Training Dynamics and Token Composition in 1-layer Transformer. (2023)

---

> > ### Comment · Reviewer_fyHi · 2024-08-10
> >
> > thank you for the detailed response. I have increased the scores.

---

### Official Review · Reviewer_yR2y · 2024-07-21

**Soundness:** 3
**Presentation:** 4
**Contribution:** 3
**Rating:** 6
**Confidence:** 3

**Summary:**

The paper studies a natural problem that generalizes the bigram model for language generation. In the bigram model, each token $x_t$  is only dependent on the previous token $x_{t-1}$. Therefore, to learn to predict the next token, one only needs to look at the bigram frequencies $P[i,j] = \Pr[x_t = i \mid x_{t-1}= j]$ which is simple to learn. The authors generalize this in two steps:


1) First there is a mapping $Q: [K] \to [T]$ that maps each token value to a sparse distribution over the indices. The algorithm picks an index and looks at the value at this index.

2) Then they use a bigram model $P$  to predict the next token based on the value chosen in step 1.


This is an interesting model and can capture natural tasks performed by transformers as they process long contexts. It requires the ability to learn the transition matrix $P$ as well as the mapping $Q$.  The authors try and show is that simplified linear model can also learn this efficiently in $N,Q$. This effectively means that they take one layer of the attention matrix and and remove soft-max and the non-linear activation in the final MLP layer. Finally they use the one hot shot embedding which greatly simplifies the model further. This results in a very simple form:
$x_{T+2} \approx VXa^{x_{T+1}}$ where $V$ is the trainable vector and $a$ is the $k^{th}$ column of a trainable matrix $A$. The key is to understand how this will be transformed using standard gradient descent and a mean squared error loss. The main result is that with the right initialization as well as a staged use of proximal gradient descent, one can learn this model using linear transformers.

**Strengths:**

The paper is well written and the assumptions are clearly stated. The analysis is mostly in the appendix but from what I have glanced at, it looks thorough. The bulk of the analysis is to deal with the noise introduced by the empirical gradient from having a small batch.  The model itself is a nice tractable model that still captures a nontrivial task that we observe performed by modern transformers.

**Weaknesses:**

Naively, one can imagine a much simpler model such as Hidden Markov Model or other sequence being capable of learning the sparse bigram model. The main interest in the paper is that a sufficiently simpler version of transformer can "provably" learn such a model.  The assumptions do seem necessary to make the problem sufficiently simple and hence approachable. However this means that the actual algorithm and model is quite far from what is used in practice. I would have liked to see a bit more empirical work to see if the introduction of non-linearity greatly reduces the sample complexity. In particular, I would have liked to see some empirical work showing that the actual attention matrix has patterns that are similar to what is learnt in the linear model itself.

**Questions:**

NA

---

> ### Author Rebuttal · Authors · 2024-08-06
>
> Thank you for your reviews! We'd like to answer your questions as follows.
>
> ## Impact of non-linearity on the sample complexity
> We discuss the relationship between softmax transformers and our linear model in the global response. Toward the
> beginning of training, where the sample complexity is potentially bad, softmax transformers and our model are approximately
> equivalent, up to a change of learning rate. As a result, we do not believe that simply adding non-linearity can
> greatly reduce the sample complexity. However, in Stage 2, softmax transformers will automatically implement something
> similar to what $\ell_1$-regularization does in our setting and will enjoy the same improved sample complexity.
>
> ## Attention pattern after training
> We add plots of the learned attention patterns after training in the PDF file attached to the global response. In short,
> they match what our theory predicts and recovers the $\mathbf{Q}$-matrix. The pattern that linear and softmax attention learns are very similar, and converges to the solutions with similar losses.

---

### Author Rebuttal · Authors · 2024-08-06

## Relationship between our model and softmax transformers

As several reviewers
asked about the difference between the softmax transformers and our linear model, we'd like to further discuss their relationship. In short, they have qualitatively similar behaviors: there will be sample-intensive
initial stage, and when there is nontrivial correlation between the model and the target, SGD will become much more
sample efficient. In the following, for simplicity, we will assume $N = 1$, write $a := a^{(1)}$, and the ground-truth
$q$ is $e_1 = (1, 0, \dots, 0)$. Most of our argument bellow can be generalized to the general setting at least at a
heuristic level. In our linear model $f(X) = VXa$, we directly optimize $a$ while keeping the constraint $\sum_t a_t = 1$.
By the softmax transformer, we mean the model $f(X) = VX\sigma(w) =: VXa$ where $\sigma$ is the softmax function
and $w \in \mathbb{R}^{T}$ is the trainable weights.

Let $l$ denote the (per-sample) loss. Since the Jacobian of the softmax function is $\mathrm{diag}(a) - aa^\top$,
we have
$$
  \nabla_{a} l(f(X))
  = (VX)^\top \nabla l(f(X)),
  \quad
  \nabla_{w} l(f(X))
  = \left( \mathrm{diag}(a) - aa^\top \right) (VX)^\top\nabla l(f(X)).
$$.
As a result, the dynamics of the attention weights $a$ are controlled by
$$
  a(\tau+1)
  \approx \begin{cases}
    a(\tau) - \eta \left(I - \frac{\mathbf{1}\mathbf{1}^\top}{T}\right) (VX)^\top \nabla l(f(X)), & \text{in our linear model}, \\\\
    %
    a(\tau) - \eta \left( \mathrm{diag}(a) - aa^\top \right)^2 (VX)^\top\nabla l(f(X)),
    & \text{in softmax transformers}.
  \end{cases}
$$
In other words, the main difference is that there will be a preconditioning matrix $\left( \mathrm{diag}(a) - aa^\top \right)^2$ in the dynamics of softmax transformers. In particular, this implies with adapative second-order methods,
the training dynamics of these two models can potentially be very similar. Moreover, even with SGD, there is a close
relationship between their training dynamics, which we will now explain.

Near initialization, i.e., when the attention pattern is still close to the uniform attention, we have
$\left( \mathrm{diag}(a) - aa^\top \right)^2 \approx \frac{1}{T^2} \left(I - \frac{\mathbf{1}\mathbf{1}^\top}{T}\right)$.
In other words, our linear model and softmax transformers are approximately equivalent, up to a change of learning rates.

Now, suppose that there is a nontrivial correlation between $a$ and $q = e_1$, say, $a_1$ is a small constant while
all other entries are $O(1/T)$. In this case, we have $\left( \mathrm{diag}(a) - aa^\top \right)^2 \approx
a_1(1 - a_1) e_1 e_1^\top + O(1/T)$. Effectively, softmax transformers automatically adjust the learning rate
according to $a_t$ and roughly ignore those positions with a small attention weight to stabilize the gradients.
Note that this is also what $\ell_1$-regularization does in our algorithm. In fact, mimicking this behavior is one of
the motivations of using $\ell_1$-regularization.

## Empirical results
Empirically, we run further experiments to stress the resemblance between softmax attention and our linear attention model (with full attention to both the token and positional encoding parts, see the attached PDF). For the effective token-position attention block, both softmax and linear attention has the same pattern, succeeded in recovering the ground-truth.

We thank Reviewer fyHi and 9wme for providing constructive suggestions on the drawbacks of our experiments, and follow-up experiments are added in the attached PDF. We tried different orders of state number $N$ and show that $\ell_1$ regularization is necessary and outperforms SGD when batch size is small (gradient noise is large).  Due to computation limitation, we experiment with real batched gradient on $N\leq 20, T\leq 5000$, and do SGD simulation by combining our population gradient + gaussian noise (to mimick the batch gradient noise) for $N\leq 500, T\leq 100000$.

We also corroborate our previous experiments with the new test loss plot to show convergence of training. Note that since there are multiple global minima for the linear attention, $\ell_1$ regularized dynamics eventually will make $A$ and $Q$ deviate from the ground-truth while representing the same function. That is why the loss converges but the distance to the ground-truth increases after some point. That's why we need the final normalization step to recover the ground-truth. Another point is that according to our theory, the regularization will eventually distort the learned pattern when trained for too many iterations. Empirically, the loss also increases a little after it converges. Therefore, we need to early stop and do the normalization before the distortion happens.

---

### Decision · Program_Chairs · 2024-09-25

**Decision:**

Accept (poster)

**Comment:**

This paper tries to improve our understanding of training dynamics of transformer-based language models. In this regard, the authors propose a novel Sparse Contextual Bigram (SCB) model and use it to analyzes the dynamics. In this model, it is shown that training can be split into two stages: an initial sample-intensive stage and a more sample-efficient stage. This existence of two stages has many consequences, for example in transfer learning scenarios, the first stage can be skipped if there's significant correlation between the pretraining and downstream tasks. The reviewers generally found the paper to be technically sound and well-presented, with novel theoretical contributions. However, a few limitations and concerns were also raised: practical relevance of the SCB data generation model, simplification of the transformer architecture, and limited experimental validation. We thank the authors and reviewers for engaging during the discussion period towards improving the paper. In their rebuttal, the authors addressed many of these concerns, providing additional experimental results, discussing connections to softmax transformers, and offering more intuition behind the proofs. Overall, the paper presents a novel and rigorous theoretical analysis that contributes to our understanding of transformer training dynamics. While the simplifications made limit direct practical applications, the work provides valuable insights and a foundation for further research on more complex transformer architectures.